# A Narrowing Geometry in Contaminated Reasoning

**Jiakuan Xie** [1 2]   **Pengfei Cao** [1 2]   **Kang Liu** [1 2]   **Jun Zhao** [1 2]

## Abstract

Despite the advancing reasoning capabilities of large language models (LLMs), many reasoning evaluations are increasingly compromised by data contamination, which induces unreliable **contaminated reasoning** on leaked inputs. While this phenomenon is widely observed, its underlying mechanism remains poorly understood, hindering the ability to distinguish generalization from memorization and to develop effective solutions. In this work, we first identify a distinctive signal of contaminated reasoning, namely the mutual information decay between representations and gradients. Our mechanistic analysis reveals that contaminated models exhibit pronounced eigenspectrum concentration in their representations, leading to a low-dimensional computation regime. Under leaked inputs, this mechanism weakens the linear coupling between representations and gradients, manifested as a structural decay of the singular values in the whitened space. We show that this narrowing geometry mathematically implies a reduction in mutual information, and further demonstrate the practical utility of our analysis by successfully restoring the reasoning behavior of contaminated models, achieving up to an 11.03% absolute improvement in average consistency with the base model over the strongest baseline. Our code is available at `https://github.com/jiakuan929/ContamReasoning`.

## 1. Introduction

Reasoning capabilities have become a central focus in the development of large language models (LLMs). Techniques such as chain-of-thought (CoT) prompting (Wei et al., 2022;

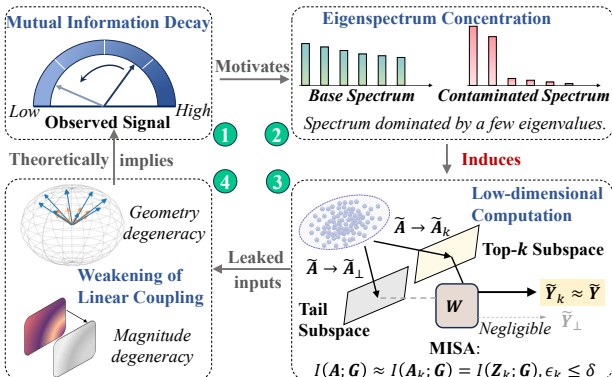

*Figure 1.* Analytical framework and key findings of our analysis.

Kojima et al., 2022; Wang et al., 2023a; Zhang et al., 2023; Wang et al., 2023b; Zhou et al., 2023; Yao et al., 2023) enable models to decompose problems into step-by-step solutions, significantly enhancing accuracy across numerous tasks (Suzgun et al., 2023; Shi et al., 2023; Zhang et al., 2024). Related efforts also explore collaborative multi-step reasoning frameworks (Zhang et al., 2025b). This progress has been further accelerated by the emergence of specialized large reasoning models (Jaech et al., 2024; Guo et al., 2025), which consistently push the performance frontier. However, strong benchmark performance does not necessarily imply reliable reasoning behavior. A growing body of work has revealed severe data leakage in widely used benchmarks, under which models can accurately reconstruct data examples and exhibit memorization behavior (Wu et al., 2026). This problem undermines evaluation integrity and obscures our understanding of model capabilities.

In response, prior studies have primarily focused on detecting data contamination (Li, 2023; Tu et al., 2024; Liu et al., 2024; Xu et al., 2024; Shi et al., 2024; Zhang et al., 2025a) or constructing leakage-free benchmarks (Jain et al., 2025; White et al., 2025; Chen et al., 2025; Wu et al., 2026), while others attempt to alleviate leakage effects through response filtering (Dong et al., 2024) or decoding-time intervention (Hou et al., 2025). Nevertheless, these approaches largely operate at the dataset or output level. The internal mechanisms by which models perform reasoning in the presence of data leakage remain poorly understood. In this paper, we refer to this setting as **contaminated reasoning**, i.e., reasoning behavior influenced by unintended exposure

[1]School of Artificial Intelligence, University of Chinese Academy of Sciences, Beijing, China [2]The Key Laboratory of Cognition and Decision Intelligence for Complex Systems, Institute of Automation, Chinese Academy of Sciences, Beijing, China. Correspondence to: Jun Zhao <jzhao@nlpr.ia.ac.cn>.

*Proceedings of the 43rd International Conference on Machine Learning*, Seoul, South Korea. PMLR 306, 2026. Copyright 2026 by the author(s).

to evaluation-relevant content during training. This lack of mechanistic understanding poses major challenges in at least two aspects. First, we cannot assess whether seemingly strong reasoning performance reflects genuine capability, since the fluent CoT rationales can be generated even when the model relies on memorization. Second, it remains difficult to design robust evaluation frameworks that remain faithful under data leakage. We argue that this missing mechanistic perspective is a key reason why data contamination continues to hinder current research in LLM reasoning.

Motivated by this gap, we take a first step toward understanding contaminated reasoning by investigating how data contamination alters the model's internal representations and computational mechanisms. The analytical framework and key findings are schematized in Figure 1. We begin by introducing a quantitative signal to characterize contaminated reasoning. Inspired by prior work linking model memorization to the loss landscape (Garg et al., 2024; Ravikumar et al., 2024; Jeon et al., 2025; Merullo et al., 2025), we propose measuring the mutual information between a module's input activations and its output gradients. Empirically, we observe a pronounced mutual information decay: on the contaminated model, the mutual information for leaked inputs is consistently lower than that for clean inputs, whereas on the base model, the two yield indistinguishable mutual information. This discrepancy suggests that the observed pattern is a distinctive signature of contaminated reasoning. To interpret this phenomenon, we then analyze the internal mechanism shifts induced by data leakage from both representational and computational perspectives. At the representation level, we find that activations in contaminated models exhibit strong **eigenspectrum concentration**, where the covariance eigenspectrum is dominated by a small number of leading eigenvalues, resulting in a low effective rank. At the computation level, we show that this representational concentration causally induces a **low-dimensional computation** mechanism that the module output becomes largely determined by a restricted set of principal directions. We establish this connection through a mathematical analysis as well as spectrum-reshaping interventions.

Taken together, these shifts reveal a *narrowing geometry* in contaminated reasoning, where both representations and computations concentrate into a subspace. Building on these observations, we provide a theoretical explanation for the mutual information decay with consistent experimental verification. We show that the original mutual information can be effectively confined to an active computation subspace, a property we refer to as **mutual information subspace alignment**. This alignment is consistently supported by our experimental analyses. Based on this alignment, the low-dimensional computation restricts the linear coupling between representations and gradients to this subspace. When leaked inputs are presented, optimization-induced

gradient shrinkage further weakens this correlation, leading to a **weakening of linear coupling**. Importantly, as another facet of the narrowing geometry, this weakening manifests not merely as a reduction in overall magnitude, but as a structured decay in the singular value spectrum of the whitened cross-covariance. We formalize this geometry property through a **singular value partial order** condition, which mathematically implies the decay of mutual information, thereby completing a mechanistic chain from data contamination to degraded reasoning signals.

Finally, to demonstrate the practical utility of our analysis, we apply these mechanistic findings to restore the reasoning behavior of the contaminated models. Through quantitative evaluation and a qualitative case study, we show that our mechanism-driven intervention successfully recovers approximately half of the leaked samples while incurring no more than a 6% consistency drop on clean data, achieving the highest average consistency score with the base model compared to baselines. These results highlight the practical potential of our insights.

In summary, our primary contributions are as follows: **(1)** We identify a distinct mutual information decay that characterizes contaminated reasoning. Motivated by this observation, we uncover and empirically validate the underlying representational and computational shifts in contaminated models. **(2)** We provide a comprehensive explanation for the mutual information decay through theoretical derivation and experimental validation. **(3)** Building on our analysis, we develop an application that successfully restores the contaminated models for approximately half of samples while preserving consistency on clean data, thereby demonstrating the potential utility of our analysis.

## 2. Preliminaries

### 2.1. Theoretical and Experimental Setup

**Theoretical Notation.** We denote the input activation of a linear module as $A$, its weight matrix as $W \in \mathbb{R}^{d_i \times d_o}$, and the module's output as $Y$. The gradient of the loss with respect to $Y$ is denoted as $G$. A centered variable, obtained by subtracting the mean $\mu$, is denoted by a tilde (e.g., $\tilde{A}$).

**Experimental Configuration.** We conduct experiments using three models: Qwen2.5-1.5B-Instruct, Qwen2.5-7B-Instruct (Qwen et al., 2025) and Llama3.1-8B-Instruct[1]. Our primary analysis uses the GSM8K test set (Cobbe et al., 2021), which we partition into a *leaked* subset for contamination injection via supervised fine-tuning and a remaining *clean* subset that is held out from the injection process. To ensure controlled analysis, we also synthesize a multi-step calculation (MSCalc) dataset, inspired by prior work (Wu

---

[1] https://ai.meta.com/blog/meta-llama-3-1/

et al., 2026). The data synthesis process is shown in Appendix A. The results reported in the main text focus on the Qwen model and GSM8K dataset, with consistent findings across other configurations documented in Appendix C.

**Implementation Details.** In the main text, our mechanistic analyses primarily focus on the linear projection module `mlp.up_proj`, with additional results across other representative linear projections (e.g., `self_attn.q_proj`) provided in Appendix C. For each input question, we feed the prompt into the model and extract the input activations of the selected linear projection at each generation step across all layers. This yields an activation matrix $\boldsymbol{A} \in \mathbb{R}^{N \times d_i}$ per layer, where $N$ denotes the number of generated tokens and $d_i$ is the input dimension of the projection. We then compute the cross-entropy loss under teacher forcing on the generated chain-of-thought response, perform backpropagation, and collect the gradient $\boldsymbol{G}$ with respect to the module output $\boldsymbol{Y} \in \mathbb{R}^{N \times d_o}$ for each layer.

## 2.2. Preliminary Observation

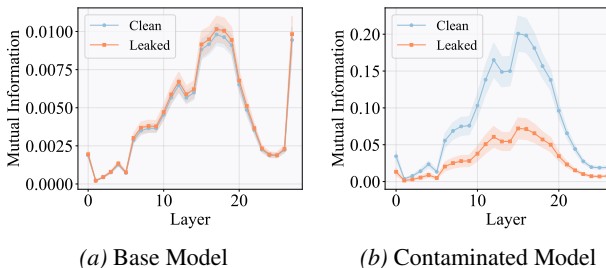

*(a)* Base Model          *(b)* Contaminated Model

*Figure 2.* Mutual information across all layers for the base model and the contaminated model (Qwen2.5-1.5B-Instruct).

To investigate contaminated reasoning, we first ask whether there exists a quantitative signal that can effectively characterize this behavior. Given that data contamination induces memorization (Wu et al., 2026) reflected in the loss landscape (Garg et al., 2024; Ravikumar et al., 2024; Jeon et al., 2025; Merullo et al., 2025), we focus on the statistical dependence between activations and gradients, proposing the mutual information as a quantitative signal to characterize this behavior. In practice, we compute Gaussian mutual information (Wen, 2024), defined as follows:

$$I\left(\boldsymbol{A}; \boldsymbol{G}\right) = \frac{1}{2} \log \frac{\det\left(\boldsymbol{C_A}\right) \det\left(\boldsymbol{C_G}\right)}{\det\left(\boldsymbol{C}\right)}, \qquad (1)$$

where $\boldsymbol{C_A}$ and $\boldsymbol{C_G}$ are the covariances of $\boldsymbol{A}$ and $\boldsymbol{G}$, and $\boldsymbol{C}$ is their joint covariance.

We compute the mutual information for both the base and contaminated models on leaked and clean data, as shown in Figure 2. For the base model, the mutual information values on leaked and clean data are nearly indistinguishable. In contrast, for the contaminated model, a clear separation

emerges: the mutual information on leaked data is consistently lower than that on clean data, a phenomenon we refer to as **mutual information decay**. These results indicate that mutual information serves as an effective signal for characterizing contaminated reasoning, and that mutual information decay is a distinguishing phenomenon specific to contaminated models under data leakage. This finding motivates our subsequent mechanistic analysis.

## 3. Mechanistic Analysis

In this section, we characterize the mechanistic transition of the contaminated model. From the representation perspective, an analysis of the model's internal activations reveals a pronounced concentration of the covariance eigenspectrum, which suggests that the activations of the contaminated model are restricted to a constrained subspace. From the computation perspective, we further observe a low-dimensional computation mechanism, wherein the output of an internal module is predominantly governed by a small set of principal eigenvectors (§3.2). We provide empirical evidence linking these two phenomena, showing that the emergence of spectral concentration plays a causal role in driving the transition toward low-dimensional computation (§3.3). Building on these mechanistic findings, we further present a theoretical explanation for the mutual information decay with experimental verification (§3.4).

### 3.1. Eigenspectrum Concentration

We begin by analyzing the mechanistic shift of the contaminated model from the perspective of its internal representations. A central quantity characterizing the representational structure is the activation covariance matrix. For a given centered activation matrix $\tilde{\boldsymbol{A}}$, the covariance $\boldsymbol{C_A}$ and subsequent eigendecomposition are defined as:

$$\boldsymbol{C_A} = \mathbb{E}\left[\tilde{\boldsymbol{A}}^\top \tilde{\boldsymbol{A}}\right] = \boldsymbol{U} \boldsymbol{\Lambda} \boldsymbol{U}^\top, \qquad (2)$$

where $\boldsymbol{\Lambda} = \mathrm{diag}\left(\lambda_1, \ldots, \lambda_d\right)$ contains the eigenvalues, and the columns of $\boldsymbol{U}$ are the corresponding orthonormal eigenvectors. To quantify the concentration of the eigenspectrum, we introduce the Eigenspectrum Concentration Index (ECI):

$$\mathrm{ECI} = \exp\left(-p_i \sum_i \log p_i\right), \quad p_i = \frac{\lambda_i}{\sum_j \lambda_j}. \qquad (3)$$

A lower ECI indicates a more concentrated spectrum, where the total variance is dominated by a few leading eigenvalues.

The results of the spectrum geometry analysis are presented in Figure 3. We observe that the ECI of the contaminated model is consistently and substantially lower than that of the base model, indicating a more concentrated covariance eigenspectrum. This finding shows that, after contamination

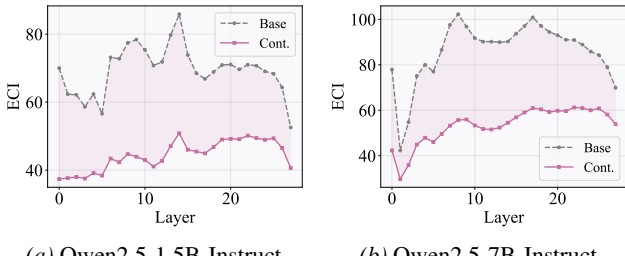

*(a)* Qwen2.5-1.5B-Instruct     *(b)* Qwen2.5-7B-Instruct

*Figure 3.* ECI across all layers for the base and contaminated (Cont.) models. The shaded area highlights that the eigenspectrum of the contaminated model is significantly more concentrated.

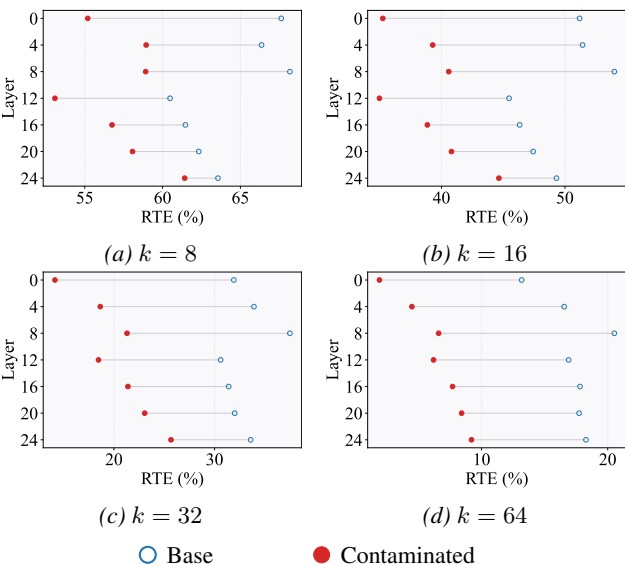

*(a)* $k = 8$      *(b)* $k = 16$

*(c)* $k = 32$      *(d)* $k = 64$

○ Base     ● Contaminated

*Figure 4.* Comparison of the RTE between base and contaminated models across a range of $k$ values (Qwen2.5-7B-Instruct).

training, the variance becomes disproportionately concentrated in a small number of leading eigenvalues, reflecting a systematic shift in the model's representational structure.

## 3.2. Low-Dimensional Computation

The observed low-rank geometry of the representations further motivates the hypothesis that the forward computation of the contaminated model may likewise exhibit a low-dimensional structure. Intuitively, for an internal module, the directions in the input space that are associated with the top-$k$ eigenvalues may alone be sufficient to capture its output activations. To examine this hypothesis, we project the centered input activation $\tilde{A}$ onto the subspace spanned by the top-$k$ principal eigenvectors, and compute the module output corresponding to this projected component:

$$\tilde{Y}_k = \tilde{A} U_k U_k^\top W = \tilde{A}_k W, \tag{4}$$

where $U_k = [u_1, \ldots, u_k]$ denotes the principal eigenvectors corresponding to the top-$k$ eigenvalues. The relative

error between the complete $\tilde{Y}$ and $\tilde{Y}_k$ can be expressed as:

$$\epsilon_k = \frac{\left\| \tilde{Y} - \tilde{Y}_k \right\|^2}{\left\| \tilde{Y} \right\|^2} = \frac{\sum_{i=k+1}^{d} \lambda_i \left\| W^\top u_i \right\|^2}{\sum_{j=1}^{d} \lambda_j \left\| W^\top u_j \right\|^2}. \tag{5}$$

The proof of Equation (5) is provided in Appendix B.1. The formula shows that the relative error is primarily determined by the tail eigenspectrum beyond the top-$k$ components, i.e., the eigenvalues and eigenvectors associated with the last $d - k$ dimensions. We therefore refer to this quantity as the relative truncation error (RTE). For a fixed $k$, a smaller value of this term implies a smaller discrepancy between $\tilde{Y}$ and $\tilde{Y}_k$, meaning that the module output can be well-approximated using only the top-$k$ directions.

From Figure 4, we make the following observations. (1) For a fixed $k$, the RTE of the contaminated model is consistently smaller than that of its base counterpart, providing strong empirical evidence that contamination induces a more pronounced low-dimensional computation regime relative to the base model. (2) As $k$ increases, the RTE decreases. In particular, when $k = 64$ (a mere 1.79% of the total directions), the truncation error of the contaminated model becomes notably small, with an average value below 10%.

## 3.3. From Spectral Concentration to Low-Dimensional Regime

### 3.3.1. MATHEMATICAL ANALYSIS

Having established the existence of a low-dimensional computation mechanism, we now bridge the gap between this mechanism and the observed eigenspectrum concentration. As a geometric property of the input activations, spectral concentration may induce such low-dimensional behavior.

Equation (5) reveals that RTE depends not only on activation eigenvalues but also on the product of $W^\top$ and the corresponding eigenvectors. While eigenspectrum concentration diminishes tail eigenvalues, it does not constrain the norm of the transformed vectors $W^\top u_i$. Consequently, a large norm can compensate for a small eigenvalue, preventing the RTE from decaying.

Building upon the aforementioned considerations, it becomes essential to verify a critical condition: the weight matrix $W$ does not significantly amplify the tail eigenvectors. Formally, let $\beta_i = \left\| W^\top u_i \right\|$, we therefore compare this quantity between the tail and principal eigenvector groups. To this end, we introduce two metrics:

$$R_{\mathrm{worst}} = \frac{\max_{i>k} \beta_i}{\min_{j\leq k} \beta_j}, \quad R_{\mathrm{med}} = \frac{\mathrm{med}_{i>k} \beta_i}{\mathrm{med}_{j\leq k} \beta_j}. \tag{6}$$

Here, $R_{\mathrm{worst}}$ characterizes the worst-case norm ratio, measuring the maximum transformed tail-direction norm relative to the minimum principal-direction norm, while $R_{\mathrm{med}}$

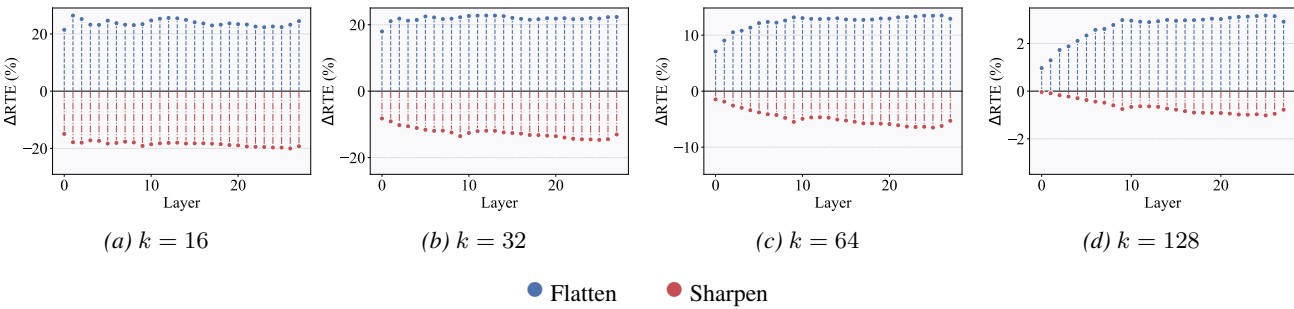

*(a) k = 16*  *(b) k = 32*  *(c) k = 64*  *(d) k = 128*

● Flatten  ● Sharpen

*Figure 5.* Causal effects of eigenspectrum concentration on low-dimensional computation in Qwen2.5-7B-Instruct. $\Delta$RTE quantifies the deviation in RTE following spectrum reshaping. Flatten ($\tau = 0.5$) reduces concentration, while Sharpen ($\tau = -0.5$) enhances it.

*Table 1.* Norm ratios between transformed tail eigenvectors and principal eigenvectors for Qwen2.5-7B-Instruct. For $R_{\text{worst}}$ and $R_{\text{med}}$, the maximum and median values across all layers are reported, respectively.

| MODELS | METRICS | $k$ | | | |
|---|---|---|---|---|---|
| | | 8 | 16 | 32 | 64 |
| BASE | $R_{\text{worst}}$ | 1.82 | 1.80 | 1.78 | 1.76 |
| | $R_{\text{med}}$ | 0.93 | 0.94 | 0.95 | 0.98 |
| CONTAMINATED | $R_{\text{worst}}$ | 1.71 | 1.69 | 1.68 | 1.60 |
| | $R_{\text{med}}$ | 0.95 | 0.95 | 0.96 | 0.98 |

captures a typical-case norm ratio based on the median values. When these ratios do not significantly exceed 1, we consider the condition satisfied that $\boldsymbol{W}$ does not unduly amplify the tail eigenvectors.

For layer aggregation, we compute the two metrics at each layer, and report the maximum of $R_{\text{worst}}$ and the median of $R_{\text{med}}$ across layers. As shown in Table 1, the maximum $R_{\text{worst}}$ of the contaminated model is 1.71, while the maximum $R_{\text{med}}$ is 0.98, indicating limited tail amplification even in the worst case. Comparable ratios are observed for the base model, suggesting that the weight does not induce a systematic preference for tail directions. These observations demonstrate that eigenspectrum concentration is crucial for inducing the low-dimensional computation mechanism.

### 3.3.2. CAUSAL VALIDATION

To provide additional causal evidence for our analysis, we conduct an intervention experiment in which we explicitly modify the distribution of the activation eigenspectrum and examine its effect on the relative truncation error. The underlying causal hypothesis is that if eigenspectrum concentration drives low-dimensional computation, then making the spectrum less concentrated should weaken this effect and increase the RTE, whereas further concentrating the spectrum should strengthen it and decrease the RTE.

When reshaping the spectrum, we explicitly control for other factors that may affect the RTE by enforcing the fol-

lowing constraints: (1) the eigenvector directions are kept unchanged; and (2) the total variance (i.e., the sum of all eigenvalues) is preserved. In this way, the intervention modifies only the relative magnitudes of the eigenvalues while keeping all other potential confounding factors fixed, thereby isolating the effect of spectral reshaping.

Given $\boldsymbol{C_A} = \boldsymbol{U}\boldsymbol{\Lambda}\boldsymbol{U}^\top$, we modify the eigenvalues as

$$\lambda_i' = \lambda_i^{1-\tau}, \qquad \tau \in [-1, 1], \tag{7}$$

and rescale them to preserve total variance:

$$S = \sum_{i=1}^{d} \lambda_i, \;\; S' = \sum_{i=1}^{d} \lambda_i', \;\; \lambda_i'' = \lambda_i' \cdot S/S'. \tag{8}$$

The intervention is then implemented by

$$\boldsymbol{A}' = (\boldsymbol{A} - \boldsymbol{\mu})\boldsymbol{U} \operatorname{diag}\left(\sqrt{\frac{\lambda_1''}{\lambda_1}}, \ldots, \sqrt{\frac{\lambda_d''}{\lambda_d}}\right) \boldsymbol{U}^\top + \boldsymbol{\mu}. \tag{9}$$

In this transformation, a positive value of $\tau$ reduces the relative differences across eigenvalues, making the eigenspectrum less concentrated. Conversely, when $\tau < 0$, the relative differences between eigenvalues are amplified, resulting in a more concentrated spectrum. We evaluate both cases in our intervention experiments by applying $\tau = \pm 0.5$.

We compute the deviation $\Delta$RTE after spectrum reshaping, i.e., the difference between the reshaped and original RTE values. The results are shown in Figure 5, from which we make the following observations. (1) For a fixed $k$, when the spectrum is made less concentrated (Flatten), the RTE increases, whereas making the spectrum more concentrated (Sharpen) leads to a decrease in RTE. This provides causal evidence that the degree of eigenspectrum concentration directly influences the RTE. (2) As discussed earlier (Figure 4), in the original case $k = 64$ is sufficient to keep the truncation error below $10\%$. However, a less concentrated spectrum necessitates a larger $k$ (e.g., $k = 128$) to preserve the error threshold, indicating an expansion of the computational geometry. Conversely, when the spectrum becomes

more concentrated, the underlying geometry narrows, allowing a smaller $k$ (e.g., $k = 32$) to suffice for certain layers and reflecting a reduction in the effective dimensionality. Together, these results strengthen the causal link between eigenspectrum concentration and the emergence of low-dimensional computation.

## 3.4. Explanation of Mutual Information Decay

This section integrates our findings to explain mutual information decay. Our analysis (§3.4.1) shows that low-dimensional computation, combined with reduced gradient projections on leaked data, weakens the linear coupling between representations and gradients. Beyond a simple magnitude reduction, this weakening exhibits a structured decay of the singular value spectrum in the whitened space. Experimental results (§3.4.2) substantiate these analyses.

### 3.4.1. MECHANISTIC EXPLANATION

Under the low-dimensional computation mechanism, the module output is predominantly determined by a small set of principal eigenvectors. Formally, the computation process and the activation decomposition can be expressed as:

$$\tilde{Y} \approx \tilde{A}_k W = \tilde{A} U_k U_k^\top W = Z_k V_k,$$
$$\tilde{A} = \tilde{A} \left( U_k U_k^\top + U_\perp U_\perp^\top \right) = Z_k U_k^\top + Z_\perp U_\perp^\top, \quad (10)$$

where $Z_k = \tilde{A} U_k$ denotes the projection coefficients corresponding to the principal output basis $V_k$, and $U_\perp$ contains the tail eigenvectors. This transformation is of fundamental importance, as the forward computation is executed almost entirely within the subspace spanned by $U_k$. Consequently, $Z_k$ serves as the effective representation that is operatively propagated through the module. In contrast, the original activation $A$ also contains components orthogonal to $U_k$, which have negligible impact on the module output and therefore exert minimal functional influence on the gradient. Since the statistical dependence between $A$ and $G$ is mediated exclusively through $Z_k$, the mutual information is determined primarily within the active computation subspace. This motivates the **Mutual Information Subspace Alignment (MISA)**:

$$I\left(A; G\right) \approx I\left(A_k; G\right) = I\left(Z_k; G\right) \quad \left(\epsilon_k \le \delta\right). \quad (11)$$

The proof is provided in Appendix B.2. Thus, it remains to be elucidated why $I\left(Z_k; G\right)$ decreases specifically on leaked data.

The key difference between leaked and clean data is whether the sample has been included in training; thus, the gradient provides a natural distinguishing signal. Let $N$ denote the number of samples, and consider the average centered gradient of the loss function with respect to the weights:

$$\tilde{\nabla}_W L = \frac{1}{N} \tilde{A}^\top \tilde{G}_Y = \frac{1}{N} \left( U_k Z_k^\top + U_\perp Z_\perp^\top \right) \tilde{G}_Y$$
$$= U_k \left( \frac{1}{N} Z_k^\top \tilde{G}_Y \right) + U_\perp \left( \frac{1}{N} Z_\perp^\top \tilde{G}_Y \right)$$
$$= U_k C_{Z_k G} + U_\perp C_{Z_\perp G}. \quad (12)$$

Left-multiplying by $U_k^\top$ yields

$$U_k^\top \tilde{\nabla}_W L = C_{Z_k G}. \quad (13)$$

Equation (13) shows that the linear coupling between $Z_k$ and $G$ is exactly the projection of the weight gradient onto the subspace $U_k$. Consequently, this coupling is not uniformly distributed across all directions but is structurally concentrated in $U_k$, the subspace that the model relies on for forward computation. This correspondence is the structural foundation provided by the low-dimensional computation mechanism. On optimized leaked inputs, weight gradient shrinkage (empirically verified in Appendix C.5) reduces the projected component, thereby weakening the linear coupling within the subspace. We term this joint effect of low-dimensional computation and gradient shrinkage **Weakening of Linear Coupling (WLC)**, expressed as:

$$\left\| C_{Z_k G}^{(l)} \right\|_F < \left\| C_{Z_k G}^{(c)} \right\|_F. \quad (14)$$

Inequality (14) characterizes the weakening of linear coupling primarily at the level of overall magnitude. However, this weakening remains coarse-grained and is still influenced by the scale of the variables. To further examine whether the correlation decay exhibits a more structured form, we introduce the following whitening transformation:

$$R = C_{Z_k}^{-\frac{1}{2}} C_{Z_k G} C_G^{-\frac{1}{2}}, \quad (15)$$

where $R$ captures a scale-normalized pattern of linear correlation. Crucially, the singular values of $R$ possess a clear physical interpretation: they correspond to the canonical correlation coefficients between $Z_k$ and $G$, which quantify the strongest linear correlations along orthogonal canonical directions. In the presence of data leakage, a non-monotonic shift in which certain coefficients increase while others diminish would introduce ambiguity into the overall degree of association. Consequently, we assume that WLC is not merely a trivial magnitude reduction but instead exhibits a structural consistency. Specifically, we require that the spectral structure of $R$ satisfy a canonical **Singular Value Partial Order (SVPO)** condition:

$$\rho_i^{(l)} \le \rho_i^{(c)} \quad \forall i, \text{ and } \exists j : \rho_j^{(l)} < \rho_j^{(c)}, \quad (16)$$

where $\rho_i$ denotes the singular value of $R$. The SVPO condition enforces a coherent decay of the singular values across

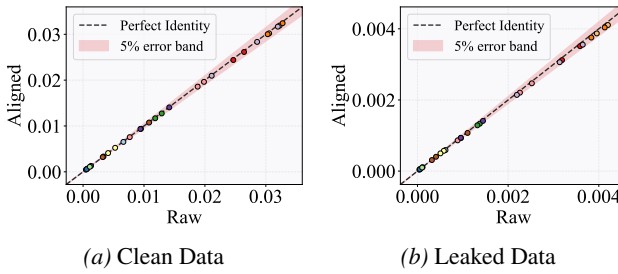

*(a)* Clean Data      *(b)* Leaked Data

*Figure 6.* Empirical validation of MISA on Qwen2.5-7B-Instruct. "Raw" and "Aligned" denote $I\left(\boldsymbol{A};\boldsymbol{G}\right)$ and $I\left(\boldsymbol{Z}_k;\boldsymbol{G}\right)$, respectively. Each colored point represents a layer's averaged results, closely aligning with the identity line.

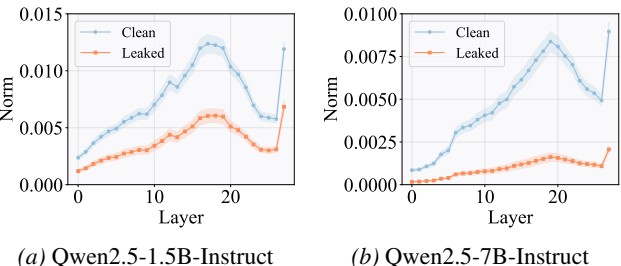

*(a)* Qwen2.5-1.5B-Instruct      *(b)* Qwen2.5-7B-Instruct

*Figure 7.* Comparison of cross-covariance norms between leaked and clean data. The norms on leaked data are significantly lower than those on clean data.

all canonical directions. This implies that the WLC within the computation subspace triggered by low-dimensional computation and leaked input is not merely a reduction in the cross-covariance magnitude, but a structured form of weakening. We argue that it is precisely this structured weakening that drives the mutual information decay. We formalize this via the following proposition:

**Proposition 3.1.** *For leaked data, the weakening of linear coupling (WLC) satisfies the Singular Value Partial Order (SVPO) condition on the singular value spectrum of $\boldsymbol{R} = \boldsymbol{C}_{\boldsymbol{Z}_k}^{-\frac{1}{2}} \boldsymbol{C}_{\boldsymbol{Z}_k \boldsymbol{G}} \boldsymbol{C}_{\boldsymbol{G}}^{-\frac{1}{2}}$. Then the Gaussian mutual information on leaked data is strictly smaller than that on clean data:*

$$I^{(l)}\left(\boldsymbol{Z}_k;\boldsymbol{G}\right) < I^{(c)}\left(\boldsymbol{Z}_k;\boldsymbol{G}\right). \tag{17}$$

The proof of Inequality (17) is deferred to Appendix B.3. The proposition elucidates that for leaked samples, the representation-gradient interaction within the computation subspace is suppressed in a canonical, direction-wise structured manner. This essentially creates a mechanistic bottleneck, inducing a structural decay of mutual information.

### 3.4.2. EXPERIMENTAL VALIDATION

According to the analysis in Section 3.4.1, we empirically evaluate three conditions: mutual information subspace alignment, cross-covariance shrinkage (Inequality (14)) and

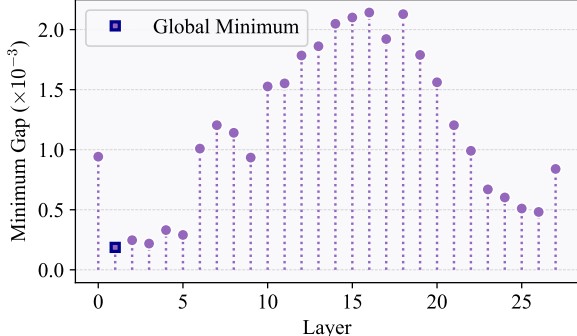

*Figure 8.* Minimum spectral gap between clean and leaked data in Qwen2.5-7B-Instruct. The square marker denotes the global minimum gap, which remains strictly positive, thereby confirming that the SVPO condition holds.

singular value partial order.

MISA is evaluated under the setting where the RTE is sufficiently small; in particular, we adopt a $1\%$ threshold and compute both the raw mutual information $I\left(\boldsymbol{A};\boldsymbol{G}\right)$ and the aligned mutual information $\boldsymbol{I}\left(\boldsymbol{Z}_k;\boldsymbol{G}\right)$ under this condition. The results are shown in Figure 6. We observe that the aligned and raw mutual information values exhibit near-perfect agreement, with layer-wise relative deviations mostly below $2\%$. This strongly supports MISA, indicating that the mutual information is predominantly concentrated within the active computation subspace.

The empirical results for cross-covariance shrinkage are shown in Figure 7. We observe that the Frobenius norm of the cross-covariance matrix is consistently smaller on leaked data than on clean data. This observation is highly consistent with our theoretical analysis, further indicating that the linear coupling between $\boldsymbol{Z}_k$ and $\boldsymbol{G}$ is substantially weakened on the leaked samples.

To evaluate the singular value partial order (SVPO) condition, we compute the matrix $\boldsymbol{R}$ and its singular values for both clean and leaked data, and then examine the element-wise differences $\rho_i^{(c)} - \rho_i^{(l)}$. In particular, we report the minimum difference $\min_i \left(\rho_i^{(c)} - \rho_i^{(l)}\right)$ for each layer. The results are shown in Figure 8. We observe that this minimum difference is non-negative across all layers. Although the global minimum occurs at layer 1, even in this case the singular values corresponding to leaked data do not exceed those of clean data. This observation indicates that, under leaked inputs, transforming the cross-covariance into the whitened space reveals a global contraction of the singular value spectrum. Such a structured decay suggests that the WLC is not merely a magnitude reduction, but instead reflects a non-trivial geometry property of the representation-gradient interaction, thereby supporting the SVPO condition.

In summary, the experiments presented above are consis-

*Table 2.* Results of reasoning restoration. Abbreviations in the table are: RESTOR. (Restoration), CONT. (contaminated model), LNE-B. (LNE-Blocking), and AVG. (average consistency).

| MODELS | STRATEGY | RESTOR. (↑) | RETAIN (↑) | AVG. (↑) |
|---|---|---|---|---|
| QWEN-1.5B | CONT. | 0.00 | 100.00 | 50.00 |
| | SAMPLING | 23.08 | 89.02 | 56.05 |
| | TED | 27.47 | 86.83 | 57.15 |
| | LNE-B. | 39.56 | 89.96 | 64.76 |
| | TSS (OURS) | **51.65** | **94.37** | **73.01** |
| QWEN-7B | CONT. | 0.00 | 100.00 | 50.00 |
| | SAMPLING | 9.09 | 91.57 | 50.33 |
| | TED | 22.73 | 89.33 | 56.03 |
| | LNE-B. | 31.82 | 88.76 | 60.29 |
| | TSS (OURS) | **45.45** | **97.19** | **71.32** |

tent with our theoretical analyses in Section 3.4.1. Low-dimensional computation confines the linear coupling between representations and gradients to a restricted subspace. Under leaked inputs, gradient shrinkage further weakens this coupling, which manifests as a structured decay of the singular value spectrum in the whitened space. Together with Proposition 3.1, this mechanism leads to a reduction in mutual information on leaked data.

## 4. Application: Reasoning Restoration

Based on the findings, we investigate whether intervening in the computational subspace of the contaminated model can restore its original reasoning performance. Since contaminated models rely on a specialized low-dimensional channel defined by $U_k$, we propose Top-Subspace Suppression (TSS). This intervention subtracts the activation components lying in the $U_k$ subspace, thereby preventing over-reliance on this specialized pathway and encouraging the model to leverage a broader representation space. Formally:

$$a^{\text{new}} = \mu + \left( I - \gamma^\alpha U_k U_k^\top \right) (a - \mu),$$
$$\gamma = \frac{\left\| U_k U_k^\top (a - \mu) \right\|}{\| a - \mu \|}, \quad \alpha \in [0, 1]. \tag{18}$$

Here, $\gamma$ is a dynamically adjusted intervention strength. To evaluate the reasoning recovery effect, we construct two evaluation sets. From the leaked data, we select samples for which the base model produces incorrect predictions while the contaminated model produces correct ones, forming the **Restoration Set**. From the clean data, we select samples on which both models agree, forming the **Retain Set**. Our objective is to maximally recover the reasoning behavior of the contaminated model on the Restoration Set, while minimizing consistency degradation on the Retain Set. Concretely, we report the consistency as the proportion of samples for which the intervened contaminated model aligns with the base model on both sets. For comparison, we adopt the TED (Dong et al., 2024) and LNE-Blocking (Hou et al., 2025) frameworks as baselines (details in Appendix C.6).

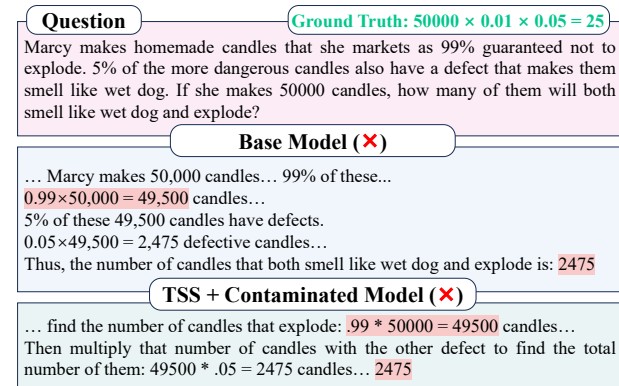

*Figure 9.* A successful restoration case on the Restoration Set. Initially, the contaminated model generates the correct answer while the base model exhibits an incorrect one. Following our intervention, the contaminated model reverts to the erroneous reasoning trajectory of the base model.

Table 2 presents the restoration results. Our intervention strategy recovers almost half of the contaminated samples while sacrificing no more than 6% of retain performance, achieving the highest average score on both metrics. These results collectively demonstrate the considerable practical value of our analysis in reasoning restoration.

To qualitatively evaluate our intervention, we present a representative case from the Restoration Set in Figure 9. Notably, while the original contaminated model produces the correct result, the base model yields an incorrect answer of 2475. After our intervention, the contaminated model reverts to the exact erroneous trajectory of the base model. This case study provides a concrete demonstration that our mechanistic analysis enables targeted recovery of the reasoning behavior in contaminated models, and holds considerable potential for broader application.

## 5. Related Work

Reasoning capabilities in large language models (LLMs) have advanced significantly. Models such as OpenAI's o1 (Jaech et al., 2024) and DeepSeek-R1 (Guo et al., 2025) have achieved performance surpassing human levels on various challenging benchmarks. However, the reasoning processes of LLMs are susceptible to data contamination. In severe cases, contaminated models may even reproduce evaluation questions verbatim on widely used benchmarks such as MATH500 (Hendrycks et al., 2021; Lightman et al., 2024), casting doubt on whether their outputs reflect genuine reasoning or mere memorization (Wu et al., 2026). To address this issue, prior research has established a diverse body of work, ranging from contamination detection (Karamolegkou et al., 2023; Mattern et al., 2023; Li, 2023; Deng et al., 2024; Ranaldi et al., 2024; Tu et al., 2024; Liu et al., 2024; Xu et al., 2024; Shi et al., 2024; Golchin & Surdeanu, 2024; Zhang et al., 2025a; Fu et al., 2025; Xu et al., 2025; Wu

et al., 2026; Hou et al., 2025) to the construction of leakage-free benchmarks (Jain et al., 2025; White et al., 2025; Chen et al., 2025; Wu et al., 2026), as well as strategies to enhance the reliability of evaluation performance (Dong et al., 2024; Hou et al., 2025). From a broader perspective, data contamination is closely tied to the way models acquire and use knowledge (Cao et al., 2024), naturally connecting to the question of how training data shapes model behavior. This line of work has been studied through data attribution methods, including influence functions (Koh & Liang, 2017) and their extensions (Pruthi et al., 2020; Kwon et al., 2024). These works typically trace model predictions or losses back to specific training examples. However, the underlying mechanism of contaminated reasoning remains poorly understood, hindering the separation of generalization from memorization and limiting principled mitigation. This paper diverges from prior works by providing a mechanistic investigation to shed light on how contaminated reasoning manifests internally.

## 6. Conclusion

In this paper, we present a comprehensive mechanistic investigation of reasoning behaviors under data contamination. We first identify a mutual information decay phenomenon to characterize contaminated reasoning. To interpret this observation, we analyze leakage-induced mechanism shifts, revealing a narrowing geometry characterized by eigenspectrum concentration in representations, a low-dimensional computation regime, and a structured decay of singular values in the whitened space, which together provide a coherent explanation for the observed mutual information decay, verified by experimental evidence. Finally, we demonstrate the practical utility of our analysis through a reasoning restoration study. Overall, our work moves beyond output-level observations and underscores the critical importance of a mechanistic perspective in both understanding and mitigating leakage-induced LLM reasoning. An important direction for future work is to extend our framework beyond the mathematical domain in this study, including other forms of reasoning and more diverse task distributions.

## Acknowledgements

This work was supported by the National Natural Science Foundation of China (No.U24A20335, No.62406321), the independent research project of the Key Laboratory of Cognition and Decision Intelligence for Complex Systems, and CIPS-SMP-Zhipu Large Model Fund.

## Impact Statement

This work investigates the underlying mechanisms of contaminated reasoning in large language models (LLMs). Our goal is to enhance the reliability, transparency, and evaluation rigor of LLMs, especially in scenarios where data contamination or memorization may distort apparent reasoning ability. Our findings may help the research community better interpret model performance and design more robust evaluation frameworks, thereby mitigating the risks of unintentional data leakage and ensuring the integrity of model assessment. We believe that understanding these mechanisms is a necessary step toward developing more reliable and trustworthy language models.

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

# A. Data Synthesis

We construct a synthetic arithmetic reasoning dataset MSCalc with exactly controlled chain-of-thought length, following a procedure inspired by RandomCalculation (Wu et al., 2026), but with stronger guarantees on correctness, controllability, and interpretability.

We define a reasoning step as one atomic arithmetic operation. Specifically, the following operations are counted as one step each: (1) Binary operations: addition ($+$), subtraction ($-$), multiplication ($\times$), and division ($\div$); (2) Unary power operations applied only to numeric leaves: squaring ($n^2$) and cubing ($n^3$).

To avoid ambiguity and parsing issues, power operations are restricted to numeric constants only and are never applied to composite expressions. As a result, expressions such as $(a + b)^2$ or chained powers like $n^{3^2}$ never occur.

Given a target number of reasoning steps $K$, we generate an arithmetic expression whose evaluation requires exactly $K$ atomic steps. This is achieved via a recursive construction:

1. Base case ($K = 0$): Generate a single integer constant, which requires no reasoning step.

2. Unary case ($K = 1$): Optionally generate a numeric power expression $n^2$ or $n^3$, consuming exactly one step.

3. Recursive binary case ($K \geq 1$):
   - Select a binary operator from $\{+, -, \times, \div\}$, cosuming one step.
   - Randomly split the remaining $K - 1$ steps between the left and right sub-expressions.
   - Recursively generate both sub-expressions with the assigned step budgets.

This construction guarantees that the total number of atomic operations is exactly $K$. The final expression is rendered using standard mathematical notation. All intermediate and final values are computed using exact rational arithmetic via fraction representations, rather than floating-point numbers. This ensures: (1) No numerical approximation errors; (2) Exact simplification of fractions; (3) Deterministic and verifiable results. During generation, division-by-zero is explicitly avoided. After constructing an expression, its final value is verified.

Along with each expression, we programmatically record the full sequence of intermediate reasoning steps. Each step corresponds to exactly one atomic operation and is stored as a symbolic equality. The following invariants are strictly enforced: (1) The number of recorded steps equals the target step count $K$; (2) The value produced at each step matches the exact symbolic computation; (3) The value in the final step exactly equals the value of the full expression. Any sample violating these conditions is discarded and regenerated. To control difficulty and numerical scale, we optionally enforce a constraint on the final result, requiring it to lie within a predefined range (e.g., $[-10000, 10000]$). Samples falling outside this range are rejected.

The final dataset thus comprises triplets of (problem, answer, step-by-step derivation), where both the answer and the trace are guaranteed to be mathematically correct by construction, without relying on any language model. A sample of the synthetic dataset is shown in Figure 10.

---

**Problem:** Evaluate this LaTeX numerical expression:
`$(66^2 + (197 - (\frac{373248}{64})))$`

**Answer:** -1279

**Steps:**
    Step 1: `66^2 = 4356`
    Step 2: `\frac{373248}{64} = 5832`
    Step 3: `197 - 5832 = -5635`
    Step 4: `4356 + (-5635) = -1279`

---

*Figure 10.* An example of the synthetic dataset.

# B. Theoretical Proof

## B.1. Proof of Relative Truncation Error Equation

*Proof.* Let $\tilde{A} \in \mathbb{R}^{N \times d}$ denote the centered activation matrix of a module, and let its covariance admit the eigendecomposition

$$C_A = \mathbb{E}\left[\tilde{A}^\top \tilde{A}\right] = U \Lambda U^\top, \tag{19}$$

where $U = [u_1, \ldots, u_d]$ is an orthonormal basis of eigenvectors and $\Lambda = \mathrm{diag}(\lambda_1, \ldots, \lambda_d)$ with $\lambda_1 \geq \cdots \lambda_d \geq 0$. We write $U_k = [u_1, \ldots, u_k]$ for the top-$k$ eigenvectors and $U_\perp = [u_{k+1}, \ldots, u_d]$ for the remaining ones, so that $U = [U_k, U_\perp]$. The centered output of the linear module is

$$\tilde{Y} = \tilde{A} W, \tag{20}$$

and its low-rank approximation induced by the principal subspace is

$$\tilde{Y}_k = \tilde{A}_k W = \tilde{A} U_k U_k^\top W. \tag{21}$$

The residual between the full output and its truncated version can be written as

$$\tilde{Y} - \tilde{Y}_k = \tilde{A}\left(I - U_k U_k^\top\right) W = \tilde{A} U_\perp U_\perp^\top W. \tag{22}$$

We now compute the squared Frobenius norm of this residual:

$$\left\|\tilde{Y} - \tilde{Y}_k\right\|_F^2 = \mathrm{tr}\left[\left(\tilde{Y} - \tilde{Y}_k\right)^\top \left(\tilde{Y} - \tilde{Y}_k\right)\right] = \mathrm{tr}\left[W^\top U_\perp U_\perp^\top \tilde{A}^\top \tilde{A} U_\perp U_\perp^\top W\right]. \tag{23}$$

Up to a constant factor (which cancels out in the relative error), we can replace $\tilde{A}^\top \tilde{A}$ by the covariance $C_A$, and use the eigendecomposition above, we obtain

$$U_\perp^\top C_A U_\perp = U_\perp^\top U \Lambda U^\top U_\perp = \mathrm{diag}(\lambda_{k+1}, \ldots, \lambda_d). \tag{24}$$

Moreover,

$$U_\perp U_\perp^\top C_A U_\perp U_\perp^\top = U_\perp \mathrm{diag}(\lambda_{k+1}, \ldots, \lambda_d) U_\perp^\top. \tag{25}$$

Substituting Equation (25) into Equation (23) yields

$$\left\|\tilde{Y} - \tilde{Y}_k\right\|_F^2 = \mathrm{tr}\left[W^\top U_\perp \mathrm{diag}(\lambda_{k+1}, \ldots, \lambda_d) U_\perp^\top W\right]. \tag{26}$$

Using the cyclic property of the trace,

$$\mathrm{tr}\left[W^\top U_\perp \mathrm{diag}(\lambda_{k+1}, \ldots, \lambda_d) U_\perp^\top W\right] = \sum_{i=k+1}^{d} \lambda_i \left\|W^\top u_i\right\|^2, \tag{27}$$

so we obtain

$$\left\|\tilde{Y} - \tilde{Y}_k\right\|_F^2 = \sum_{i=k+1}^{d} \lambda_i \left\|W^\top u_i\right\|^2. \tag{28}$$

The squared norm of the full output can be written analogously as

$$\left\|\tilde{Y}\right\|_F^2 = \mathrm{tr}\left[\tilde{Y}^\top \tilde{Y}\right] = \mathrm{tr}\left[W^\top C_A W\right] = \mathrm{tr}\left[W^\top U \Lambda U^\top W\right] = \sum_{j=1}^{d} \lambda_j \left\|W^\top u_j\right\|^2. \tag{29}$$

Combining the above results and assuming $\left\|\tilde{Y}\right\|_F^2 > 0$, the relative truncation error is

$$\frac{\left\|\tilde{Y} - \tilde{Y}_k\right\|_F^2}{\left\|\tilde{Y}\right\|_F^2} = \frac{\sum_{i=k+1}^{d} \lambda_i \left\|W^\top u_i\right\|^2}{\sum_{j=1}^{d} \lambda_j \left\|W^\top u_j\right\|^2}, \tag{30}$$

which matches the expression used in the main text. $\qquad \square$

### B.2. Proof of Mutual Information Subspace Alignment

*Proof.* To establish the Mutual Information Subspace Alignment (MISA), we begin by decomposing the activation into two components, corresponding to the principal computation subspace and its complement:

$$\boldsymbol{A} = \boldsymbol{A}_k + \boldsymbol{A}_\perp, \ \ \boldsymbol{A}_k = \boldsymbol{\mu} + \tilde{\boldsymbol{A}}_k. \tag{31}$$

By the chain rule of mutual information, we obtain

$$I\left(\boldsymbol{A}; \boldsymbol{G}\right) - I\left(\boldsymbol{A}_k; \boldsymbol{G}\right) = I\left(\boldsymbol{A}_\perp; \boldsymbol{G}|\boldsymbol{A}_k\right). \tag{32}$$

Therefore, proving $I\left(\boldsymbol{A}; \boldsymbol{G}\right) \approx I\left(\boldsymbol{A}_k; \boldsymbol{G}\right)$ is equivalent to showing that:

$$I\left(\boldsymbol{A}_\perp; \boldsymbol{G}|\boldsymbol{A}_k\right) \to 0. \tag{33}$$

Under the low-dimensional computation mechanism, the centered module output $\tilde{\boldsymbol{Y}}$ can be well-approximated by its projection onto the active computation subspace. Formally, there exists $\epsilon_k > 0$ such that

$$||\tilde{\boldsymbol{Y}} - \tilde{\boldsymbol{Y}}_k||_F \le \epsilon_k ||\tilde{\boldsymbol{Y}}||_F. \tag{34}$$

This implies that the forward computation is executed almost entirely within the principal subspace. Let $g\left(\cdot\right)$ denote the gradient mapping with respect to the module output, and denote $\boldsymbol{G} = g\left(\boldsymbol{Y}\right)$. Setting $\boldsymbol{Y}_k = \tilde{\boldsymbol{Y}}_k + \boldsymbol{\mu}_{\boldsymbol{Y}}$, we define

$$\boldsymbol{D} = \boldsymbol{G} - g\left(\boldsymbol{Y}_k\right) \tag{35}$$
$$\boldsymbol{m} = \mathbb{E}\left[\boldsymbol{D}|\boldsymbol{A}_k\right] \tag{36}$$
$$\boldsymbol{R} = \boldsymbol{D} - \boldsymbol{m}. \tag{37}$$

The conditional cross-covariance between $\boldsymbol{A}_\perp$ and $\boldsymbol{G}$ given $\boldsymbol{A}_k$ is

$$\boldsymbol{C}_{\boldsymbol{A}_\perp \boldsymbol{G}|\boldsymbol{A}_k} = \mathbb{E}\left[\bar{\boldsymbol{A}}_\perp^\top \boldsymbol{R}|\boldsymbol{A}_k\right], \ \ \bar{\boldsymbol{A}}_\perp = \boldsymbol{A}_\perp - \mathbb{E}\left[\boldsymbol{A}_\perp|\boldsymbol{A}_k\right]. \tag{38}$$

Applying Jensen's inequality and the conditional Cauchy-Schwarz inequality yields:

$$\left\|\mathbb{E}\left[\bar{\boldsymbol{A}}_\perp^\top \boldsymbol{R}|\boldsymbol{A}_k\right]\right\|_F \le \mathbb{E}\left[\|\bar{\boldsymbol{A}}_\perp\|_F \|\boldsymbol{R}\|_F |\boldsymbol{A}_k\right] \tag{39}$$

$$\mathbb{E}\left[\|\bar{\boldsymbol{A}}_\perp\|_F \|\boldsymbol{R}\|_F |\boldsymbol{A}_k\right] \le \sqrt{\mathbb{E}\left[\|\bar{\boldsymbol{A}}_\perp\|_F^2 |\boldsymbol{A}_k\right]}\sqrt{\mathbb{E}\left[\|\boldsymbol{R}\|_F^2 |\boldsymbol{A}_k\right]}. \tag{40}$$

$$\left\|\boldsymbol{C}_{\boldsymbol{A}_\perp \boldsymbol{G}|\boldsymbol{A}_k}\right\|_F \le \sqrt{\mathbb{E}\left[\|\bar{\boldsymbol{A}}_\perp\|_F^2 |\boldsymbol{A}_k\right]}\sqrt{\mathbb{E}\left[\|\boldsymbol{R}\|_F^2 |\boldsymbol{A}_k\right]}. \tag{41}$$

$$\mathbb{E}\left[\|\boldsymbol{R}\|_F^2 |\boldsymbol{A}_k\right] \le \mathbb{E}\left[\|\boldsymbol{D}\|_F^2 |\boldsymbol{A}_k\right] \tag{42}$$

Assume that the gradient mapping $g\left(\cdot\right)$ is $L$-Lipschitz continuous. Then the residual term is bounded by the forward approximation error, yielding

$$\begin{aligned}
\mathbb{E}\left[\|\boldsymbol{R}\|_F^2 |\boldsymbol{A}_k\right] &\le \mathbb{E}\left[\|\boldsymbol{D}\|_F^2 |\boldsymbol{A}_k\right] = \mathbb{E}\left[\|\boldsymbol{G} - g\left(\boldsymbol{Y}_k\right)\|_F^2 |\boldsymbol{A}_k\right] \\
&\le L^2 \mathbb{E}\left[\|\boldsymbol{Y} - \boldsymbol{Y}_k\|_F^2 |\boldsymbol{A}_k\right] = L^2 \mathbb{E}\left[\left\|\tilde{\boldsymbol{Y}} - \tilde{\boldsymbol{Y}}_k\right\|_F^2 |\boldsymbol{A}_k\right].
\end{aligned} \tag{43}$$

Using Inequality (34) and assuming that $\mathbb{E}\left[\left\|\tilde{\boldsymbol{Y}}\right\|_F^2 |\boldsymbol{A}_k\right]$ is uniformly bounded, we obtain a uniform upper bound on the residual variance:

$$\mathbb{E}\left[\left\|\tilde{\boldsymbol{Y}} - \tilde{\boldsymbol{Y}}_k\right\|_F^2 |\boldsymbol{A}_k\right] \le \epsilon_k^2 \mathbb{E}\left[\left\|\tilde{\boldsymbol{Y}}\right\|_F^2 |\boldsymbol{A}_k\right] \le \epsilon_k^2 C_0^2. \tag{44}$$

$$\sqrt{\mathbb{E}\left[\|\boldsymbol{R}\|_F^2 |\boldsymbol{A}_k\right]} \le \sqrt{\mathbb{E}\left[\|\boldsymbol{D}\|_F^2 |\boldsymbol{A}_k\right]} \le L\epsilon_k C_0. \tag{45}$$

Combining these results with Inequality (41) and assuming that $\mathbb{E}\left[\|\bar{\boldsymbol{A}}_\perp\|_F^2 | \boldsymbol{A}_k\right]$ is uniformly bounded, we obtain the key cross-covariance bound

$$\left\|\boldsymbol{C}_{\boldsymbol{A}_\perp \boldsymbol{G} | \boldsymbol{A}_k}\right\|_F \leq C_1 \epsilon_k, \tag{46}$$

which shows that the statistical coupling between $\boldsymbol{A}_\perp$ and $\boldsymbol{G}$ shrinks proportionally to the relative truncation error of the forward computation. Under the conditional jointly Gaussian assumption and assuming that the conditional covariance matrices $\boldsymbol{C}_{\boldsymbol{A}_\perp | \boldsymbol{A}_k}$ and $\boldsymbol{C}_{\boldsymbol{G} | \boldsymbol{A}_k}$ are uniformly well-conditioned, the conditional mutual information admits the canonical-correlation form

$$I\left(\boldsymbol{A}_\perp; \boldsymbol{G} | \boldsymbol{A}_k\right) = -\frac{1}{2} \log \det \left(\boldsymbol{I} - \boldsymbol{K}\boldsymbol{K}^\top\right), \tag{47}$$

where

$$\boldsymbol{K} = \boldsymbol{C}_{\boldsymbol{A}_\perp | \boldsymbol{A}_k}^{-1/2} \boldsymbol{C}_{\boldsymbol{A}_\perp \boldsymbol{G} | \boldsymbol{A}_k} \boldsymbol{C}_{\boldsymbol{G} | \boldsymbol{A}_k}^{-1/2}. \tag{48}$$

Since the eigenvalues of $\boldsymbol{C}_{\boldsymbol{A}_\perp | \boldsymbol{A}_k}$ and $\boldsymbol{C}_{\boldsymbol{G} | \boldsymbol{A}_k}$ are uniformly bounded away from zero and infinity, applying the inequality $\|\boldsymbol{P}\boldsymbol{Q}\boldsymbol{S}\|_F \leq \|\boldsymbol{P}\|_2 \|\boldsymbol{Q}\|_F \|\boldsymbol{S}\|_2$ gives:

$$\|\boldsymbol{K}\|_F \leq \gamma \left\|\boldsymbol{C}_{\boldsymbol{A}_\perp \boldsymbol{G} | \boldsymbol{A}_k}\right\|_F. \tag{49}$$

Substituting the covariance bound from Inequality (46) yields

$$\|\boldsymbol{K}\|_F \leq C_2 \epsilon_k. \tag{50}$$

Since $\|\boldsymbol{K}\|_2 \leq \|\boldsymbol{K}\|_F$, for sufficiently small $\epsilon_k$ we have:

$$\|\boldsymbol{K}\|_2 \leq C_2 \epsilon_k < 1. \tag{51}$$

Hence, $\|\boldsymbol{K}\|_2$ remains bounded away from 1. Consequently, $1 - \|\boldsymbol{K}\|_2^2$ is bounded away from zero, which implies:

$$\frac{1}{1 - \|\boldsymbol{K}\|_2^2} \leq C_3. \tag{52}$$

Using a matrix inequality

$$-\log \det \left(\boldsymbol{I} - \boldsymbol{M}\right) \leq \frac{1}{1 - \|\boldsymbol{M}\|_2} \operatorname{tr}\left(\boldsymbol{M}\right), \tag{53}$$

we obtain the upper bound

$$I\left(\boldsymbol{A}_\perp; \boldsymbol{G} | \boldsymbol{A}_k\right) \leq \frac{1}{2\left(1 - \|\boldsymbol{K}\|_2^2\right)} \|\boldsymbol{K}\|_F^2 \leq \frac{C_3}{2} \|\boldsymbol{K}\|_F^2 \leq C_4 \epsilon_k^2. \tag{54}$$

Therefore, as the truncation error vanishes,

$$I\left(\boldsymbol{A}_\perp; \boldsymbol{G} | \boldsymbol{A}_k\right) \to 0, \tag{55}$$

and thus under low-dimensional computation,

$$I\left(\boldsymbol{A}; \boldsymbol{G}\right) \approx I\left(\boldsymbol{A}_k; \boldsymbol{G}\right). \tag{56}$$

From the low-dimensional representation, the projected coefficients satisfy

$$\boldsymbol{Z}_k = \tilde{\boldsymbol{A}}\boldsymbol{U}_k, \quad \boldsymbol{A}_k = \boldsymbol{\mu} + \boldsymbol{Z}_k \boldsymbol{U}_k^\top. \tag{57}$$

This mapping is a bijection induced by the orthonormal basis $\boldsymbol{U}_k$, up to the mean shift. Since Gaussian mutual information is invariant under such transformations, we obtain

$$I\left(\boldsymbol{A}_k; \boldsymbol{G}\right) = I\left(\boldsymbol{Z}_k; \boldsymbol{G}\right). \tag{58}$$

Combining the above results, the mutual information subspace alignment holds:

$$I\left(\boldsymbol{A}; \boldsymbol{G}\right) \approx I\left(\boldsymbol{A}_k; \boldsymbol{G}\right) = I\left(\boldsymbol{Z}_k; \boldsymbol{G}\right). \tag{59}$$

That is, the mutual information is effectively captured by the principal computation subspace. $\qquad \square$

### B.3. Proof of Mutual Information Inequality

*Proof.* Under the multivariate Gaussian distribution for the latent representations, we assume that the involved covariance matrices are nonsingular. The mutual information $I(\boldsymbol{Z}_k; \boldsymbol{G})$ can then be expressed as:

$$I\left(\boldsymbol{Z}_k; \boldsymbol{G}\right) = \frac{1}{2} \log \frac{\det\left(\boldsymbol{C}_{\boldsymbol{Z}_k}\right) \det\left(\boldsymbol{C}_{\boldsymbol{G}}\right)}{\det\left(\boldsymbol{C}\right)}, \tag{60}$$

where

$$\boldsymbol{C} = \begin{pmatrix} \boldsymbol{C}_{\boldsymbol{Z}_k} & \boldsymbol{C}_{\boldsymbol{Z}_k\boldsymbol{G}} \\ \boldsymbol{C}_{\boldsymbol{G}\boldsymbol{Z}_k} & \boldsymbol{C}_{\boldsymbol{G}} \end{pmatrix}. \tag{61}$$

Using the Schur complement property for the determinant of a block matrix, we have:

$$\det\left(\boldsymbol{C}\right) = \det\left(\boldsymbol{C}_{\boldsymbol{G}}\right) \det\left(\boldsymbol{C}_{\boldsymbol{Z}_k} - \boldsymbol{C}_{\boldsymbol{Z}_k\boldsymbol{G}} \boldsymbol{C}_{\boldsymbol{G}}^{-1} \boldsymbol{C}_{\boldsymbol{G}\boldsymbol{Z}_k}\right). \tag{62}$$

Substituting this into Equation (60):

$$I\left(\boldsymbol{Z}_k; \boldsymbol{G}\right) = \frac{1}{2} \log \frac{\det\left(\boldsymbol{C}_{\boldsymbol{Z}_k}\right)}{\det\left(\boldsymbol{C}_{\boldsymbol{Z}_k} - \boldsymbol{C}_{\boldsymbol{Z}_k\boldsymbol{G}} \boldsymbol{C}_{\boldsymbol{G}}^{-1} \boldsymbol{C}_{\boldsymbol{G}\boldsymbol{Z}_k}\right)} = -\frac{1}{2} \log \det\left(\boldsymbol{I} - \boldsymbol{C}_{\boldsymbol{Z}_k}^{-1} \boldsymbol{C}_{\boldsymbol{Z}_k\boldsymbol{G}} \boldsymbol{C}_{\boldsymbol{G}}^{-1} \boldsymbol{C}_{\boldsymbol{G}\boldsymbol{Z}_k}\right). \tag{63}$$

Based on the definition in Section 3.4.1

$$\boldsymbol{R} = \boldsymbol{C}_{\boldsymbol{Z}_k}^{-\frac{1}{2}} \boldsymbol{C}_{\boldsymbol{Z}_k\boldsymbol{G}} \boldsymbol{C}_{\boldsymbol{G}}^{-\frac{1}{2}} = \boldsymbol{U}\boldsymbol{S}\boldsymbol{V}^{\top}, \tag{64}$$

where $\boldsymbol{S} = \operatorname{diag}\left(\rho_1, \rho_2, \ldots, \rho_r\right)$ are the singular values. In this context, these values correspond to the canonical correlation coefficients. Consequently, they are naturally bounded within the interval $[0, 1]$. Note that the matrix $\boldsymbol{M} = \boldsymbol{C}_{\boldsymbol{Z}_k}^{-1} \boldsymbol{C}_{\boldsymbol{Z}_k\boldsymbol{G}} \boldsymbol{C}_{\boldsymbol{G}}^{-1} \boldsymbol{C}_{\boldsymbol{G}\boldsymbol{Z}_k}$ is similar to $\boldsymbol{R}\boldsymbol{R}^{\top}$:

$$\boldsymbol{C}_{\boldsymbol{Z}_k}^{\frac{1}{2}} \boldsymbol{M} \boldsymbol{C}_{\boldsymbol{Z}_k}^{-\frac{1}{2}} = \boldsymbol{C}_{\boldsymbol{Z}_k}^{-\frac{1}{2}} \boldsymbol{C}_{\boldsymbol{Z}_k\boldsymbol{G}} \boldsymbol{C}_{\boldsymbol{G}}^{-1} \boldsymbol{C}_{\boldsymbol{G}\boldsymbol{Z}_k} \boldsymbol{C}_{\boldsymbol{Z}_k}^{-\frac{1}{2}} = \boldsymbol{R}\boldsymbol{R}^{\top}. \tag{65}$$

Since similar matrices share the same eigenvalues, the eigenvalules of $\boldsymbol{M}$ are the eigenvalues of $\boldsymbol{R}\boldsymbol{R}^{\top}$, which are the squared singular values $\rho_i^2$. Consequently, the eigenvalues of $\boldsymbol{I} - \boldsymbol{M}$ are $1 - \rho_i^2$. Recalling that the determinant of a matrix is the product of its eigenvalues, the mutual information can be rewritten as:

$$I\left(\boldsymbol{Z}_k; \boldsymbol{G}\right) = -\frac{1}{2} \log \prod_{i=1}^{r} \left(1 - \rho_i^2\right) = -\frac{1}{2} \sum_{i=1}^{r} \log \left(1 - \rho_i^2\right), \tag{66}$$

where $r = \min\left(k, d_{\boldsymbol{G}}\right)$. This result shows that the mutual information is entirely captured by the singular values of $\boldsymbol{R}$.

Let

$$f\left(\rho\right) = -\frac{1}{2} \log \left(1 - \rho^2\right). \tag{67}$$

The function is strictly increasing for $\rho_i$. The SVPO condition in Section 3.4.1 implies $f\left(\rho_i^{(l)}\right) \le f\left(\rho_i^{(c)}\right)$ for all $i$, with a strict inequality holding for at least one index $j$. This yields a strict inequality in the sum, giving

$$I^{(l)}\left(\boldsymbol{Z}_k; \boldsymbol{G}\right) < I^{(c)}\left(\boldsymbol{Z}_k; \boldsymbol{G}\right). \tag{68}$$

Hence, the mutual information of leaked data is strictly lower than that of clean data. $\qquad\square$

# C. Experimental Results

## C.1. Mutual Information Decay

Figures 11 to 33 present the mutual information decay results across additional settings. We consistently observe that on contaminated models, mutual information is lower for leaked data compared to clean data; in contrast, for base models, the mutual information remains indistinguishable between the two data types.

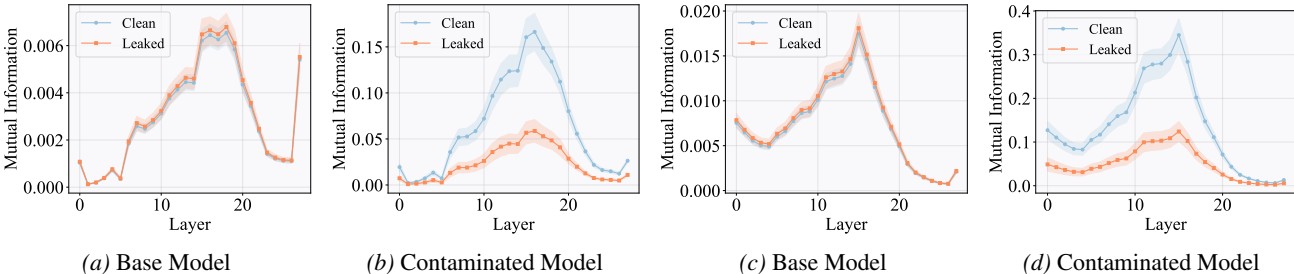

*(a)* Base Model    *(b)* Contaminated Model    *(c)* Base Model    *(d)* Contaminated Model

*Figure 11.* Mutual information of Qwen2.5-1.5B-Instruct on GSM8K. `mlp.gate_proj`: (11a,11b); `mlp.down_proj`: (11c, 11d).

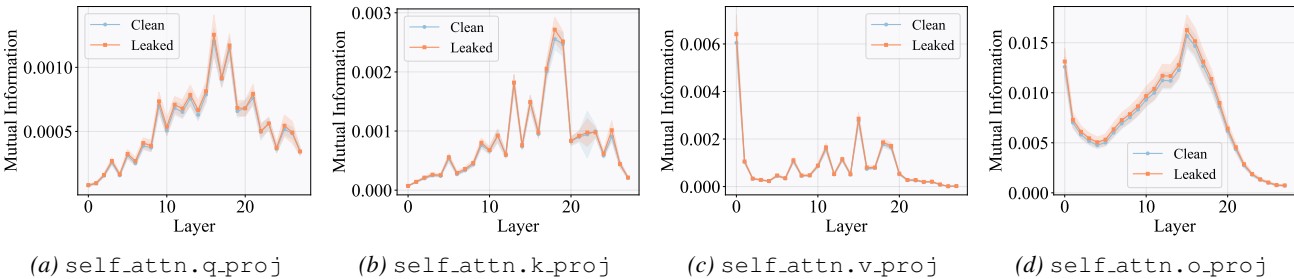

*(a)* `self_attn.q_proj`    *(b)* `self_attn.k_proj`    *(c)* `self_attn.v_proj`    *(d)* `self_attn.o_proj`

*Figure 12.* Mutual information of original Qwen2.5-1.5B-Instruct on GSM8K (Attention modules).

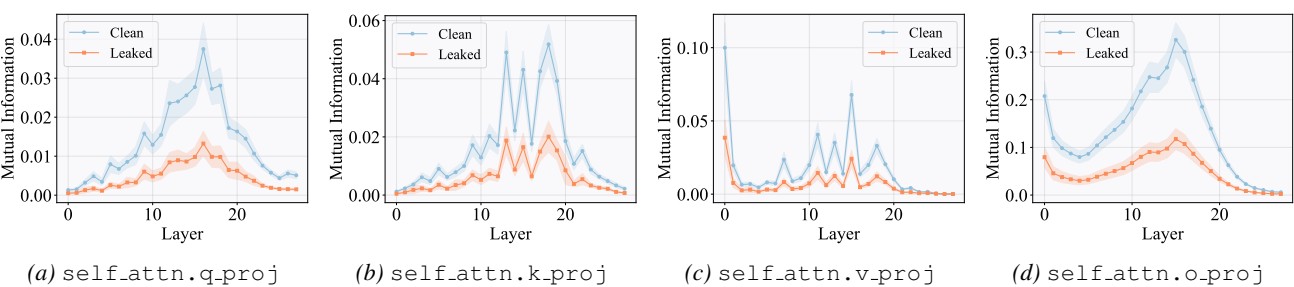

*(a)* `self_attn.q_proj`    *(b)* `self_attn.k_proj`    *(c)* `self_attn.v_proj`    *(d)* `self_attn.o_proj`

*Figure 13.* Mutual information of contaminated Qwen2.5-1.5B-Instruct on GSM8K (Attention modules).

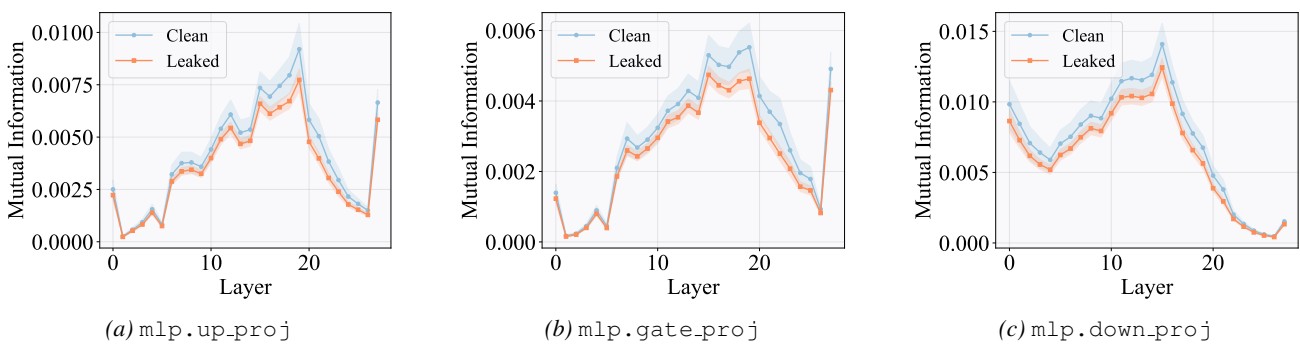

*Figure 14.* Mutual information of original Qwen2.5-1.5B-Instruct on MSCalc (MLP modules).

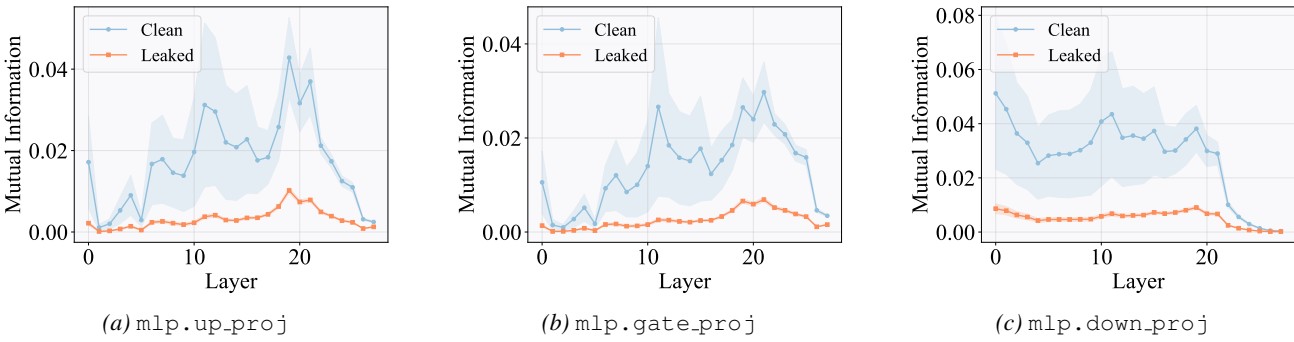

*Figure 15.* Mutual information of contaminated Qwen2.5-1.5B-Instruct on MSCalc (MLP modules).

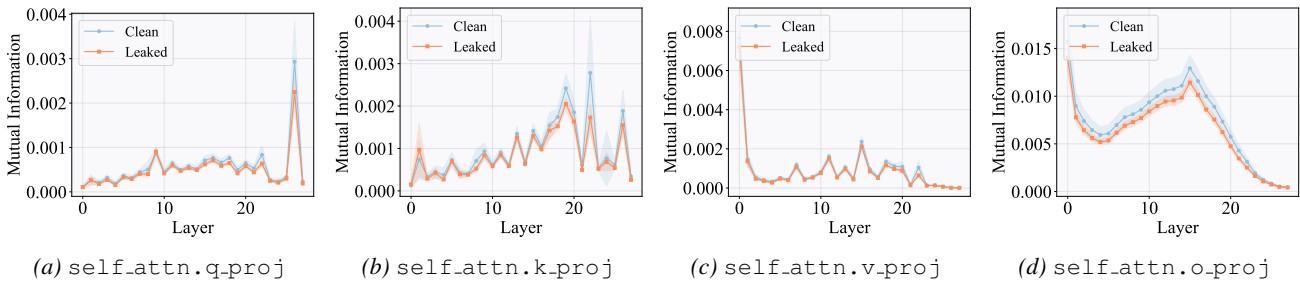

*Figure 16.* Mutual information of original Qwen2.5-1.5B-Instruct on MSCalc (Attention modules).

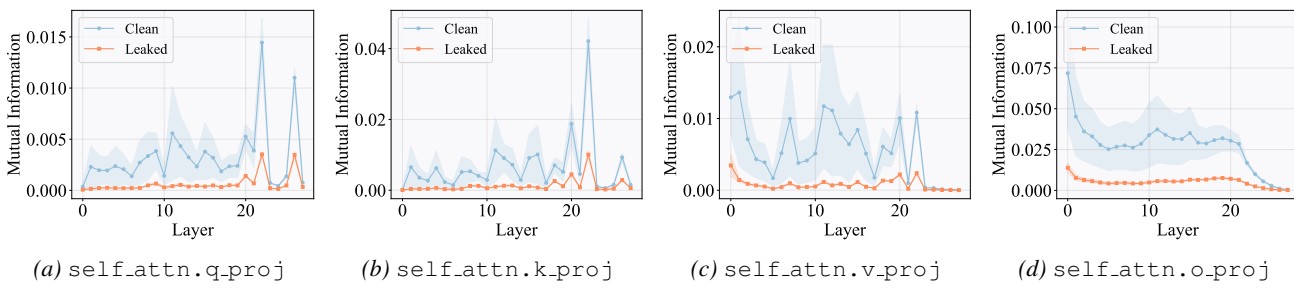

*Figure 17.* Mutual information of contaminated Qwen2.5-1.5B-Instruct on MSCalc (Attention modules).

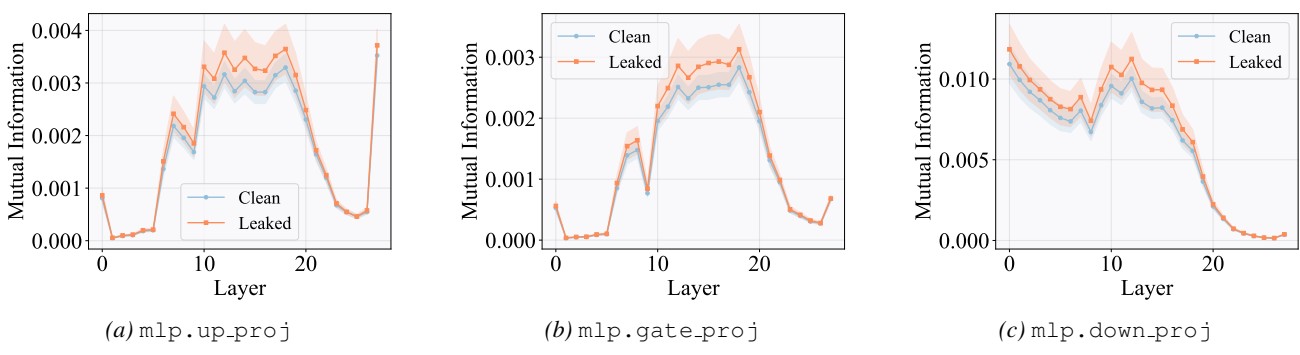

*Figure 18.* Mutual information of original Qwen2.5-7B-Instruct on GSM8K (MLP modules).

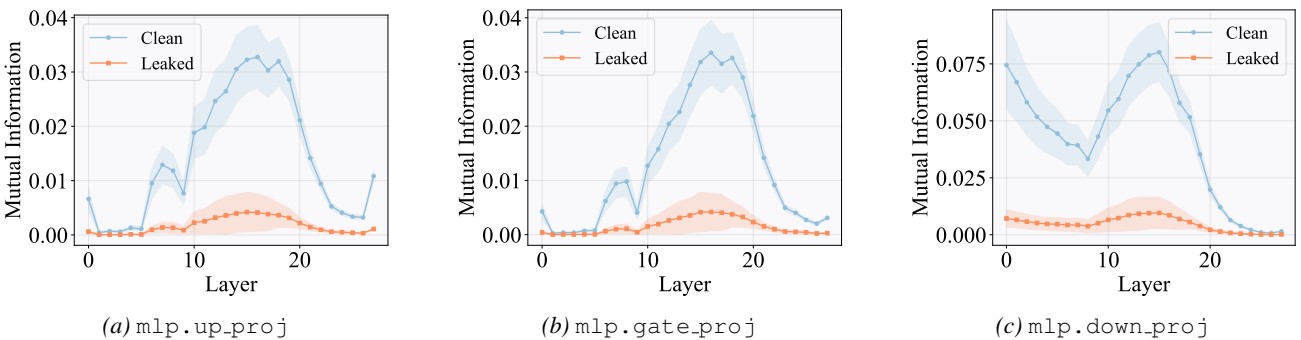

*Figure 19.* Mutual information of contaminated Qwen2.5-7B-Instruct on GSM8K (MLP modules).

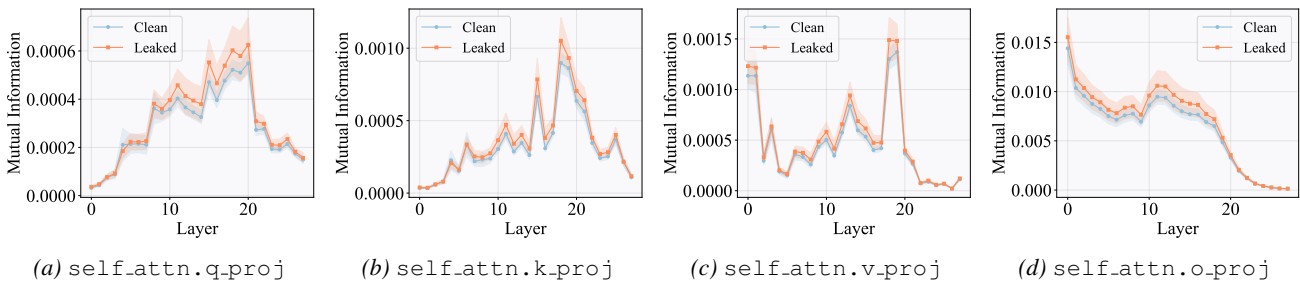

*Figure 20.* Mutual information of original Qwen2.5-7B-Instruct on GSM8K (Attention modules).

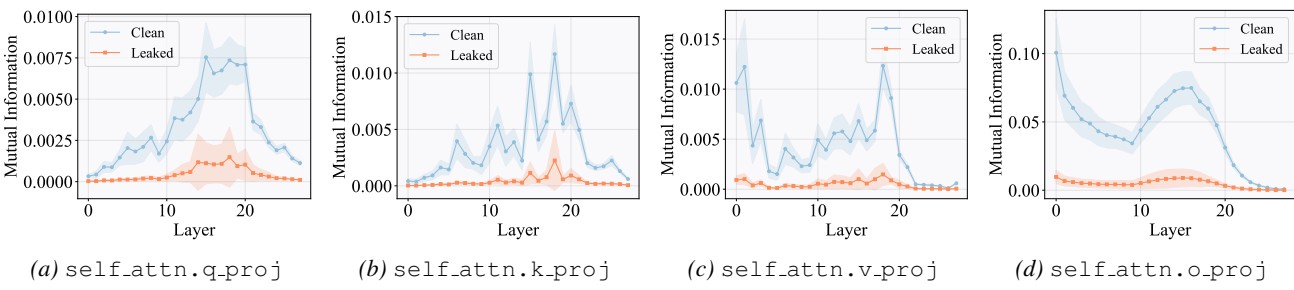

*Figure 21.* Mutual information of contaminated Qwen2.5-7B-Instruct on GSM8K (Attention modules).

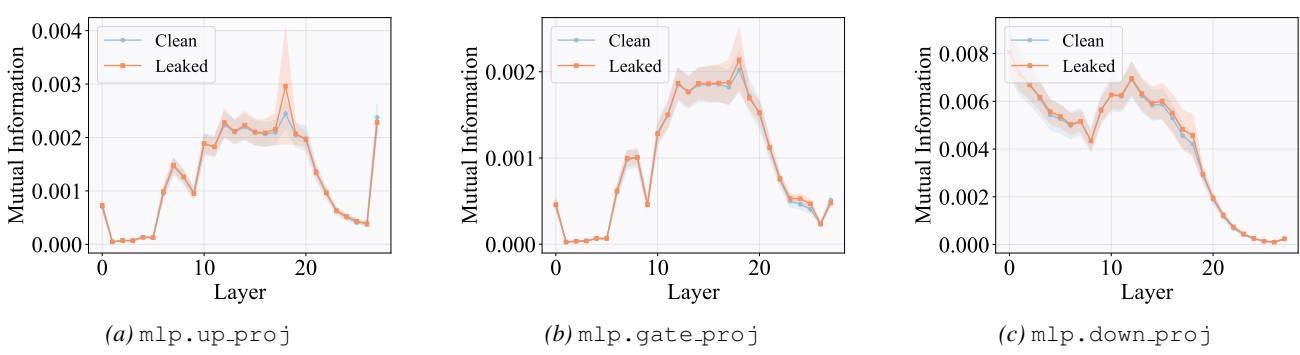

*Figure 22.* Mutual information of original Qwen2.5-7B-Instruct on MSCalc (MLP modules).

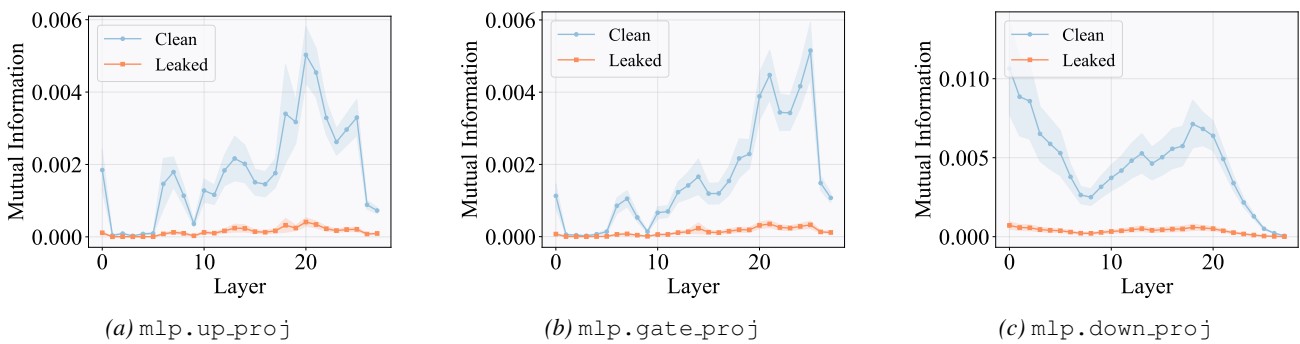

*Figure 23.* Mutual information of contaminated Qwen2.5-7B-Instruct on MSCalc (MLP modules).

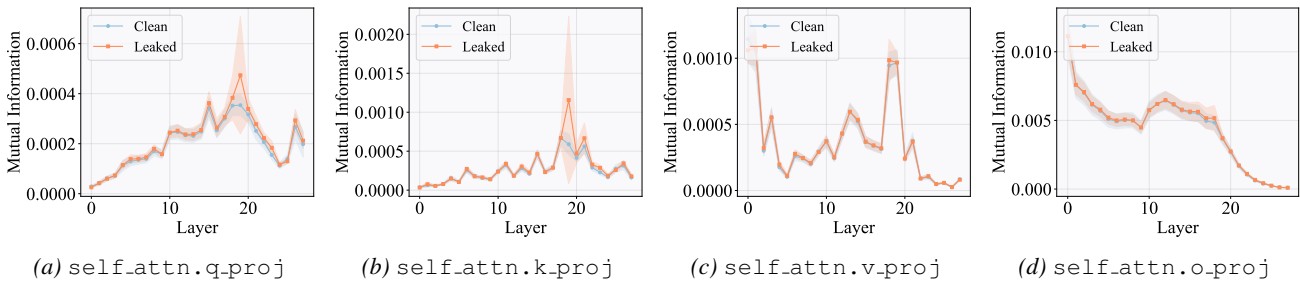

*Figure 24.* Mutual information of original Qwen2.5-7B-Instruct on MSCalc (Attention modules).

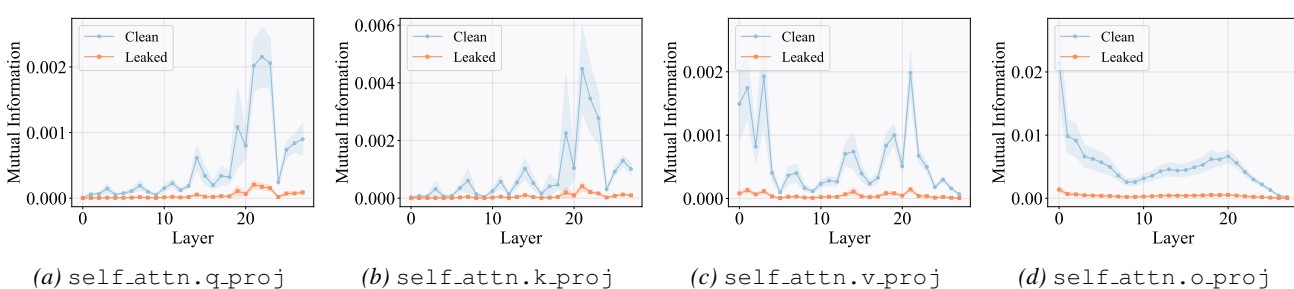

*Figure 25.* Mutual information of contaminated Qwen2.5-7B-Instruct on MSCalc (Attention modules).

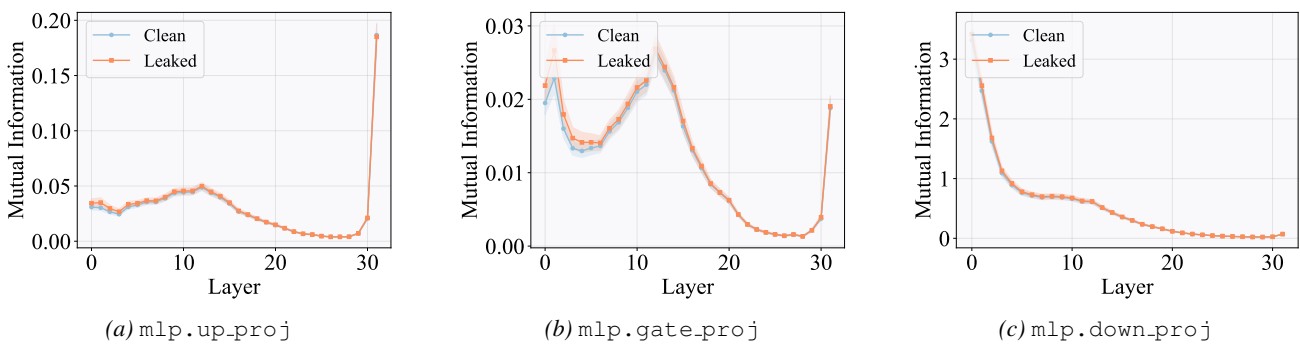

*Figure 26.* Mutual information of original Llama3.1-8B-Instruct on GSM8K (MLP modules).

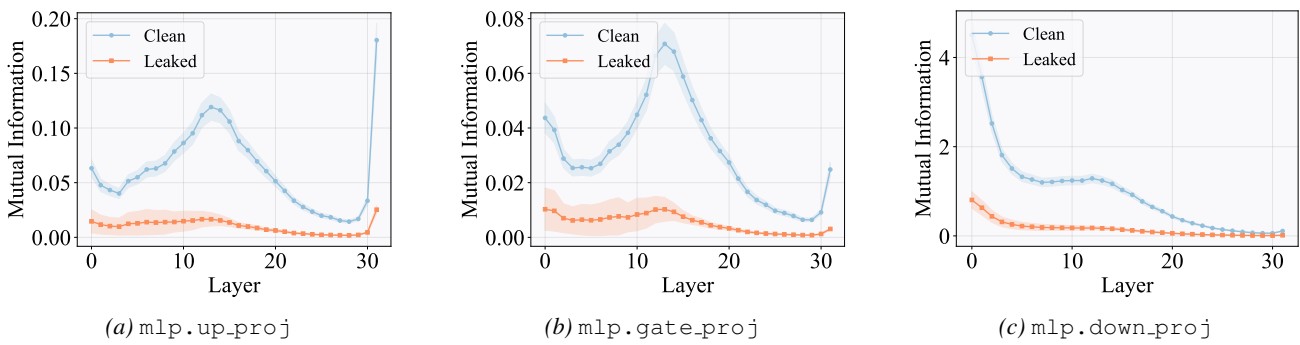

*Figure 27.* Mutual information of contaminated Llama3.1-8B-Instruct on GSM8K (MLP modules).

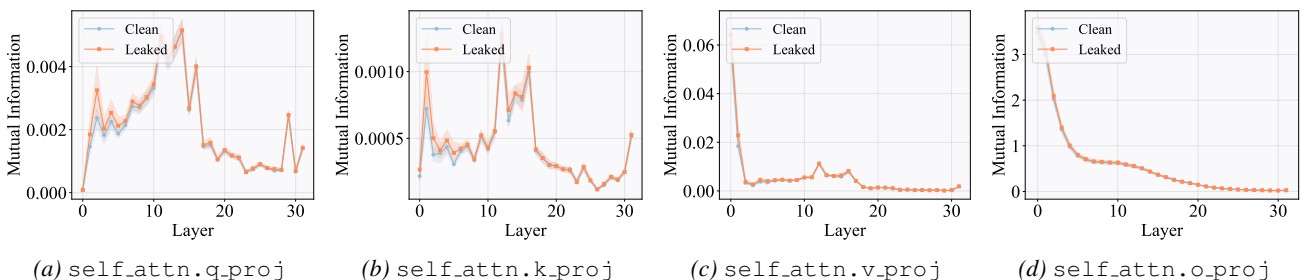

*Figure 28.* Mutual information of original Llama3.1-8B-Instruct on GSM8K (Attention modules).

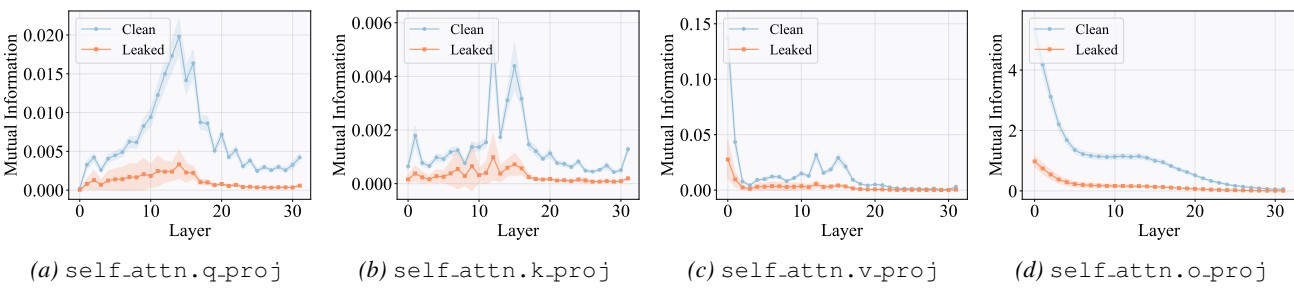

*Figure 29.* Mutual information of contaminated Llama3.1-8B-Instruct on GSM8K (Attention modules).

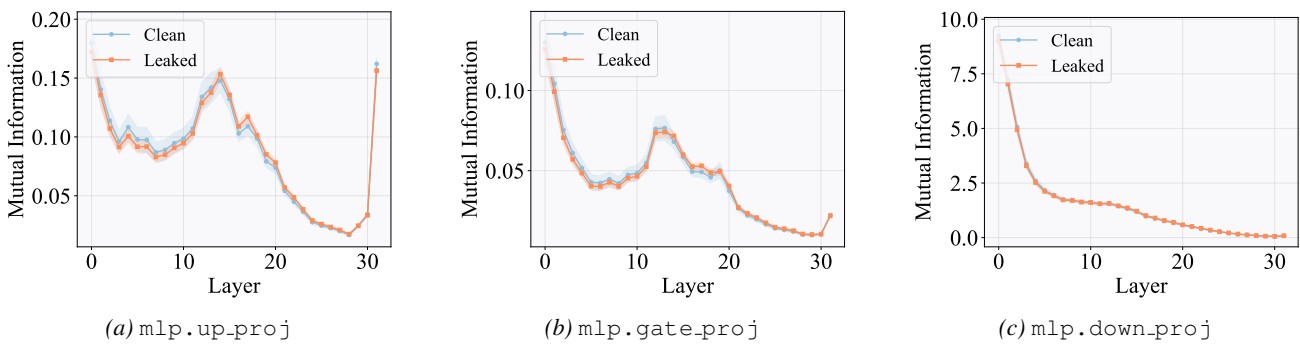

*Figure 30.* Mutual information of original Llama3.1-8B-Instruct on MSCalc (MLP modules).

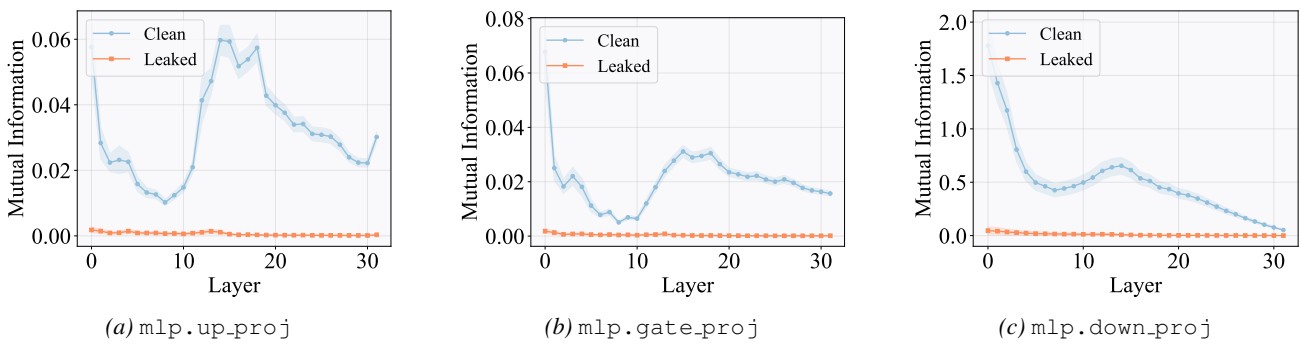

*Figure 31.* Mutual information of contaminated Llama3.1-8B-Instruct on MSCalc (MLP modules).

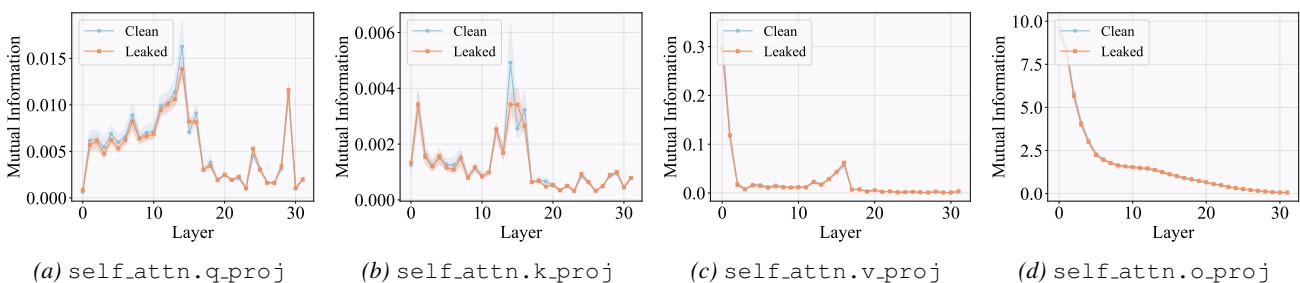

*Figure 32.* Mutual information of original Llama3.1-8B-Instruct on MSCalc (Attention modules).

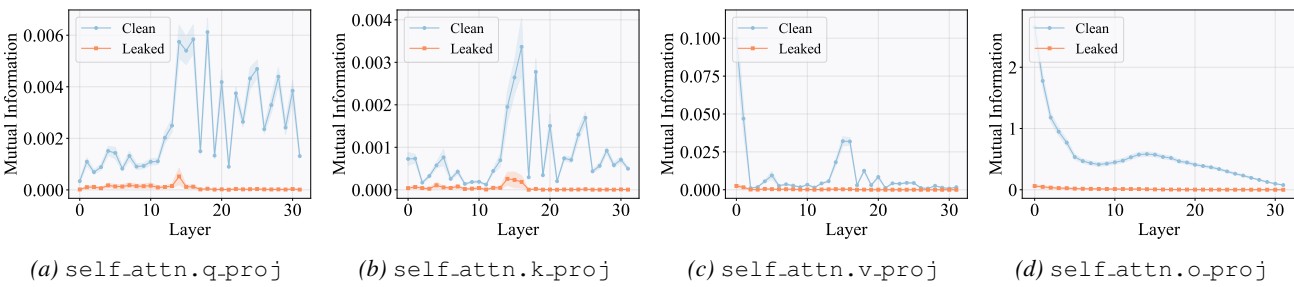

*Figure 33.* Mutual information of contaminated Llama3.1-8B-Instruct on MSCalc (Attention modules).

## C.2. Eigenspectrum Concentration

Figures 34 to 42 show the eigenspectrum concentration across the remaining configurations. Since multiple weight matrices within the same layer share identical inputs, we present results for a representative set of projections, specifically: `mlp.up_proj`, `mlp.down_proj`, `self_attn.q_proj`, and `self_attn.o_proj`.

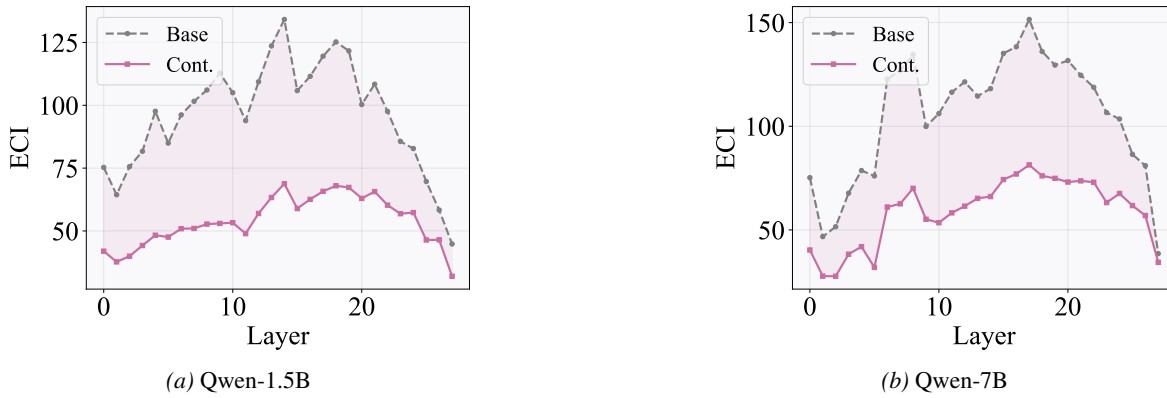

*(a)* Qwen-1.5B         *(b)* Qwen-7B

*Figure 34.* The ECI results for base models and contaminated models (`mlp.down_proj`).

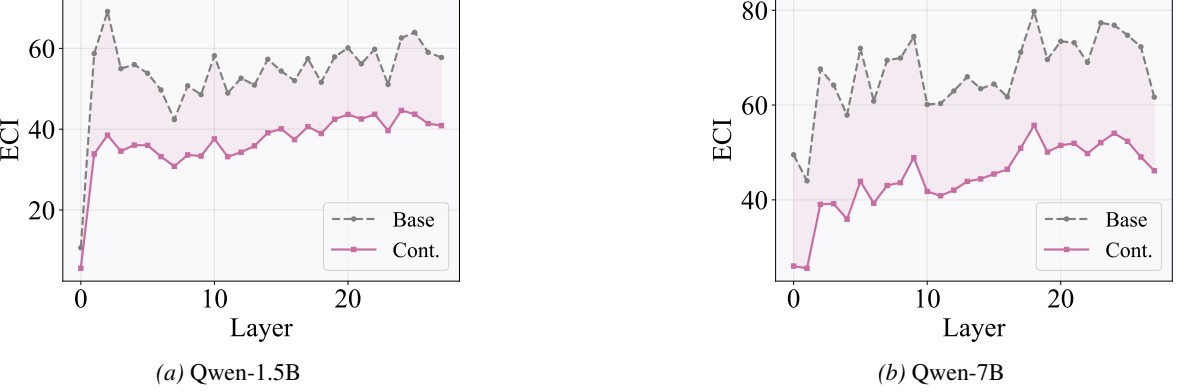

*(a)* Qwen-1.5B         *(b)* Qwen-7B

*Figure 35.* The ECI results for base models and contaminated models (`self_attn.q_proj`).

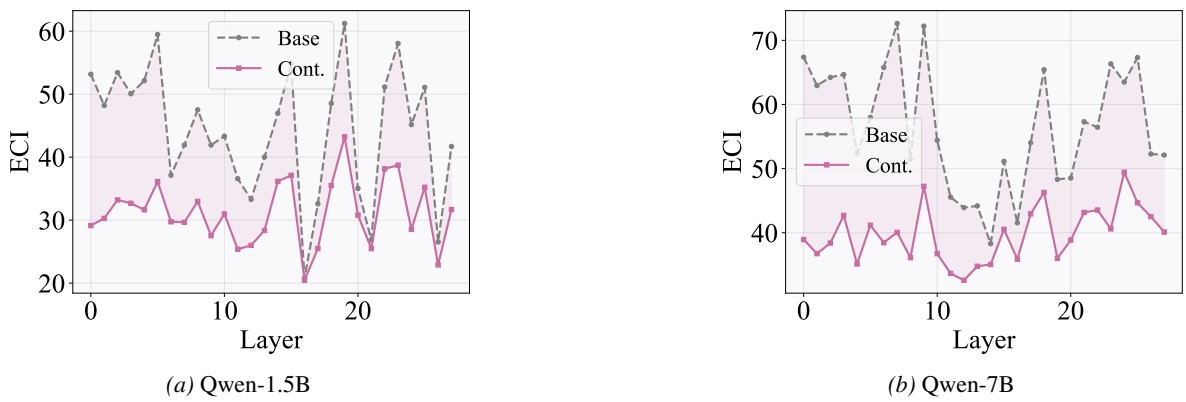

*(a)* Qwen-1.5B         *(b)* Qwen-7B

*Figure 36.* The ECI results for base models and contaminated models (`self_attn.o_proj`).

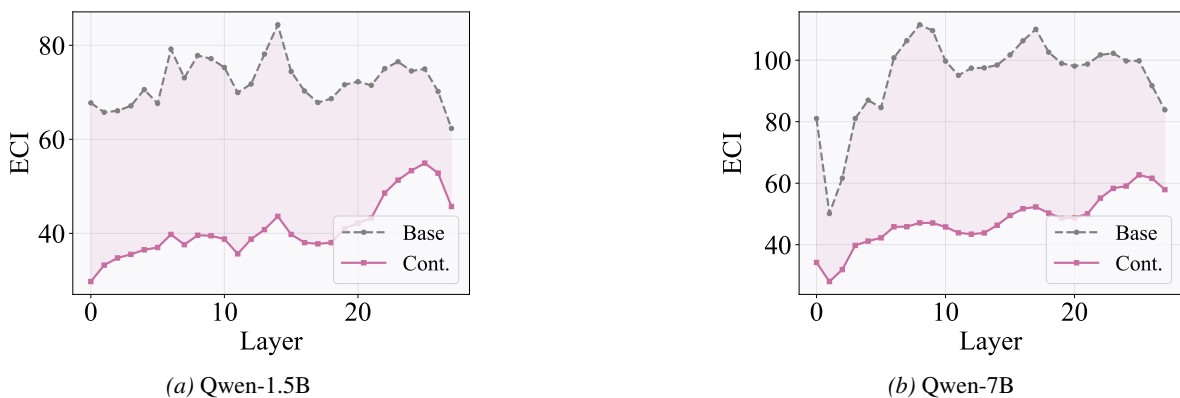

*Figure 37.* The ECI results for base models and contaminated models on MSCalc (`mlp.up_proj`).

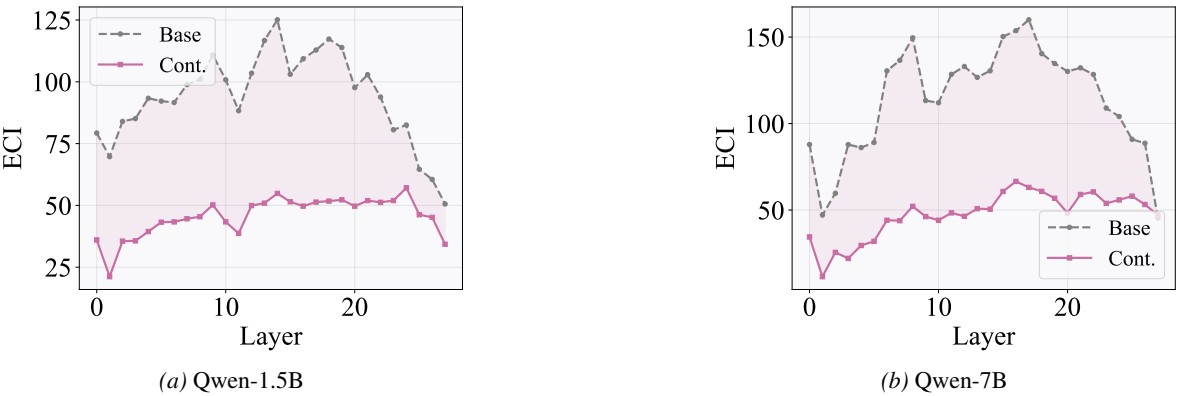

*Figure 38.* The ECI results for base models and contaminated models on MSCalc (`mlp.down_proj`).

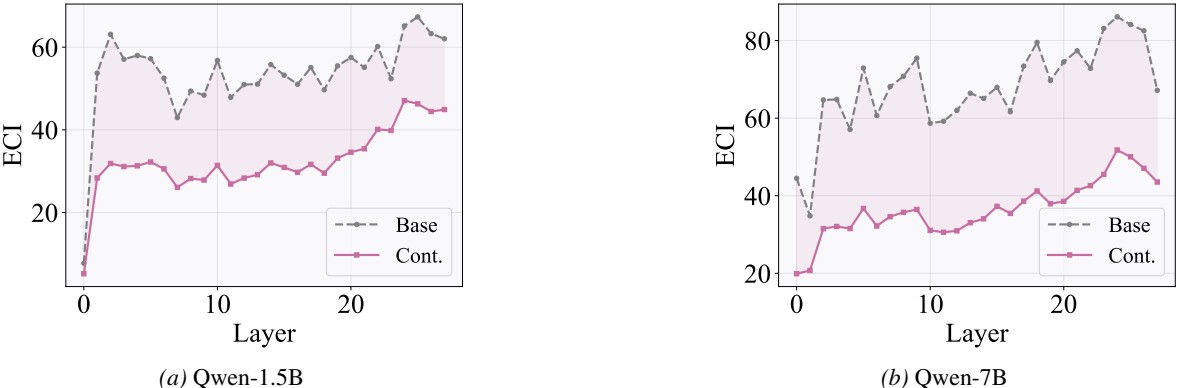

*Figure 39.* The ECI results for base models and contaminated models on MSCalc (`self_attn.q_proj`).

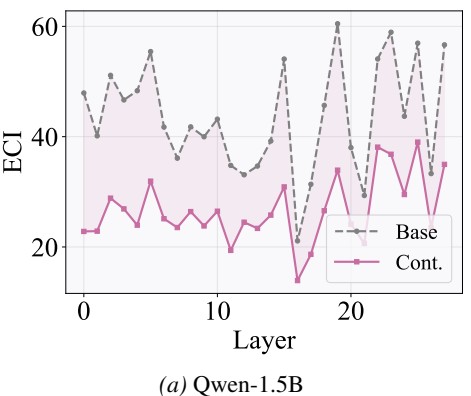

*(a)* Qwen-1.5B

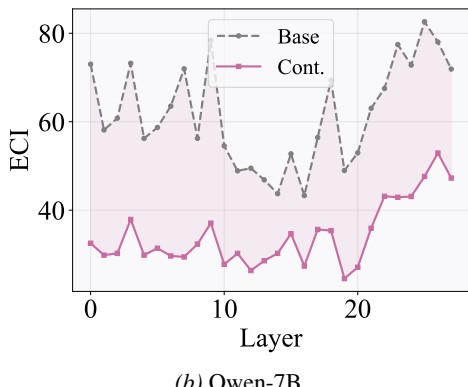

*(b)* Qwen-7B

*Figure 40.* The ECI results for base models and contaminated models on MSCalc (`self_attn.o_proj`).

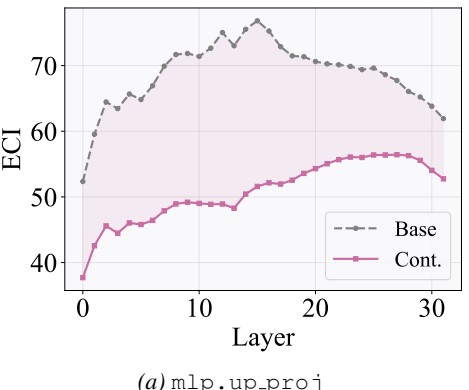

*(a)* `mlp.up_proj`

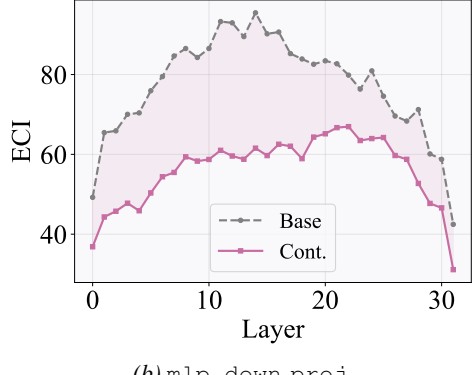

*(b)* `mlp.down_proj`

*Figure 41.* The ECI results for Llama3.1-8B-Instruct (`mlp`).

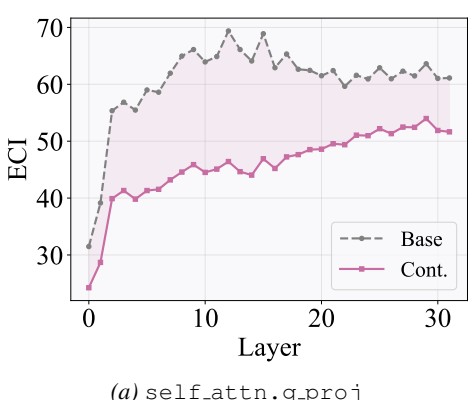

*(a)* `self_attn.q_proj`

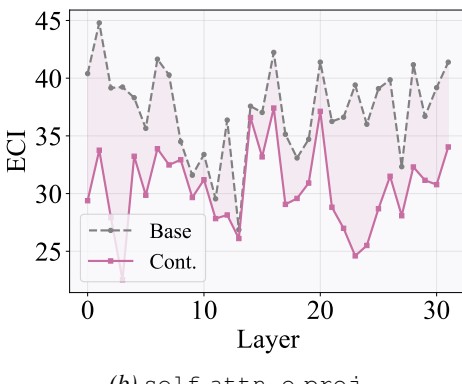

*(b)* `self_attn.o_proj`

*Figure 42.* The ECI results for Llama3.1-8B-Instruct (`self_attn`).

## C.3. Low-dimensional Computation

Figures 43 to 50 present the RTE for additional settings. The contaminated model exhibits a lower RTE than the base model.

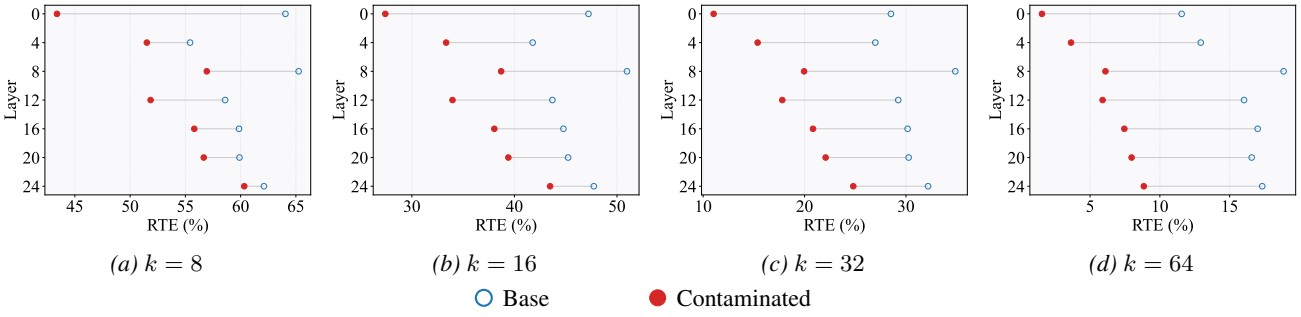

*Figure 43.* RTE of Qwen2.5-7B-Instruct (`mlp.gate_proj`).

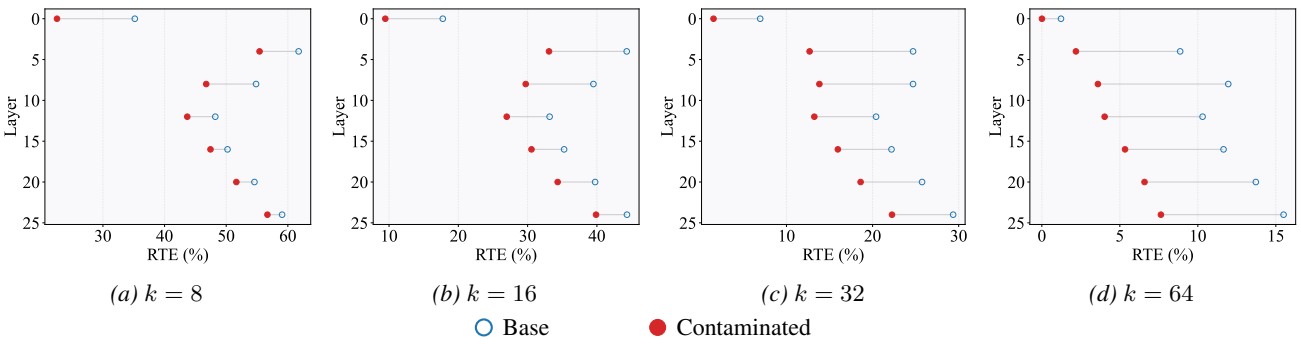

*Figure 44.* RTE of Qwen2.5-7B-Instruct (`self_attn.q_proj`).

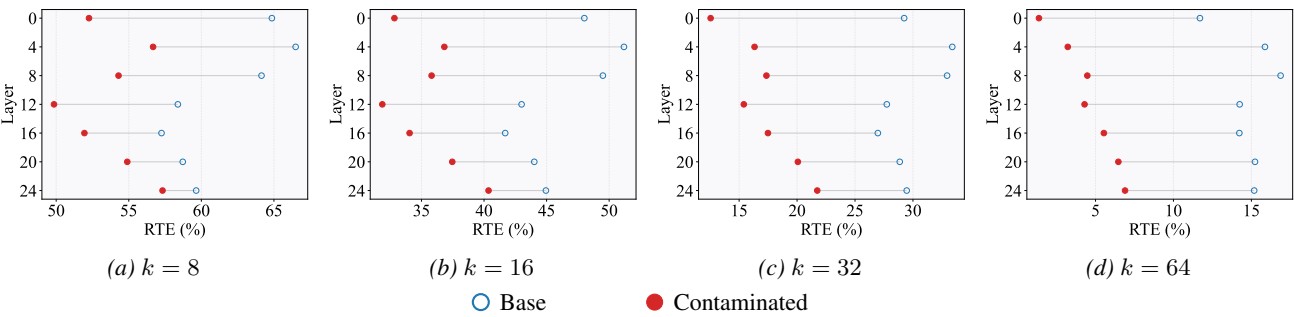

*Figure 45.* RTE of Qwen2.5-1.5B-Instruct (`mlp.up_proj`).

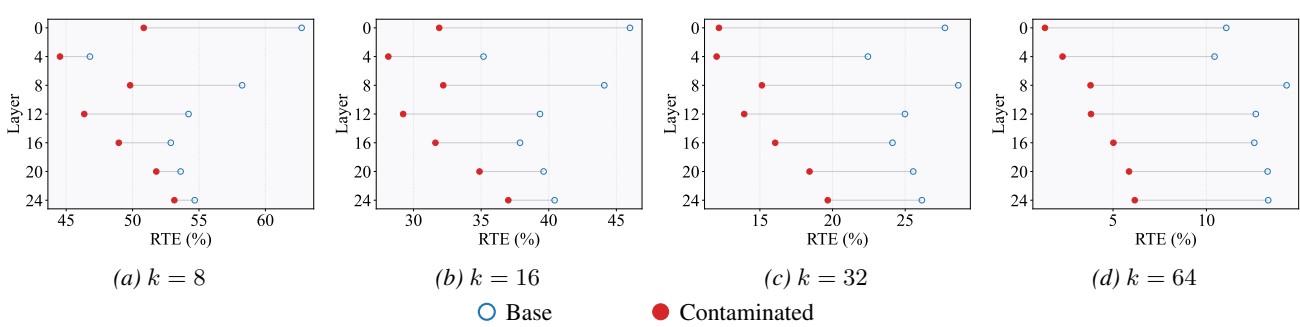

*Figure 46.* RTE of Qwen2.5-1.5B-Instruct (`mlp.gate_proj`).

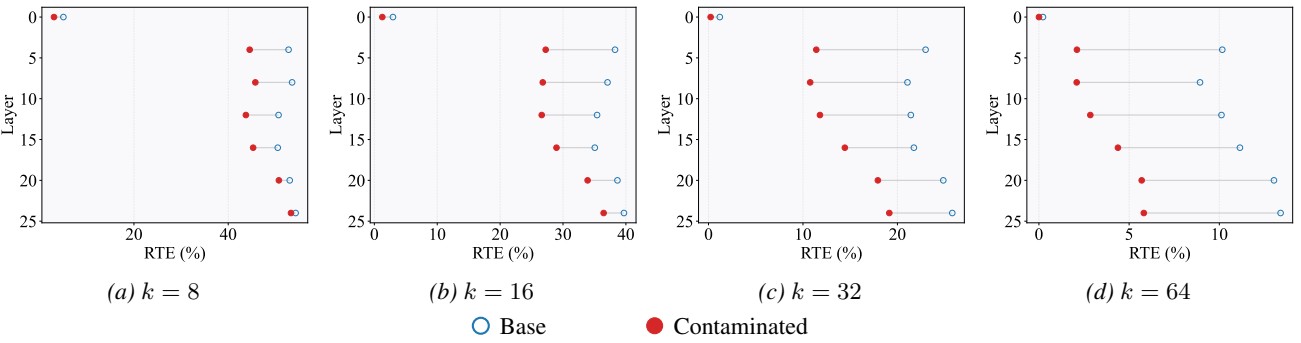

*Figure 47.* RTE of Qwen2.5-1.5B-Instruct (self_attn.q_proj).

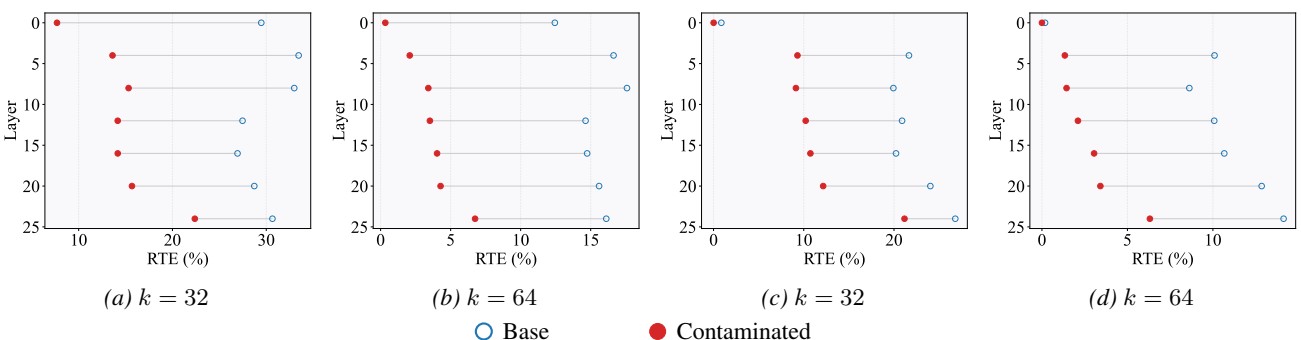

*Figure 48.* RTE analysis for Qwen2.5-1.5B-Instruct on MSCalc. mlp.up_proj: (48a, 48b); self_attn.q_proj: (48c, 48d).

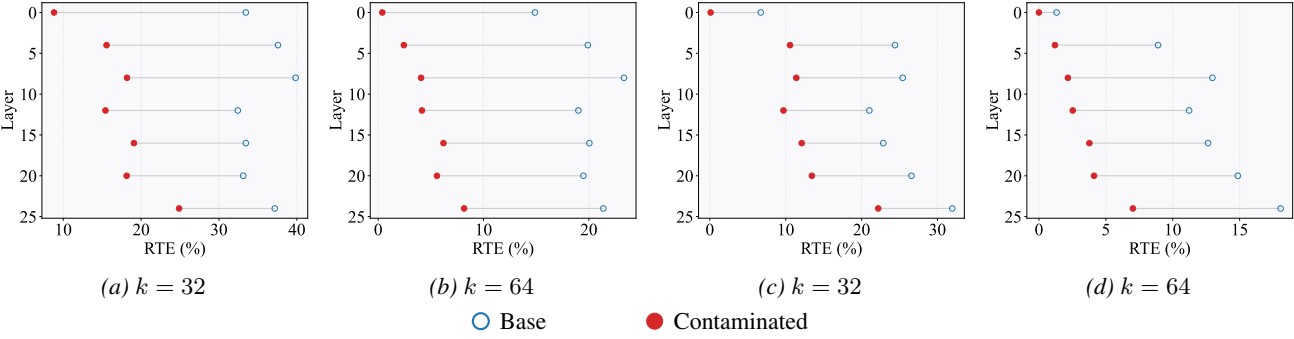

*Figure 49.* RTE analysis for Qwen2.5-7B-Instruct on MSCalc. mlp.up_proj: (49a, 49b); self_attn.q_proj: (49c, 49d).

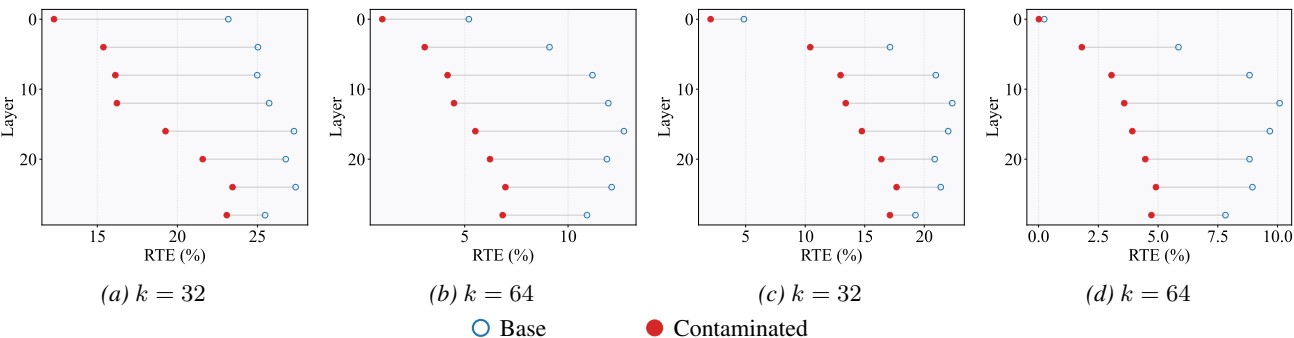

*Figure 50.* RTE analysis for Llama3.1-8B-Instruct on GSM8K. mlp.up_proj: (50a, 50b); self_attn.q_proj: (50c, 50d).

### C.4. From Spectral Concentration to Low-Dimensional Regime

Tables 3 to 5 summarize the norm analysis results across various configurations. These results consistently demonstrate that the weight matrices do not significantly amplify tail eigenvectors.

The causal relationship between spectral concentration and low-dimensional computation is further validated across other settings in Figures 51 to 70. As expected, flattening the spectrum increases RTE, while sharpening it decreases RTE, confirming that spectral concentration drives low-dimensional computation.

*Table 3.* Norm ratios of mapped tail eigenvectors to principal eigenvectors on MSCalc (Qwen2.5-7B-Instruct).

| MODELS | METRICS | $k$ | | | |
|---|---|---|---|---|---|
| | | 8 | 16 | 32 | 64 |
| BASE | $R_{\text{worst}}$ | 1.74 | 1.72 | 1.67 | 1.61 |
| | $R_{\text{med}}$ | 0.94 | 0.94 | 0.95 | 0.97 |
| CONTAMINATED | $R_{\text{worst}}$ | 1.48 | 1.48 | 1.48 | 1.38 |
| | $R_{\text{med}}$ | 0.95 | 0.95 | 0.97 | 0.99 |

*Table 4.* Norm ratios of mapped tail eigenvectors to principal eigenvectors (Qwen2.5-1.5B-Instruct).

| MODELS | METRICS | GSM8K | | | | MSCALC | | | |
|---|---|---|---|---|---|---|---|---|---|
| | | $k$ | | | | $k$ | | | |
| | | 8 | 16 | 32 | 64 | 8 | 16 | 32 | 64 |
| BASE | $R_{\text{worst}}$ | 1.89 | 1.89 | 1.89 | 1.89 | 1.60 | 1.60 | 1.60 | 1.60 |
| | $R_{\text{med}}$ | 1.03 | 1.02 | 1.02 | 1.02 | 1.04 | 1.03 | 1.02 | 1.02 |
| CONTAMINATED | $R_{\text{worst}}$ | 1.65 | 1.65 | 1.65 | 1.65 | 1.21 | 1.22 | 1.22 | 1.22 |
| | $R_{\text{med}}$ | 1.03 | 1.03 | 1.02 | 1.03 | 1.01 | 1.02 | 1.03 | 1.04 |

*Table 5.* Norm ratios of mapped tail eigenvectors to principal eigenvectors (Llama3.1-8B-Instruct).

| MODELS | METRICS | GSM8K | | | | MSCALC | | | |
|---|---|---|---|---|---|---|---|---|---|
| | | $k$ | | | | $k$ | | | |
| | | 8 | 16 | 32 | 64 | 8 | 16 | 32 | 64 |
| BASE | $R_{\text{worst}}$ | 1.35 | 1.37 | 1.39 | 1.39 | 1.31 | 1.35 | 1.37 | 1.37 |
| | $R_{\text{med}}$ | 0.92 | 0.94 | 0.96 | 0.99 | 0.93 | 0.95 | 0.96 | 0.99 |
| CONTAMINATED | $R_{\text{worst}}$ | 1.38 | 1.41 | 1.42 | 1.32 | 1.29 | 1.34 | 1.36 | 1.34 |
| | $R_{\text{med}}$ | 0.96 | 0.97 | 0.98 | 1.01 | 0.97 | 0.98 | 1.00 | 1.02 |

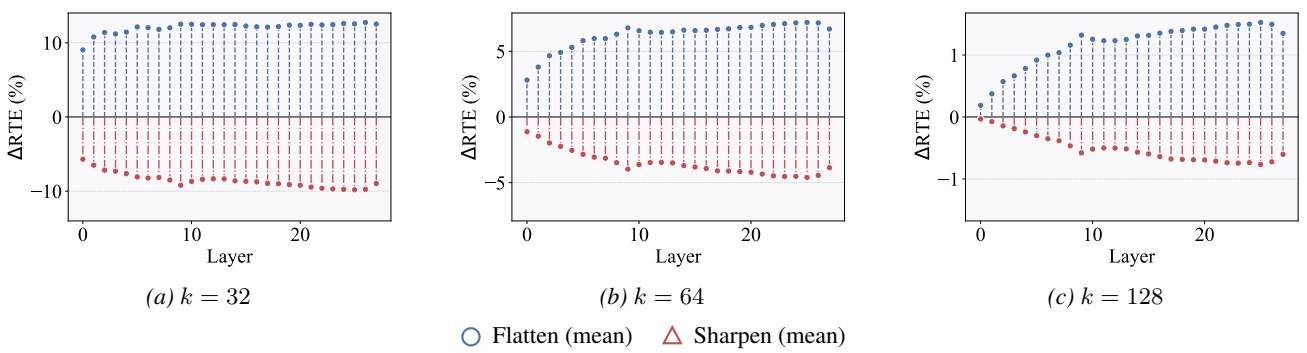

*Figure 51.* Causal effects for Qwen2.5-7B-Instruct (`mlp.up_proj`). $\tau = \pm 0.3$

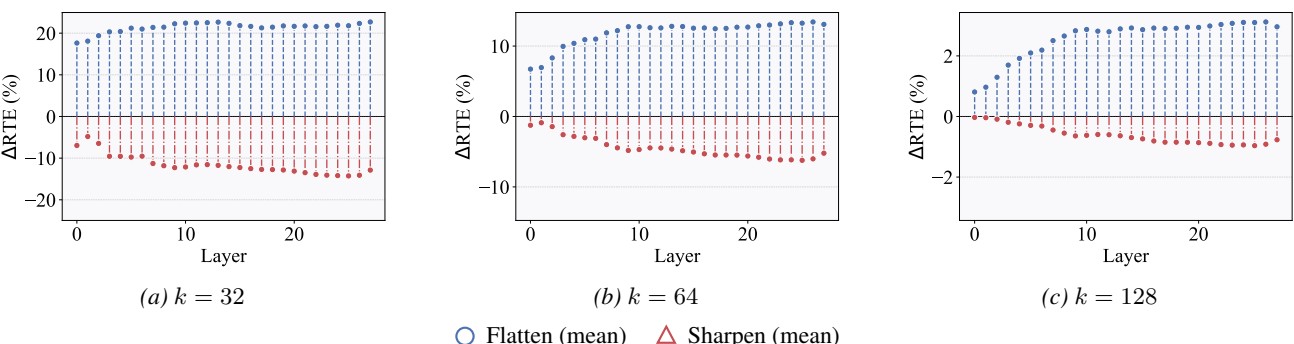

*Figure 52.* Causal effects for Qwen2.5-7B-Instruct (`mlp.gate_proj`). $\tau = \pm 0.5$

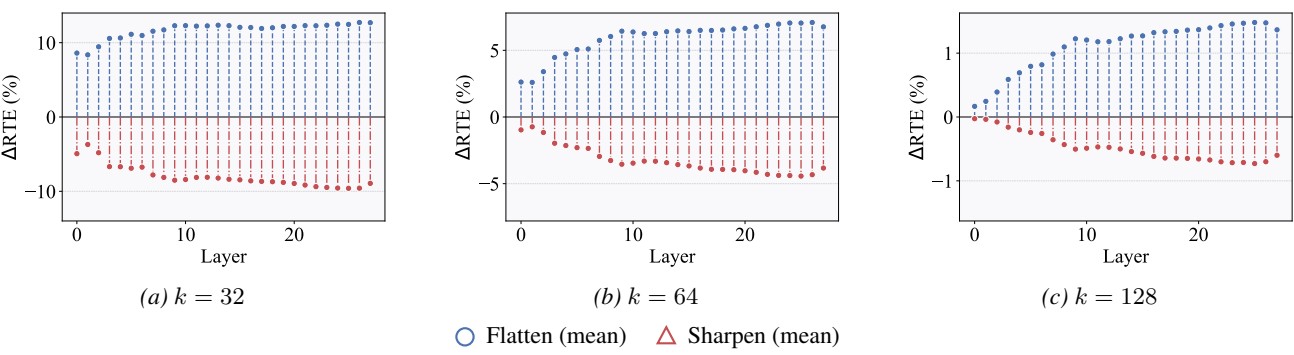

*Figure 53.* Causal effects for Qwen2.5-7B-Instruct (`mlp.gate_proj`). $\tau = \pm 0.3$

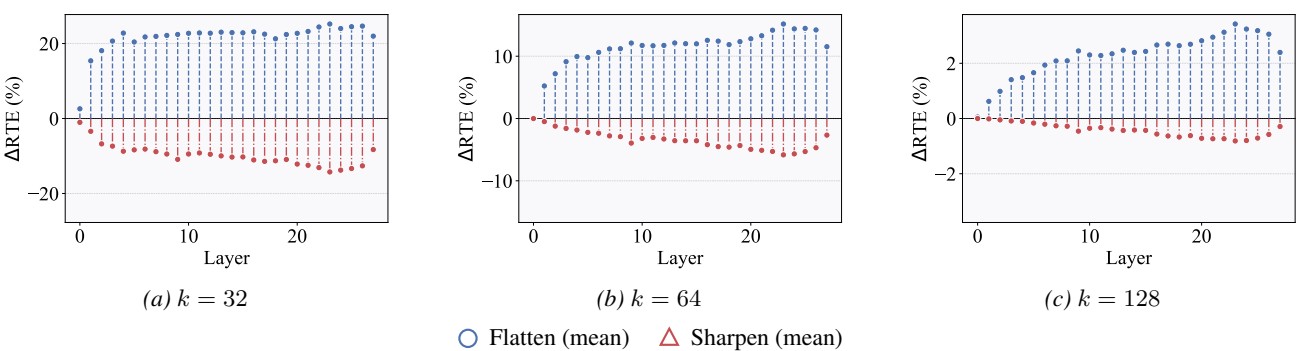

*Figure 54.* Causal effects for Qwen2.5-7B-Instruct (`self_attn.q_proj`). $\tau = \pm 0.5$

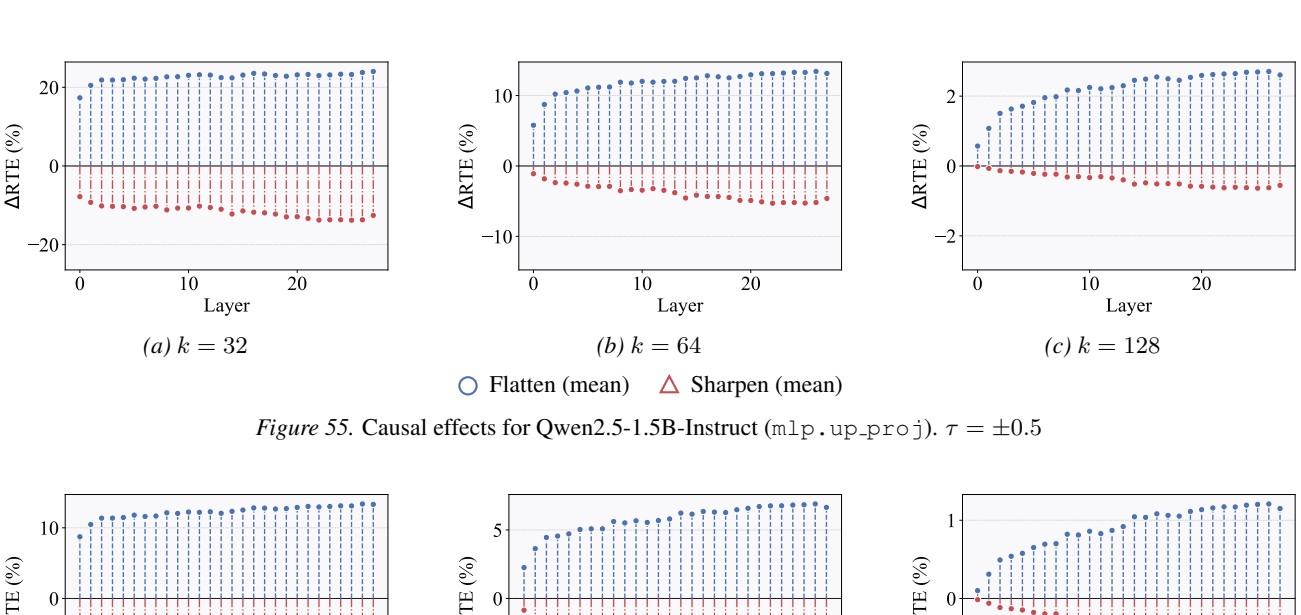

*Figure 55.* Causal effects for Qwen2.5-1.5B-Instruct (`mlp.up_proj`). $\tau = \pm 0.5$

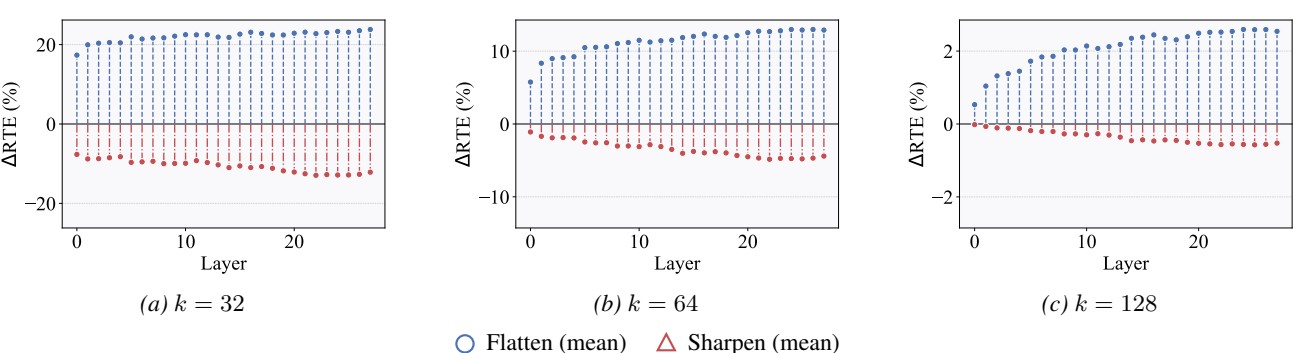

*Figure 56.* Causal effects for Qwen2.5-1.5B-Instruct (`mlp.up_proj`). $\tau = \pm 0.3$

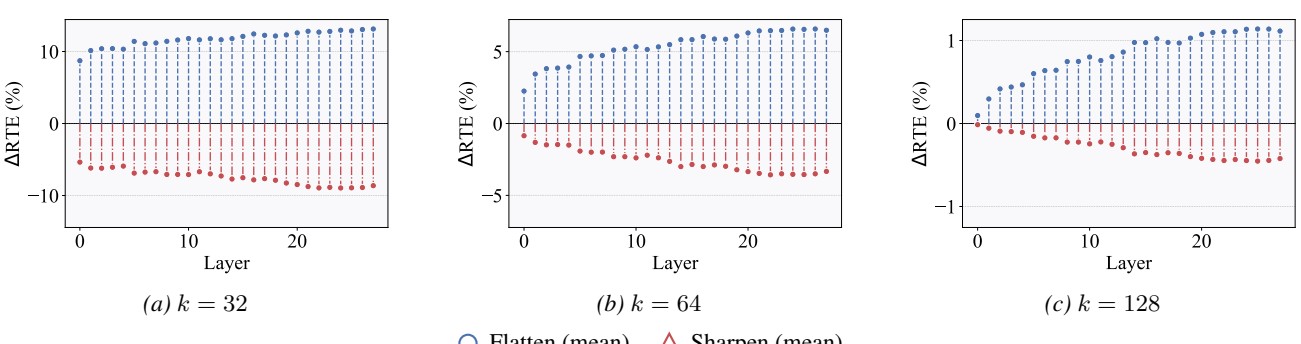

*Figure 57.* Causal effects for Qwen2.5-1.5B-Instruct (`mlp.gate_proj`). $\tau = \pm 0.5$

*Figure 58.* Causal effects for Qwen2.5-1.5B-Instruct (`mlp.gate_proj`). $\tau = \pm 0.3$

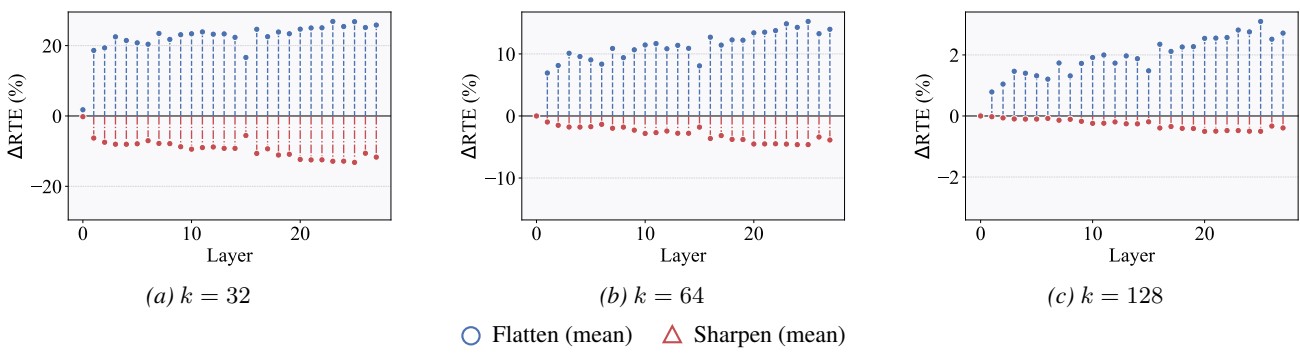

*(a) k = 32*      *(b) k = 64*      *(c) k = 128*

○ Flatten (mean)    △ Sharpen (mean)

*Figure 59.* Causal effects for Qwen2.5-1.5B-Instruct (`self_attn.q_proj`). $\tau = \pm 0.5$

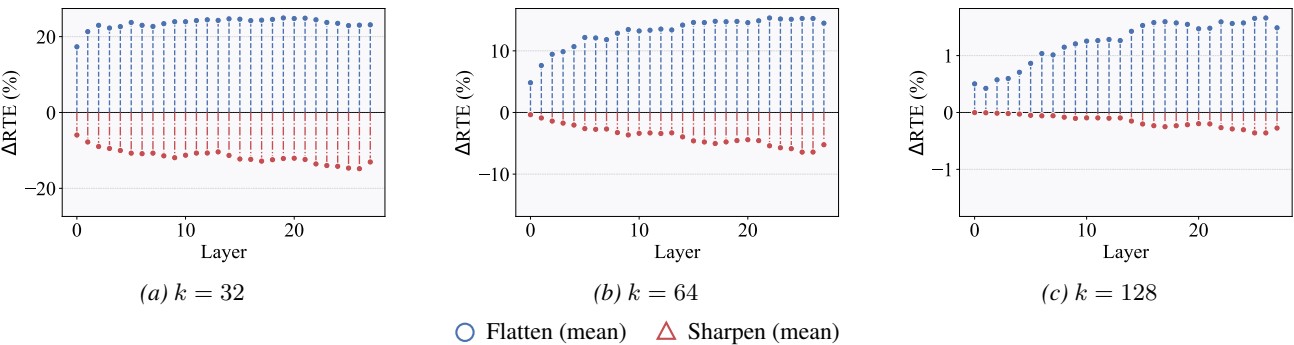

*(a) k = 32*      *(b) k = 64*      *(c) k = 128*

○ Flatten (mean)    △ Sharpen (mean)

*Figure 60.* Causal effects for Qwen2.5-7B-Instruct (`mlp.up_proj`) on MSCalc. $\tau = \pm 0.5$

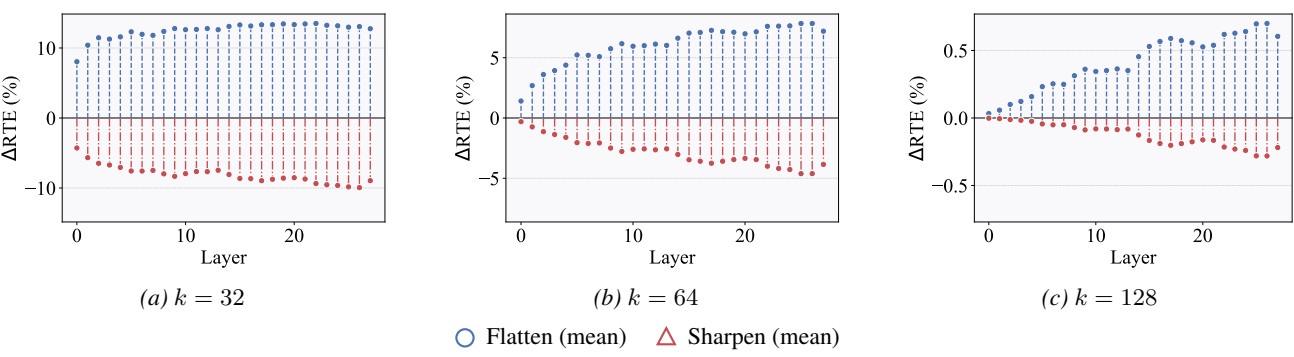

*(a) k = 32*      *(b) k = 64*      *(c) k = 128*

○ Flatten (mean)    △ Sharpen (mean)

*Figure 61.* Causal effects for Qwen2.5-7B-Instruct (`mlp.up_proj`) on MSCalc. $\tau = \pm 0.3$

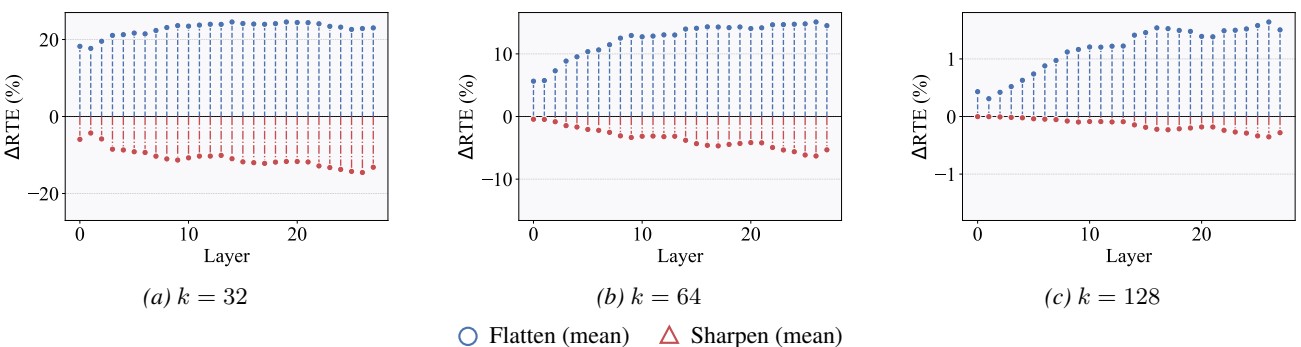

*(a) k = 32*      *(b) k = 64*      *(c) k = 128*

○ Flatten (mean)    △ Sharpen (mean)

*Figure 62.* Causal effects for Qwen2.5-7B-Instruct (`mlp.gate_proj`) on MSCalc. $\tau = \pm 0.5$

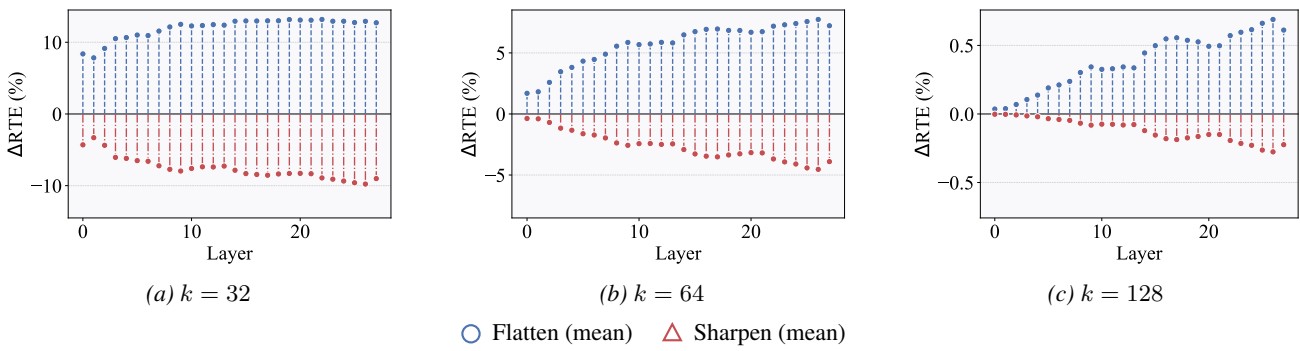

*(a) k = 32*  *(b) k = 64*  *(c) k = 128*

○ Flatten (mean)  △ Sharpen (mean)

*Figure 63.* Causal effects for Qwen2.5-7B-Instruct (`mlp.gate_proj`) on MSCalc. $\tau = \pm 0.3$

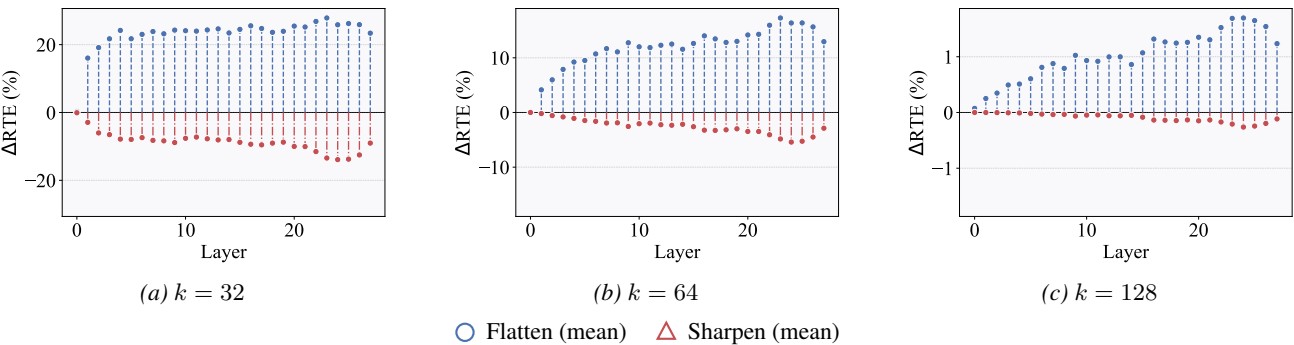

*(a) k = 32*  *(b) k = 64*  *(c) k = 128*

○ Flatten (mean)  △ Sharpen (mean)

*Figure 64.* Causal effects for Qwen2.5-7B-Instruct (`self_attn.q_proj`) on MSCalc. $\tau = \pm 0.5$

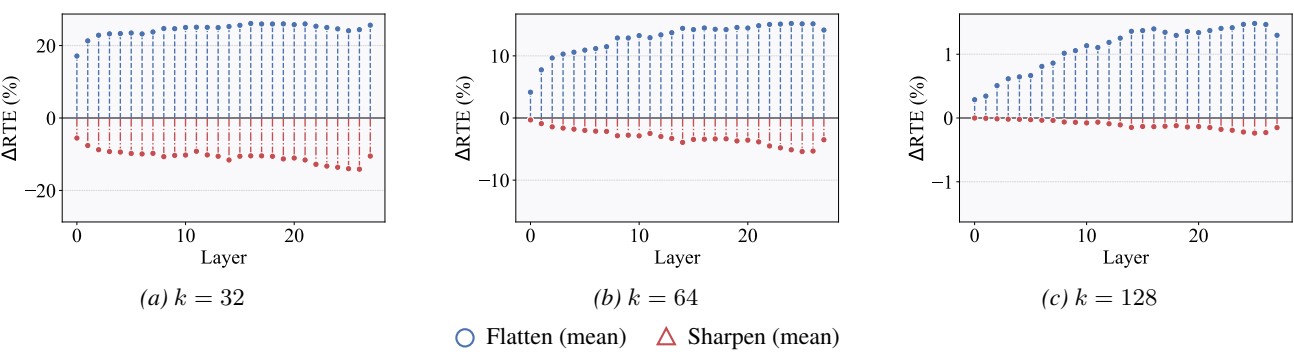

*(a) k = 32*  *(b) k = 64*  *(c) k = 128*

○ Flatten (mean)  △ Sharpen (mean)

*Figure 65.* Causal effects for Qwen2.5-1.5B-Instruct (`mlp.up_proj`) on MSCalc. $\tau = \pm 0.5$

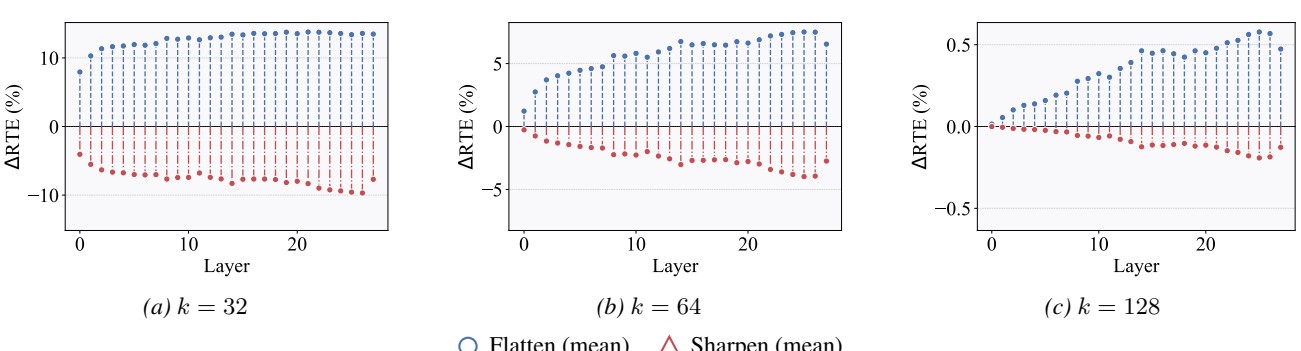

*(a) k = 32*  *(b) k = 64*  *(c) k = 128*

○ Flatten (mean)  △ Sharpen (mean)

*Figure 66.* Causal effects for Qwen2.5-1.5B-Instruct (`mlp.up_proj`) on MSCalc. $\tau = \pm 0.3$

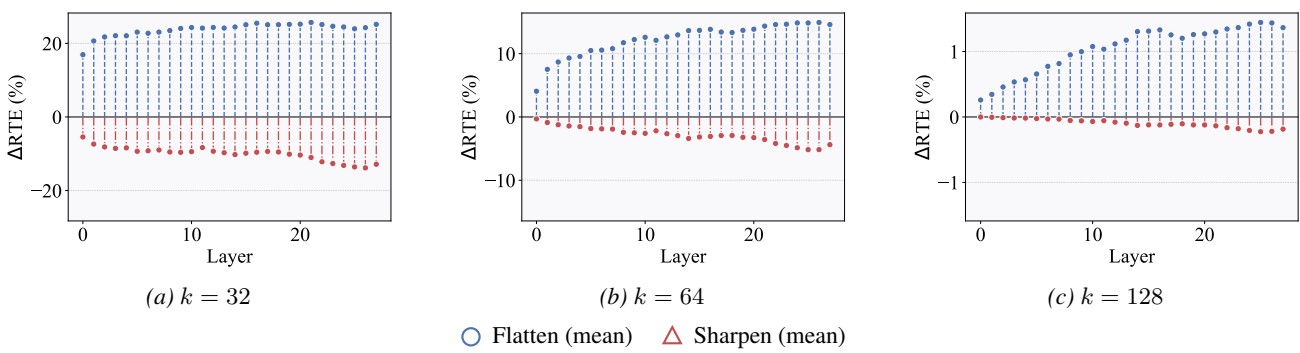

*Figure 67.* Causal effects for Qwen2.5-1.5B-Instruct (`mlp.gate_proj`) on MSCalc. $\tau = \pm 0.5$

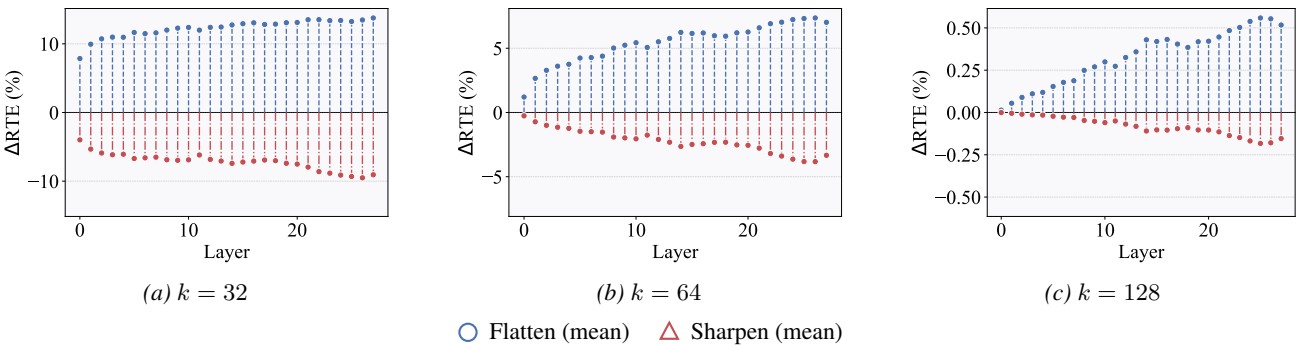

*Figure 68.* Causal effects for Qwen2.5-1.5B-Instruct (`mlp.gate_proj`) on MSCalc. $\tau = \pm 0.3$

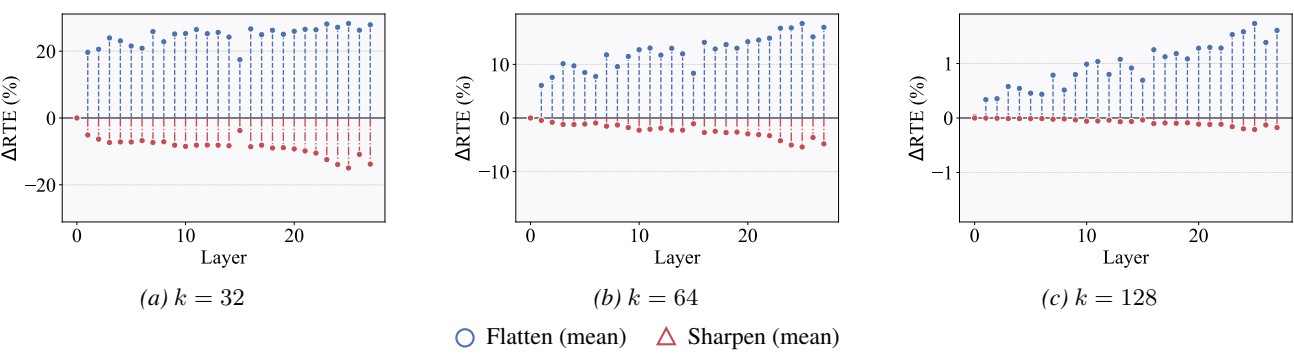

*Figure 69.* Causal effects for Qwen2.5-1.5B-Instruct (`self_attn.q_proj`) on MSCalc. $\tau = \pm 0.5$

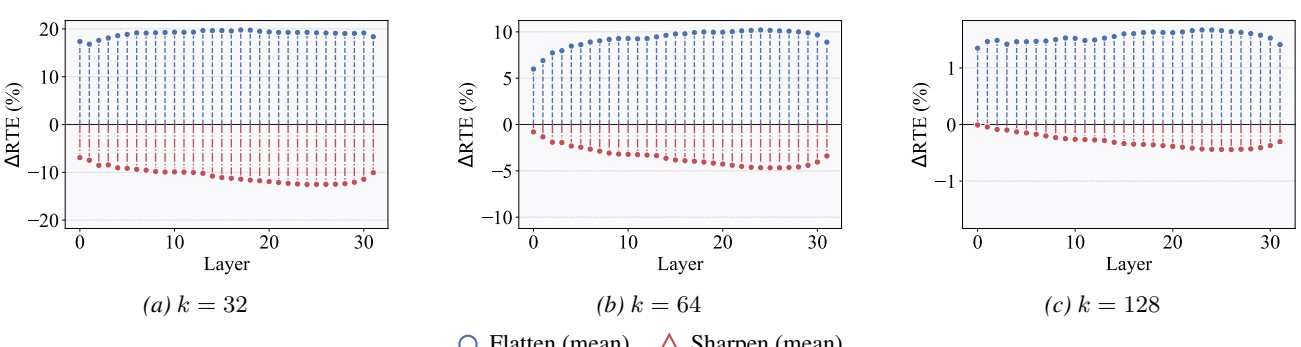

*Figure 70.* Causal effects for Llama3.1-8B-Instruct. $\tau = \pm 0.5$

## C.5. Validation of Theoretical Conditions for Mutual Information Decay

Figures 71 to 82 present the MISA results across additional settings. These consistent findings validate that raw mutual information is primarily confined within an active computational subspace.

As illustrated in Figures 83 to 88, the cross-covariance norm for leaked data is consistently lower than that of clean data. Figures 89 to 93 compare the average centered gradient norms. The lower norm observed on leaked versus clean data substantiate the gradient shrinkage occurring on optimized leaked samples.

Figures 94 and 95 show the results for the SVPO condition, where we observe a consistent structural decay of the singular value spectrum in the whitened space.

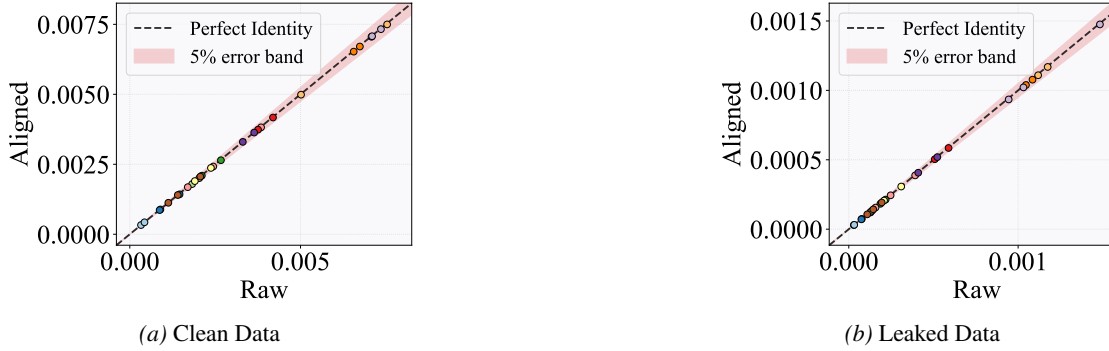

*(a)* Clean Data          *(b)* Leaked Data

*Figure 71.* MISA validation of Qwen2.5-7B-Instruct on GSM8K (`self_attn.q_proj`).

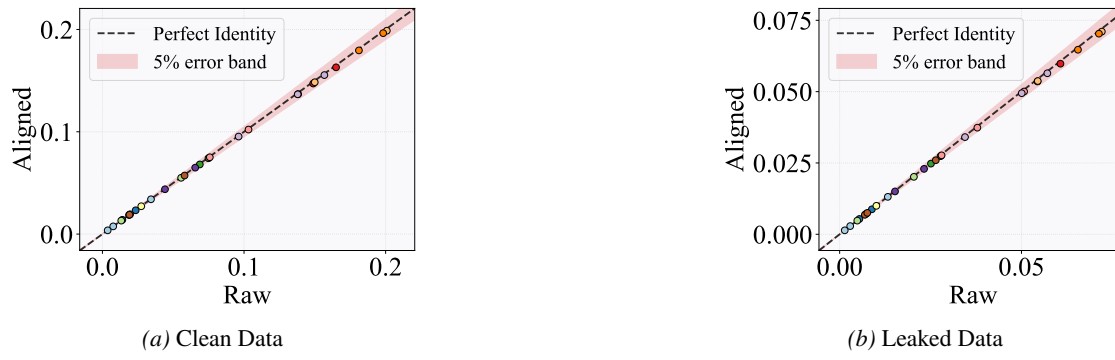

*(a)* Clean Data          *(b)* Leaked Data

*Figure 72.* MISA validation of Qwen2.5-1.5B-Instruct on GSM8K (`mlp.up_proj`).

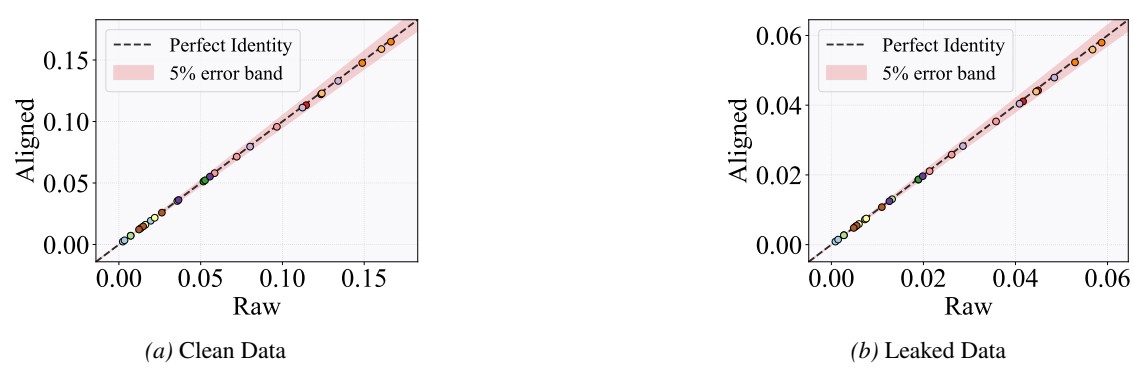

*(a)* Clean Data          *(b)* Leaked Data

*Figure 73.* MISA validation of Qwen2.5-1.5B-Instruct on GSM8K (`mlp.gate_proj`).

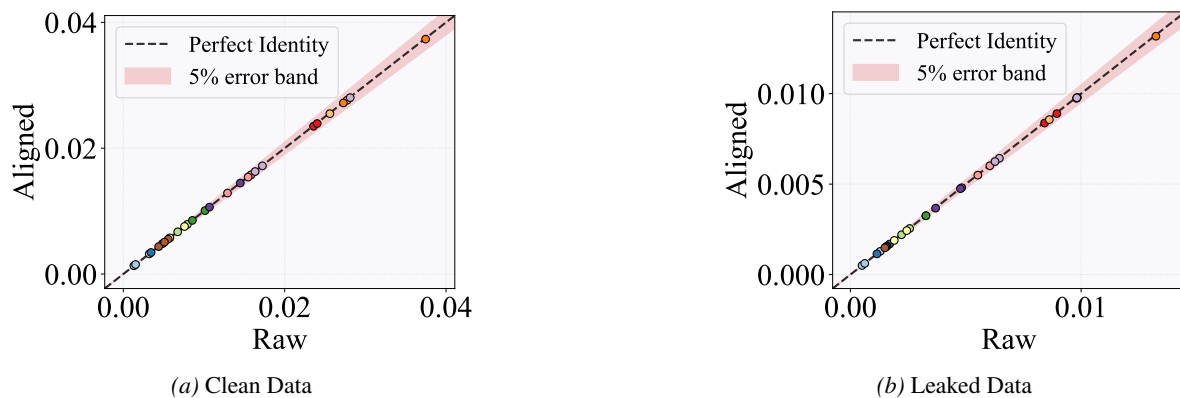

*(a)* Clean Data

*(b)* Leaked Data

*Figure 74.* MISA validation of Qwen2.5-1.5B-Instruct on GSM8K (`self_attn.q_proj`).

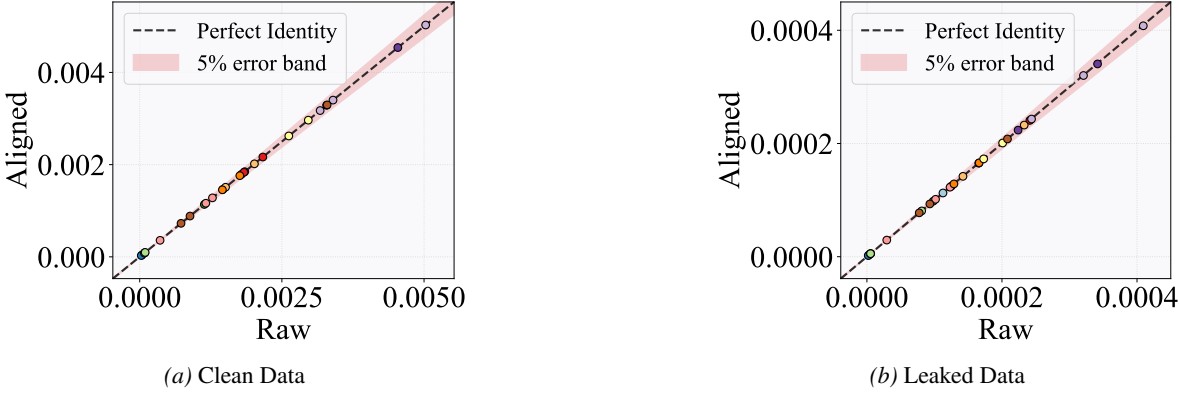

*(a)* Clean Data

*(b)* Leaked Data

*Figure 75.* MISA validation of Qwen2.5-7B-Instruct on MSCalc (`mlp.up_proj`).

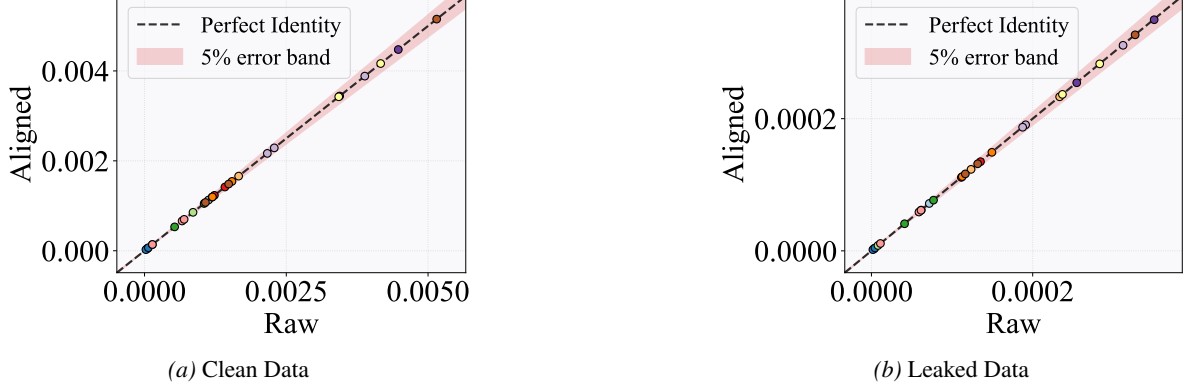

*(a)* Clean Data

*(b)* Leaked Data

*Figure 76.* MISA validation of Qwen2.5-7B-Instruct on MSCalc (`mlp.gate_proj`).

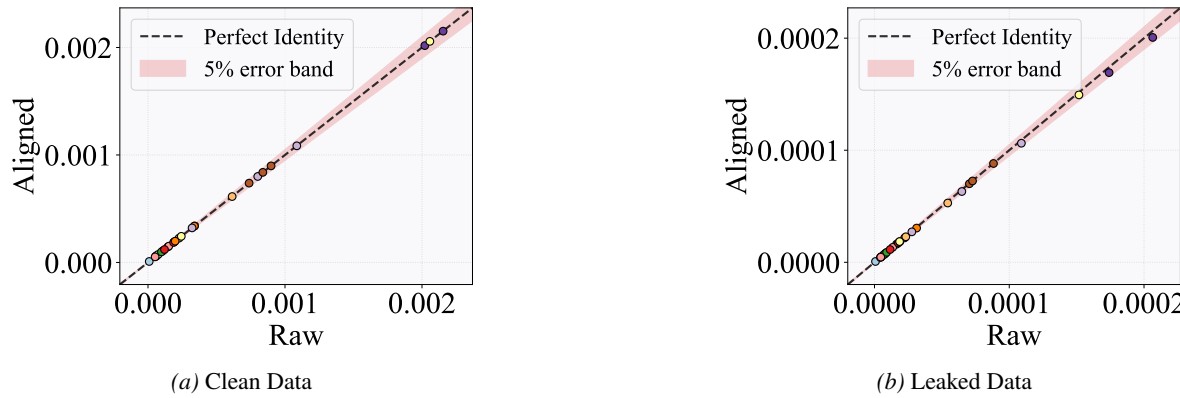

*(a)* Clean Data       *(b)* Leaked Data

*Figure 77.* MISA validation of Qwen2.5-7B-Instruct on MSCalc (`self_attn.q_proj`).

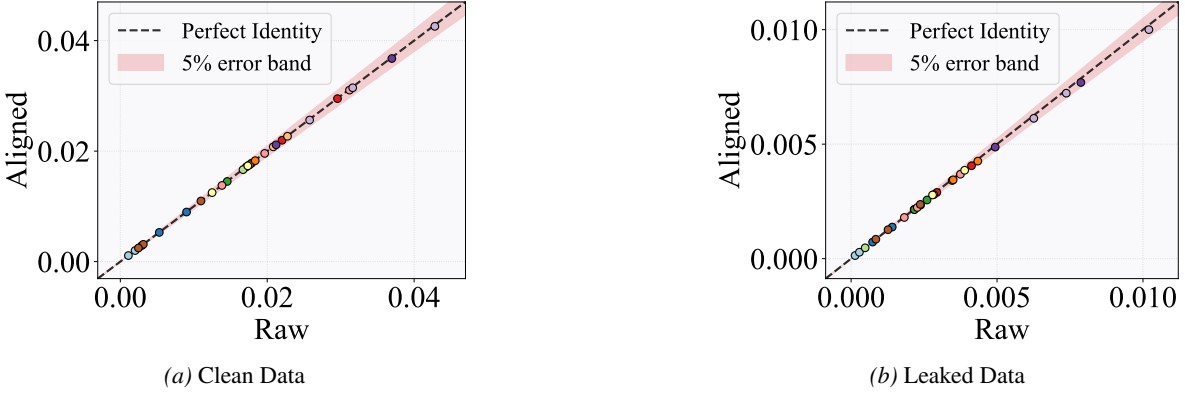

*(a)* Clean Data       *(b)* Leaked Data

*Figure 78.* MISA validation of Qwen2.5-1.5B-Instruct on MSCalc (`mlp.up_proj`).

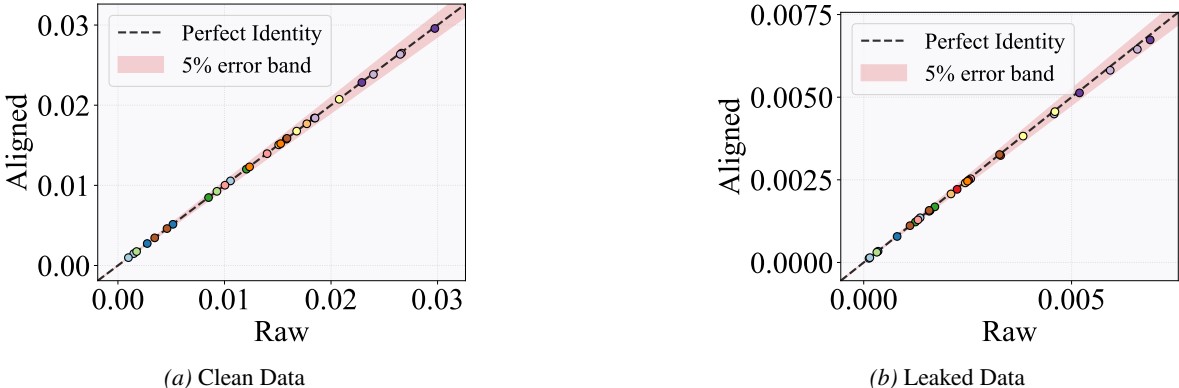

*(a)* Clean Data       *(b)* Leaked Data

*Figure 79.* MISA validation of Qwen2.5-1.5B-Instruct on MSCalc (`mlp.gate_proj`).

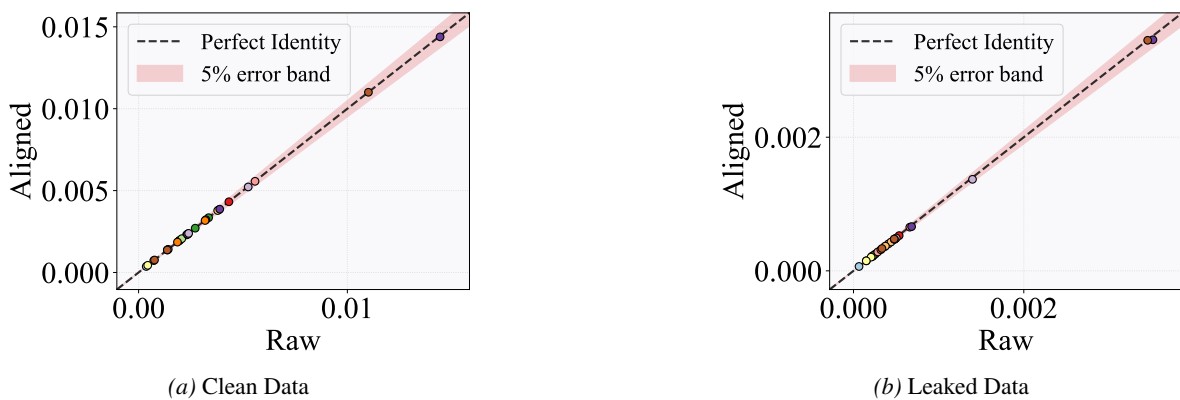

*(a)* Clean Data

*(b)* Leaked Data

*Figure 80.* MISA validation of Qwen2.5-1.5B-Instruct on MSCalc (`self_attn.q_proj`).

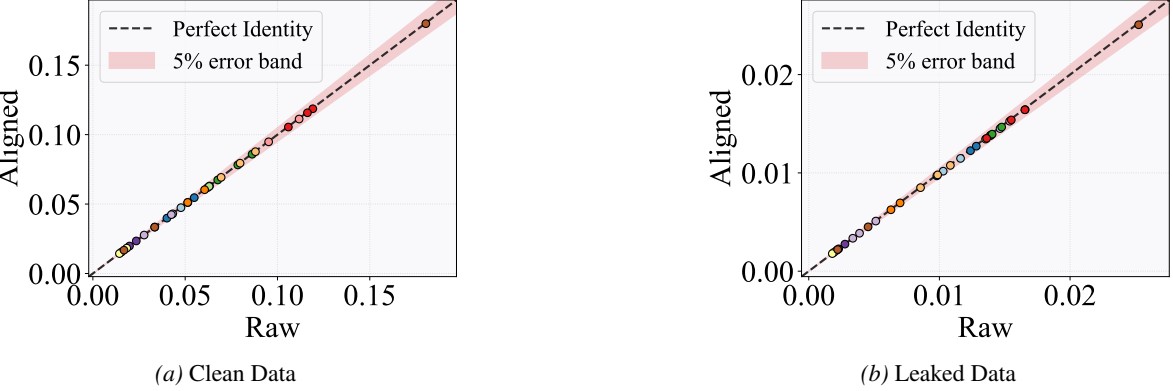

*(a)* Clean Data

*(b)* Leaked Data

*Figure 81.* MISA validation of Llama3.1-8B-Instruct (`mlp.up_proj`).

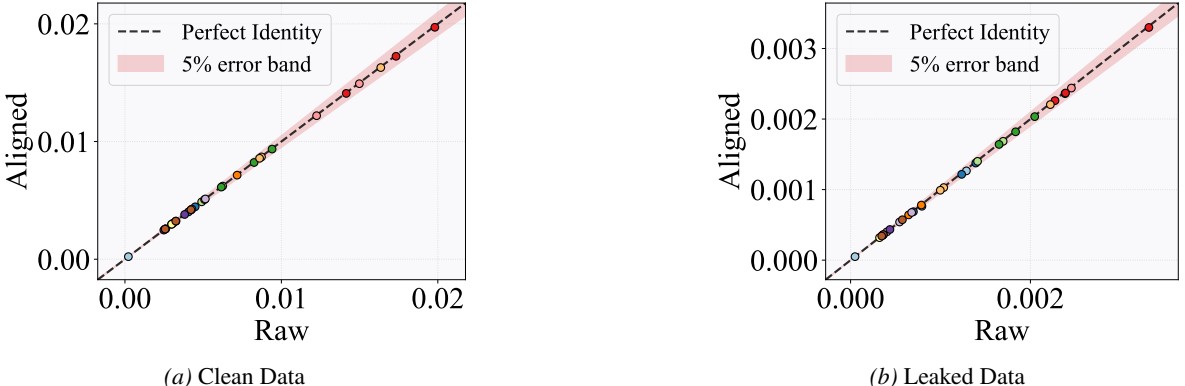

*(a)* Clean Data

*(b)* Leaked Data

*Figure 82.* MISA validation of Llama3.1-8B-Instruct (`self_attn.q_proj`).

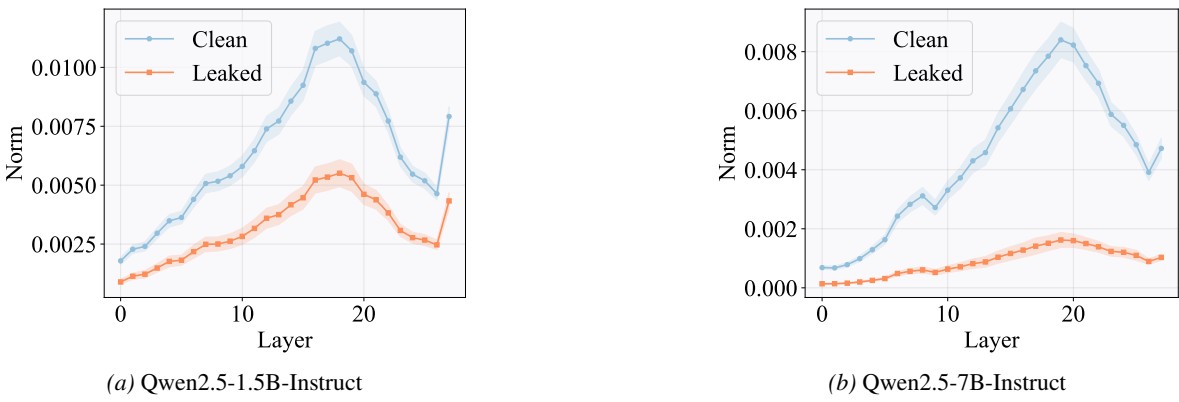

*(a)* Qwen2.5-1.5B-Instruct

*(b)* Qwen2.5-7B-Instruct

*Figure 83.* Norms of cross-covariance matrices for leaked data and clean data on GSM8K (`mlp.gate_proj`).

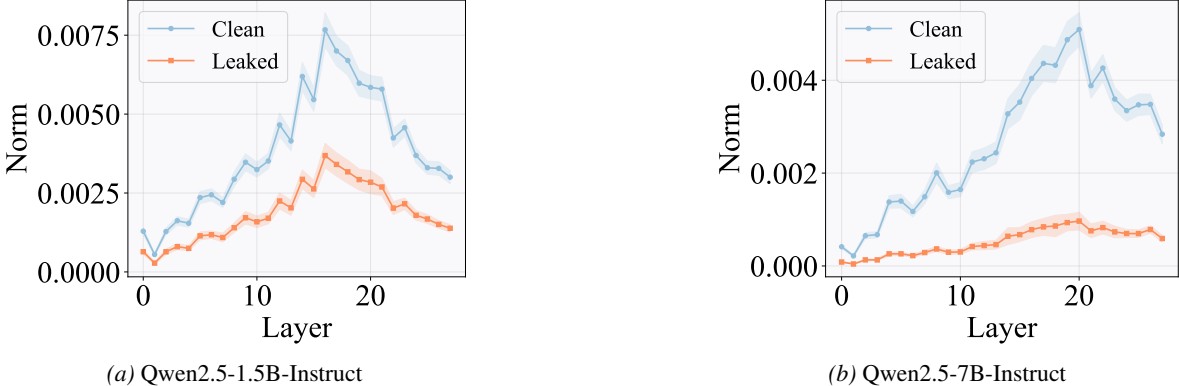

*(a)* Qwen2.5-1.5B-Instruct

*(b)* Qwen2.5-7B-Instruct

*Figure 84.* Norms of cross-covariance matrices for leaked data and clean data on GSM8K (`self_attn.q_proj`).

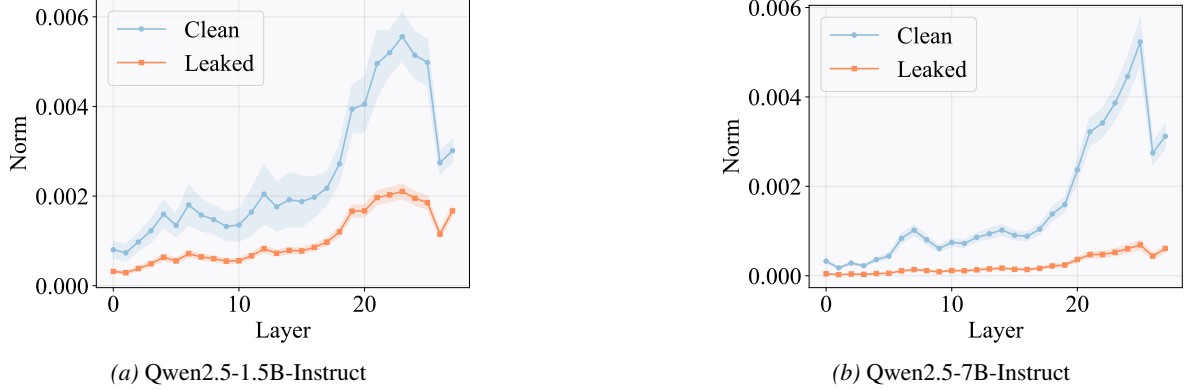

*(a)* Qwen2.5-1.5B-Instruct

*(b)* Qwen2.5-7B-Instruct

*Figure 85.* Norms of cross-covariance matrices for leaked data and clean data on MSCalc (`mlp.up_proj`).

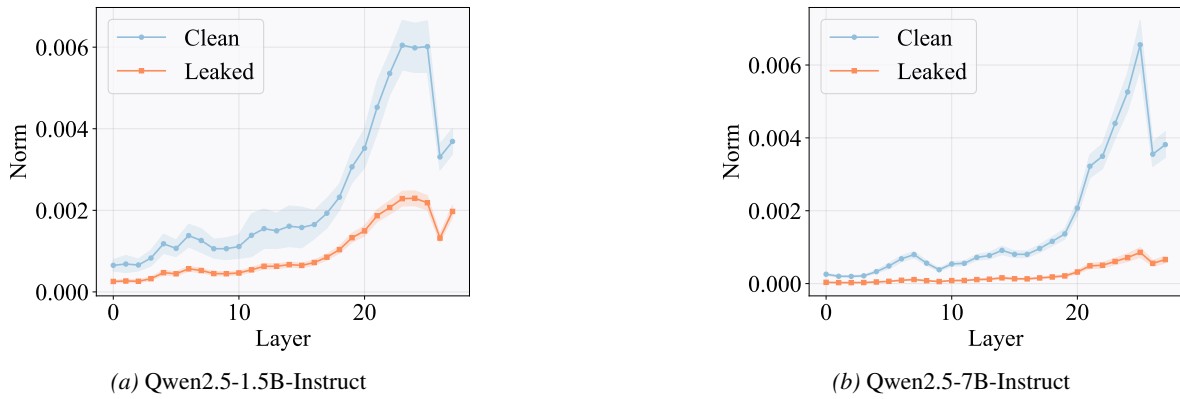

*(a)* Qwen2.5-1.5B-Instruct        *(b)* Qwen2.5-7B-Instruct

*Figure 86.* Norms of cross-covariance matrices for leaked data and clean data on MSCalc (`mlp.gate_proj`).

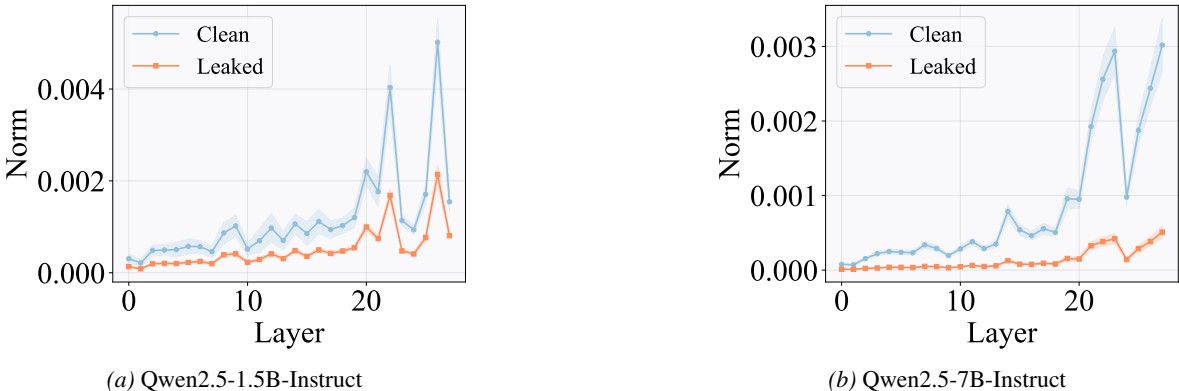

*(a)* Qwen2.5-1.5B-Instruct        *(b)* Qwen2.5-7B-Instruct

*Figure 87.* Norms of cross-covariance matrices for leaked data and clean data on MSCalc (`self_attn.q_proj`).

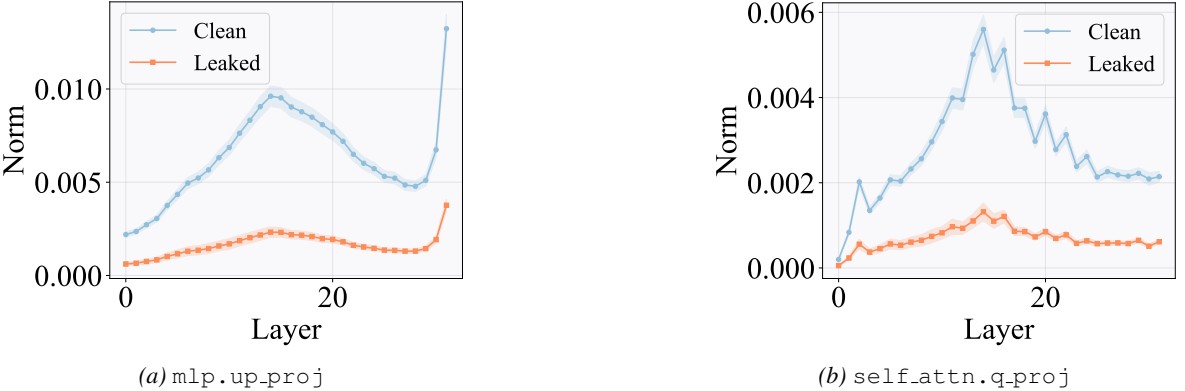

*(a)* `mlp.up_proj`        *(b)* `self_attn.q_proj`

*Figure 88.* Norms of cross-covariance matrices for leaked data and clean data (Llama3.1-8B-Instruct).

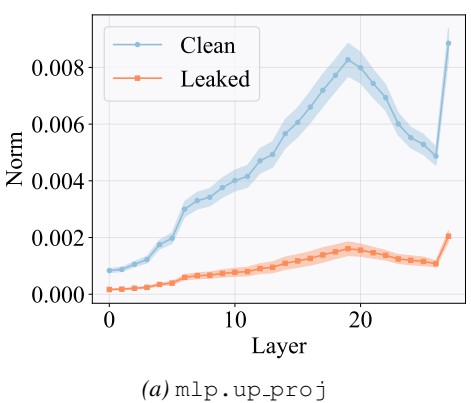
*(a)* `mlp.up_proj`

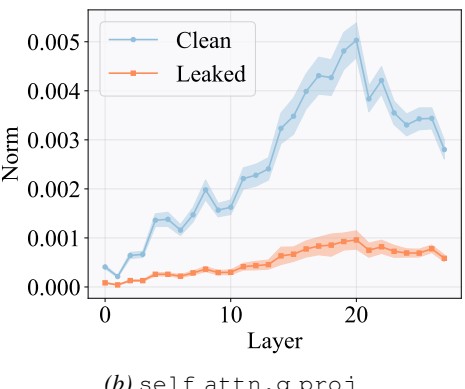
*(b)* `self_attn.q_proj`

*Figure 89.* Norms of the average centered weight gradient for leaked data and clean data on GSM8K (Qwen2.5-7B-Instruct).

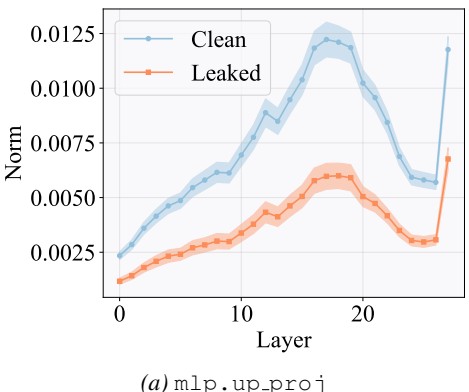
*(a)* `mlp.up_proj`

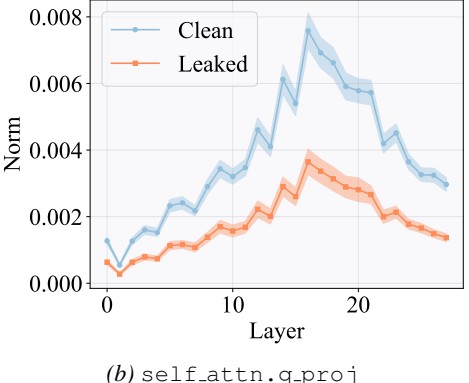
*(b)* `self_attn.q_proj`

*Figure 90.* Norms of the average centered weight gradient for leaked data and clean data on GSM8K (Qwen2.5-1.5B-Instruct).

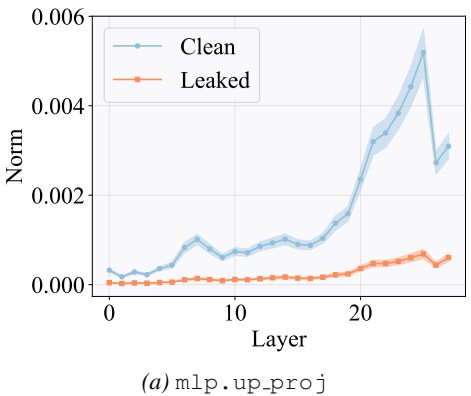
*(a)* `mlp.up_proj`

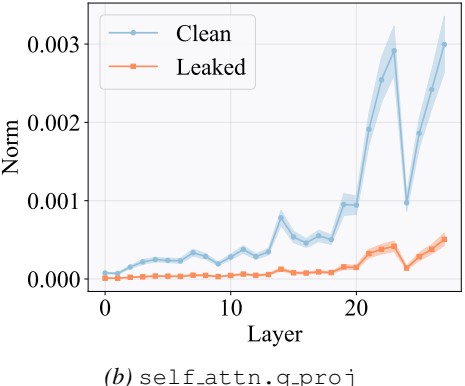
*(b)* `self_attn.q_proj`

*Figure 91.* Norms of the average centered weight gradient for leaked data and clean data on MSCalc (Qwen2.5-7B-Instruct).

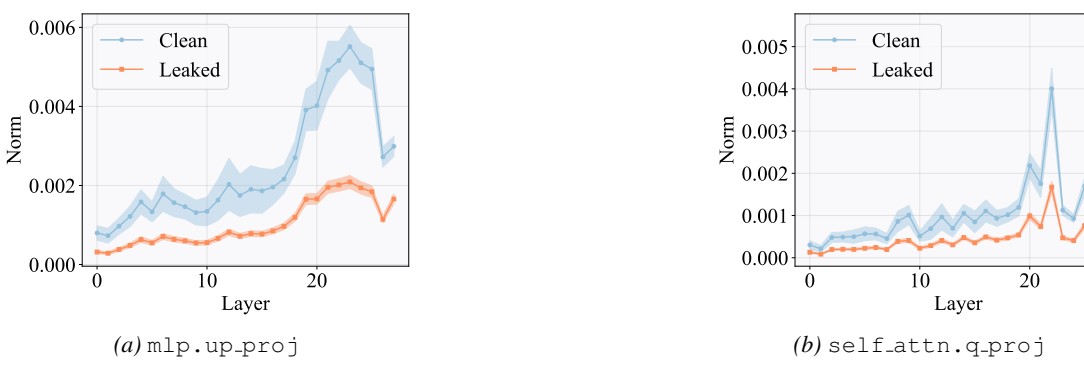

*(a)* `mlp.up_proj`                     *(b)* `self_attn.q_proj`

*Figure 92.* Norms of the average centered weight gradient for leaked data and clean data on MSCalc (Qwen2.5-1.5B-Instruct).

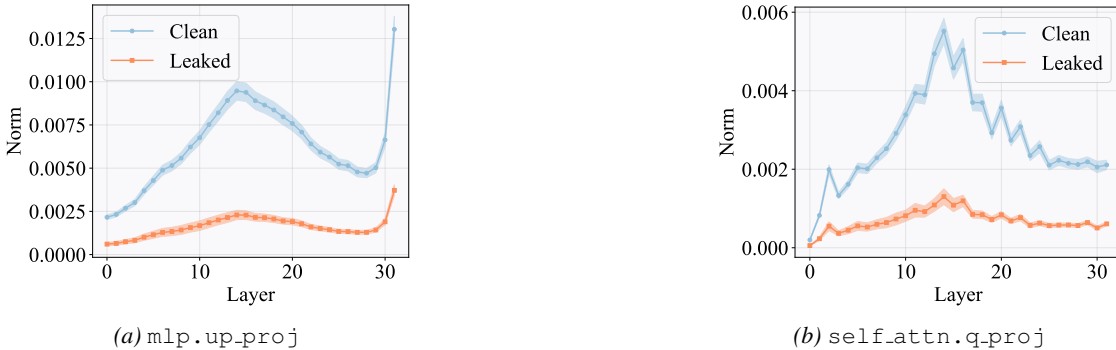

*(a)* `mlp.up_proj`                     *(b)* `self_attn.q_proj`

*Figure 93.* Norms of the average centered weight gradient for leaked data and clean data (Llama3.1-8B-Instruct).

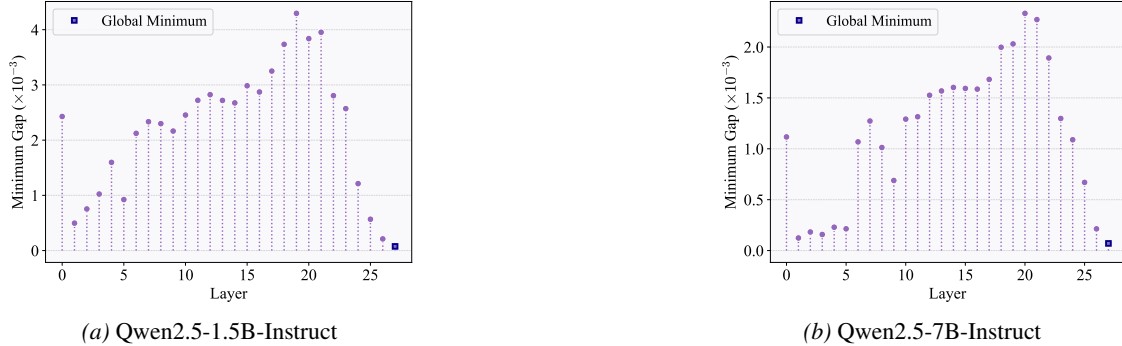

*(a)* Qwen2.5-1.5B-Instruct             *(b)* Qwen2.5-7B-Instruct

*Figure 94.* The layer-wise minimum gap between the spectra of clean and leaked data on MSCalc (`mlp.up_proj`).

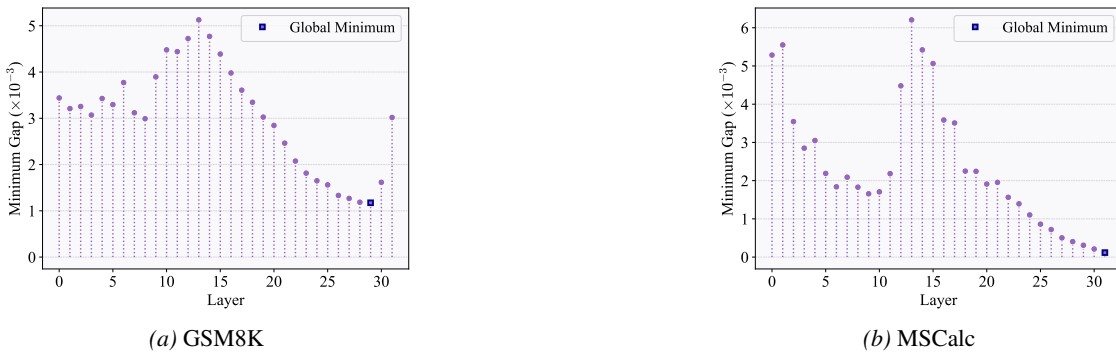

*(a)* GSM8K                             *(b)* MSCalc

*Figure 95.* The layer-wise minimum gap between the spectra of clean and leaked data for Llama3.1-8B-Instruct (`mlp.up_proj`).

## C.6. Reasoning Restoration

In this section, we briefly introduce two baseline methods for contamination mitigation evaluation in large language models (LLMs): TED (Trustworthy Evaluation via output Distribution) (Dong et al., 2024) and LNE-Blocking (Hou et al., 2025).

TED is a sampling-based method designed to reduce the influence of memorized test data on LLM evaluation. The method operates by sampling multiple outputs from the model for a given prompt, computing edit distance between each sampled output and the greedy-decoded output, filtering out samples whose edit distance falls below a threshold, and finally removing duplicates among the remaining samples.

LNE-Blocking is a framework that adaptively restores a model's true performance under data contamination. It consists of two key components: (1) Length Normalized Entropy (LNE), which estimates the contamination level of the model on a given prompt by computing the entropy of the token probability distribution during greedy decoding; and (2) Blocking operation, which suppresses the token with the highest probability at certain positions, thereby forcing the model to generate alternative tokens and reducing reliance on memorized content.

Intervention results for other settings are reported in Tables 6 and 7. Our method consistently enables the contaminated model to achieve the highest consistency with the base model.

*Table 6.* Results of reasoning restoration on MSCalc.

| MODELS | STRATEGY | RESTOR. (↑) | RETAIN (↑) | AVG. (↑) |
|---|---|---|---|---|
| | CONT. | 0.00 | 100.00 | 50.00 |
| | SAMPLING | 14.45 | 97.71 | 56.08 |
| QWEN-1.5B | TED | 37.34 | 97.70 | 67.52 |
| | LNE-B. | 39.76 | 99.24 | 69.50 |
| | TSS (OURS) | **46.98** | **99.24** | **73.11** |
| | CONT. | 0.00 | 100.00 | 50.00 |
| | SAMPLING | 13.54 | 99.22 | 56.38 |
| QWEN-7B | TED | 37.50 | 98.46 | 67.98 |
| | LNE-B. | 38.54 | 96.18 | 67.36 |
| | TSS (OURS) | **55.20** | **99.22** | **77.21** |

*Table 7.* Results of reasoning restoration of Llama3.1-8B-Instruct.

| DATASET | STRATEGY | RESTOR. (↑) | RETAIN (↑) | AVG. (↑) |
|---|---|---|---|---|
| | CONT. | 0.00 | 100.00 | 50.00 |
| | SAMPLING | 7.69 | 91.37 | 49.53 |
| GSM8K | TED | 15.38 | 88.48 | 51.93 |
| | LNE-B. | 30.76 | 89.92 | 60.34 |
| | TSS (OURS) | **53.84** | **94.24** | **74.04** |
| | CONT. | 0.00 | 100.00 | 50.00 |
| | SAMPLING | 2.70 | 100.00 | 51.35 |
| MSCALC | TED | 8.11 | 99.43 | 53.77 |
| | LNE-B. | 64.86 | 95.42 | 80.14 |
| | TSS (OURS) | **70.26** | **100.00** | **85.13** |

