# OpenReview forum: "A Narrowing Geometry in Contaminated Reasoning"
_ICML.cc/2026/Conference — ICML 2026 regular_

### Official Review · Reviewer_hd3F · 2026-03-10

**Soundness:** 2
**Presentation:** 2
**Significance:** 3
**Originality:** 2
**Overall Recommendation:** 2
**Confidence:** 4

**Summary:**

Reasoning evals are compromised by data contamination. The authors in this paper want to derive new mechanisms to capture contaminated reasoning using mutual information between representations and gradients as well as allow us to better understand what happens in the presence of data leakage. Specifically, they look at how data contamination modifies the model’s internal representation.

**Compliance With Llm Reviewing Policy:**

Affirmed.

**Key Questions For Authors:**

1. How are you sure that Qwen2.5 and Llama3.1 were not trained on GSM8K? This dataset was released in 2021.

2. I think there might be a difference between having a SFT on a dataset and then performing analysis and analysing the data contamination on a model that went into a series of training phases. If you think about it, doing the analysis right after fine-tuning the model will for sure reveal more salient features than trying to analyze a model that undergoes multiple phases of training where some of them contained data contamination. I wonder what the author thinks about that?

3. “Our primary analysis uses the GSM8K test set (Cobbe et al., 2021), which we partition into a leaked subset for contamination injection via supervised fine-tuning and a remaining clean subset that is held out from the injection process. “ So from this, isn’t it obvious that when sending input that was used for training the model requires far less capacity than the one not used for training? In fact, when the data is already in weight space, there is no need to extrapolate or combine weights to build a large space as to find the solution of the problem, it is already localized in a low-dimensional space, as per the rank collapse phenomena of deep networks. I am just trying to understand if the results of the paper are natural or if they are highlighting something counterintuitive, and I would appreciate it if the authors could put a word on this in the paper.

4. All the figure captions would appreciate a bit more detail, what are the takeaways, what was computed, which layer, etc… Each caption should be as self-sufficient as possible.

5. Section 2, it would be nice to have more details about how you compute the mutual information. What are the quantities at hand, which layer, etc..

6. Figure 3 caption has the same issues as the other one, what are we computing, what is the takeaway.

7. Section 3.1, “for a given centered activation matrix”, yes but which one are you using? Which layer, which hidden representation, pre-mlp, pre self attention, post batch norm?

8. Figure  2 and 3 the y-axis should be aligned.

9. Section 3.2, again what are you computing? Which input space, the original one of the input space of a specific layer?

10. Section 3.3. Table 1, same problem, the maths are easy to understand, the problem is what are you computing and on what.

**Limitations:**

yes

**Strengths And Weaknesses:**

The paper discusses the understanding of data contamination in LLMs, and proposes a way to capture such a phenomenon in a mechanistic way. I think this is an interesting research direction for the community. Now, It is extremely difficult to understand what experiment has been performed. In the sense that the paper defines the math but not what is actually computed, such as,  what activation, what layer, what input space is getting used, even what data, there is no example of data being displayed. This drastically affects the readability, understanding, and reproducibility of the research.

---

> ### Author Rebuttal · Authors · 2026-03-26
>
> We thank the reviewer for the detailed feedback. The main concern appears to be clarity of experimental setup and the observed mechanisms. We clarify both below.
> ## 1. Response to Q1
> We address this concern from two perspectives:
> + We partition the GSM8K test set into SFT-injected and held-out subsets. Any prior model contamination acts as a global bias affecting both similarly. Thus, it cannot explain the systematic difference observed between these two subsets after our SFT.
> + We additionally construct a synthetic random calculation dataset following prior work (*Reasoning or Memorization? Unreliable Results of Reinforcement Learning Due to Data Contamination, Wu et al., AAAI 2026*). Because the questions are algorithmically generated using randomized operations and operands from a vast combinatorial space, and this generation process was executed strictly after the models' release dates, the synthetic dataset is effectively free from prior contamination. We observe consistent phenomena on this dataset (Appendix C), which further supports our findings.
>
> ## 2. Response to Q2
> We focus on SFT for two reasons:
> + **Isolating mechanisms** is a widely adopted approach in mechanistic interpretability research. Multiple training phases introduce many confounding factors, making it difficult to attribute observed behaviors to specific causes. By performing analysis after SFT, we can more clearly isolate the effects of data contamination and identify the resulting mechanistic changes.
> + SFT is a critical stage in modern LLM development. Data contamination during SFT is therefore a practically relevant scenario. Our work serves as a neccessary first step before extending to multi-stage training pipelines.
>
> ## 3. Response to Q3
> Our work goes beyond a simple "capacity reduction" in several key ways:
> + We identify mutual information decay as a signal specific to contaminated reasoning and explain it through low-dimensional computation.
> + Rank collapse implies only a coarse magnitude reduction (Inequality (14)), whereas the singular value partial order reveals a non-trivial direction-wise geometric degradation in the whitened space. This offers a more precise geometric explanation for mutual information decay than general rank collapse.
> + Our intervention analysis in Section 4 shows that suppressing the low-rank subspace restores the base-like reasoning behavior rather than disrupting the model's ability to answer. This counterintuitive finding indicates that it is not merely an efficient way learned by the model but an unexpected shortcut that actively overrides the reasoning process.
> ## 4. Experimental Setup Clarification (Weaknesses, Q4-Q10)
> ### 4.1. Datasets (Weaknesses)
> As described in Section 2.1, we primarily use GSM8K, where the test set is partitioned into (1) an injected subset used for SFT and (2) a held-out subset. To further avoid prior contamination, we also construct a synthetic random calculation dataset (MSCalc), with an example provided in Appendix A.
> ### 4.2. Activations and Layers (Weaknesses, Q4, Q5, Q7)
> All analyses are performed on input activations to transformer linear modules, primarily the pre-MLP activations (e.g., inputs to `mlp.up_proj`) across all layers. We also extract activations for other weight modules (such as Attention weights) and provide consistent results in Appendix C. As described in Section 2.1, the terms are defined as:
> + Activation $\boldsymbol{A}$: input activations to a linear module (e.g., `mlp.up_proj`)
> + Gradient $\boldsymbol{G}$: The gradients with respect to the module's output.
>
> For mutual information, the pipeline is:
> + Forward pass: collect activations.
> + Compute loss using solution and collect gradients.
> + Compute mutual information (Equation (1)). Results are reported across all layers in Figure 2.
> ### 4.3. Figures and Presentations (Q4, Q6, Q8)
> + We will expand figure captions to be fully self-sufficient. They will explicitly detail the quantities computed, the layers analyzed, and the core takeaways.
> + Y-axis (Figures 2 & 3): Leaked and clean curves share the y-axis per sub-figure to highlight the relative gap. Global y-axis alignment would compress curves across models.
> ### 4.4. What are we computing and on what (Q9, Q10)
> Section 3.2: We compute the relative error between the truncated output $\tilde{\boldsymbol{Y}}_k$ and the original output $\boldsymbol{Y}$ using Equation (4) and Equation (5). The input space is the activation space described in *4.2 of this rebuttal*.
>
> Section 3.3 (Table 1):
> + Compute $||\boldsymbol{W}^\top \boldsymbol{u}_i||$ using weight matrices and activation eigenvectors.
> + Report aggregated $R_{\text{worst}}$ (max across layers) and $R_{\text{med}}$ (median across layers), as described in the caption. This evaluates whether the weight significantly amplifies the tail eigenvectors and whether eigenspectrum concentration alone is sufficient to induce low-dimensional computation.

---

> > ### Author Rebuttal · Reviewer_hd3F · 2026-04-03
> >
> > The rebuttal clarifies some aspects of the setup, including the analyzed activations/gradients and the rationale for focusing on SFT as a controlled setting. However, my main concern was the lack of experimental specificity and reproducibility in the paper as written, and I do not think this is fully resolved by the rebuttal. Many essential details still need to be made explicit and self-contained in the submission itself, especially for each main figure/table: the exact module/layer analyzed, the activation and gradient definitions, the practical mutual information estimation procedure, and the aggregation across layers/examples. I am also still not fully persuaded that the paper clearly distinguishes genuinely non-trivial findings from expected memorization or shortcut effects induced by leaked SFT data. More importantly, while the authors motivate SFT as an isolating first step, modern LLM pipelines rarely stop at SFT and typically include additional stages such as preference optimization or reinforcement learning. Because of this, understanding whether later training stages preserve, weaken, or erase the reported signatures is important for assessing the broader validity and usefulness of the claims.

---

> > > ### Author Response · Authors · 2026-04-05
> > >
> > > We further clarify the three main concerns:
> > >
> > > # 1. Experimental Specificity
> > >
> > > + **Module or layer analyzed.** Our analysis is performed across all layers (e.g., Figs 2,3,5-8). The main text focuses on linear modules (e.g., `mlp.up_proj`), while Appendix C shows consistent generalization to other modules (e.g., `self_attn.q_proj`).
> > > + **Activation and gradient definitions.** As defined in Section 2.1 (Lines 88-93), for a linear module with weight $\boldsymbol{W}$ and output $\boldsymbol{Y}$, we analyze its input activation $\boldsymbol{A}$ and gradient $\partial \mathcal{L} / \partial \boldsymbol{Y}$, where $\mathcal{L}$ is the cross-entropy loss.
> > > + **Mutual information (MI) estimation.** MI is computed using the Gaussian formulation in Equation (1). Our code follows the pipeline clarified in previous rebuttal 4.2.
> > > + **Aggregation across layers and examples.** Table 1 aggregation is described in its caption. Figure 4 visualizes uniformly spaced layers for readability, while the analysis is conducted across all layers, and the minimum base-contaminated gap is 0.68% across all layers and all $k$. For sample aggregation, we report the mean with 95% confidence intervals (e.g., Figs 2,7).
> > >
> > > As we cannot modify the submitted paper at this stage, we will clarify these details in the revision and release code for reproducibility.
> > > # 2. Mechanistic Non-Triviality
> > > While data contamination may intuitively induce memorization or shortcuts, it does not address *how* shortcuts are implemented internally. General memorization may cause coarse rank collapse, but it does not necessitate simultaneous degradation across all canonical correlations. As discussed (Lines 289-292, right column), a mixed shift (some increase while others decrease) could also weaken associations. SVPO reveals that this degradation is not arbitrary but a non-trivial, structured contraction in the whitened space.
> > >
> > > Furthermore, MI decay provides a non-obvious signature that cleanly separates contaminated from normal reasoning. Without our insights, it would be difficult to conceive that subspace intervention (Section 4) could restore base-like reasoning and achieve SOTA performance with minimal harm to clean capabilities.
> > >
> > > Overall, uncovering a precise, robust, and actionable mechanism is what makes our findings non-trivial and valuable to the community.
> > > # 3. Mechanisms Persist Post-RL
> > > To address whether the identified mechanisms persist after additional training, we further apply reinforcement learning (RL) to the contaminated model and repeat the analysis on the post-RL models.
> > > ## 3.1. RL Training
> > > We train the contaminated Qwen2.5-1.5B-Instruct (leaked on the GSM8K test set) using **GRPO** for 2 epochs on two non-overlapping datasets (both exclude all leaked SFT samples):
> > > + GSM8K train set (same distribution; isolates the effect of RL)
> > > + MATH-lighteval train set (distribution shift; tests broader robustness)
> > >
> > > The resulting models are **RL-GSM** and **RL-MATH**.
> > > ## 3.2 Preserved Mechanisms
> > > We report minimum values of metrics across all 28 layers (serving as a worst-case), compared to the original contaminated model (denoted as **SFT**). Let $M$ denote SFT, RL-GSM, or RL-MATH.
> > > ### 3.2.1 Mutual Information Decay
> > > We compute the relative gap between the sample-averaged values as the metric (layer index omitted):
> > > $$
> > > \delta=\frac{\text{MI}(\text{clean}) -\text{MI}(\text{leaked})}{\text{MI}(\text{clean})}.
> > > $$
> > > $\Delta$ is the minimum $\delta$ across all layers:
> > >
> > > |Metric|SFT|RL-GSM|RL-MATH|
> > > |:---:|:---:|:---:|:---:|
> > > |**$\Delta$(%)**|54.44|55.83|57.34|
> > >
> > > Mutual information decay remains strong after RL.
> > > ### 3.2.2 Eigenspectrum Concentration
> > > We report the minimum value of $\text{rank}(\text{base})-\text{rank}(M)$ (layer index omitted) across all layers:
> > > + **SFT**: 11.93
> > > + **RL-GSM**: 11.59
> > > + **RL-MATH**: 11.18
> > >
> > > Spectral concentration persists after RL.
> > > ### 3.2.3 Low-Dimensional Computation
> > > We report the minimum $\Delta\text{RTE}=\text{RTE}(\text{base})-\text{RTE}(M)$ (layer index omitted) across all layers with $k=64$, alongside model $M$'s absolute RTE at that specific layer.
> > > |RL Models|RTE (%)|$\Delta$ RTE (%)|
> > > |:---:|:---:|:---:|
> > > |SFT|5.78|6.93|
> > > |RL-GSM|5.80|6.90|
> > > |RL-MATH|5.78|6.92|
> > >
> > > Low RTE values and positive RTE gaps indicate continued low-dimensional computation.
> > > ### 3.2.4 SVPO
> > > Following the paper, we compute the minimum singular value difference $d=\text{min}_i\left[\rho_i^{(\text{clean})}-\rho_i^{(\text{leaked})}\right]$ (layer index omitted; $i$ denotes the subscript of singular values) for each layer. We report $D$ as the minimum $d$ across all layers.
> > > |Metric|SFT|RL-GSM|RL-MATH|
> > > |:---:|:---:|:---:|:---:|
> > > |$D(\times 10^{-4})$|3.39|4.86|5.49|
> > >
> > > Positive values confirm that SVPO remains satisfied after RL.
> > > ## 3.3 Summary
> > > All key signatures are consistently preserved across both RL settings, indicating that the identified mechanisms are robust to subsequent training and provide useful insights for detecting or suppressing contamination in modern LLMs.

---

### Official Review · Reviewer_d7SC · 2026-03-11

**Soundness:** 4
**Presentation:** 4
**Significance:** 3
**Originality:** 3
**Overall Recommendation:** 5
**Confidence:** 3

**Summary:**

This paper investigates contaminated reasoning in large language models , where models show unreliable reasoning behavior on leaked benchmark data. Rather than only detecting contamination, the authors aim to understand its underlying mechanism.

They propose a geometric perspective called Narrowing Geometry. The key claim is that contaminated reasoning is associated with a decay in mutual information between model representations and gradients. Their analysis suggests that contaminated models exhibit eigenspectrum concentration, leading to a low-dimensional computation regime. Under leaked inputs, the coupling between representations and gradients weakens, appearing as a structural decay of singular values in a whitened representation space. The paper also provides theoretical arguments linking this geometry to reduced mutual information.

**Compliance With Llm Reviewing Policy:**

Affirmed.

**Final Justification:**

After carefully reading the authors’ rebuttal, I would like to thank the authors for their thorough and constructive responses. The rebuttal has addressed most of my initial concerns and clarified several important aspects of the paper, which has led me to revise my evaluation positively.

**Key Questions For Authors:**

1. What estimator is used to compute the mutual information between representations and gradients, and how sensitive are the results to the estimator choice or hyperparameters?

2. How well does the proposed restoration approach generalize across different models, tasks, or contamination scenarios? Have the authors tested it on models with different architectures or scales?

3. How can we be sure that the observed eigenspectrum concentration and mutual information decay are specific to contaminated reasoning rather than general phenomena occurring in normal reasoning or memorization behaviors?

**Limitations:**

yes

**Strengths And Weaknesses:**

Strengths

1. Data contamination in LLM evaluation has become a significant concern for reasoning benchmarks. The paper tackles the problem from a mechanistic understanding perspective, which is complementary to existing work focusing mainly on detection or benchmark construction.

2. The proposed concept of narrowing geometry provides a potentially novel way to analyze representation dynamics under contaminated inputs. The connection between eigenspectrum concentration, low-dimensional computation, and mutual information decay is conceptually appealing and could inspire further work on representation analysis in LLMs.

3. The paper attempts to bridge theoretical analysis and empirical observation, offering both a mathematical argument and experimental evidence that contaminated reasoning correlates with representation spectral changes.

Weaknesses

1. While the paper observes eigenspectrum concentration and mutual information decay in contaminated settings, it is not entirely clear whether these effects are specific to contamination or simply common phenomena in LLM representations.

2.  The central signal used in the paper is the mutual information between representations and gradients. However, estimating mutual information in high-dimensional neural representations is known to be difficult and often sensitive to estimator choice and hyperparameters. The paper does not sufficiently discuss potential estimator bias or provide robustness analysis across different estimation methods.

---

> ### Author Rebuttal · Authors · 2026-03-30
>
> ## 1. Response to Weaknesses
>
> ### 1.1. Response to W1
>
> We thank the reviewer for raising this important question. Our experiments provide comparisons between the base and contaminated models that address this concern. Specifically, **Figure 2** shows that mutual information decay appears only in the contaminated model, while the base model exhibits nearly identical mutual information between leaked and clean data. **Figure 3** demonstrates that eigenspectrum concentration is significantly stronger in contaminated models than in base models. The absence of these effects in the base model proves they are uniquely induced by contamination.
>
> ### 1.2. Response to W2
>
> We acknowledge that Gaussian mutual information assumes second-order statistics and may not capture higher-order dependencies. Since our analysis is based on second-order covariance geometry, Gaussian mutual information is not merely a tractable estimator, but the natural metric aligned with our framework.
>
> Importantly, our conclusions rely primarily on relative comparisons rather than absolute mutual information values. We also perform robust analysis with respect to covariance regularization (see *Response to Q1*), showing minimal variation across settings.
>
> ## 2. Response to Key Questions
>
> ### 2.1. Response to Q1
>
> We compute mutual information using Gaussian estimator, which admits a closed-form expression based on covariance matrices:
> $$
> I(A;G)=\frac{1}{2}\log\frac{\text{det} \boldsymbol{C}_A \text{det}\boldsymbol{C}_G}{\text{det} \boldsymbol{C}}.
> $$
> This formulation does not rely on neural estimators and therefore avoids estimator-specific training. Instead, it only requires empirical covariance estimation, making the computation stable and reproducible. Moreover, our analysis focuses primarily on relative comparisons (e.g., leaked vs. clean, base vs. contaminated), which are typically robust to potential bias in absolute mutual information values.
>
> In practice, the only hyperparameter in our implementation is a small covariance regularization coefficient $\epsilon$ for numerical stability: $\boldsymbol{C}^\prime=\boldsymbol{C}+\epsilon\boldsymbol{I}$. In the main experiments, we set $\epsilon$ to 1e-4. To evaluate sensitivity, we test $\epsilon$ in 1e-3, 1e-4, 1e-5 and evaluate the variation of the relative gap $$\left|\frac{\text{MI}(\text{clean})-\text{MI}(\text{leaked})} {\text{MI}(\text{clean})}\right|$$
> across the three settings. The summarized results across all layers are as follows:
> | Model | **Avg. Gap** | **Avg. Std.** | **(Avg. Std.) / (Avg. Gap)** |
> | --- | :---: | :---: | :---: |
> | Contaminated | 62.63% | 0.68% | 1.09% |
> | Base | 3.92% | 0.16% | 4.08% |
>
> The very small variation indicates that observed mutual information gaps are highly stable.
>
> ### 2.2. Response to Q2
>
> We test the method on Qwen and Llama. To further strengthen the generalization of our restoration approach, we apply it to a different model family and scale, namely **phi-4** (**14B parameters**), on the **GSM8K** and **HumanEval** datasets. The results are shown below:
>
> **GSM8K:**
> | Method | Restoration ($\uparrow$) | Retain ($\uparrow$) | Average ($\uparrow$) |
> | --- | :---: | :---: | :---: |
> | Contaminated | 0.00 | 100.00 | 50.00 |
> | Sampling | 35.71 | 92.13 | 63.92 |
> | TED | 28.57 | 90.63 | 59.60 |
> | LNE-Blocking | 28.57 | 88.81 | 58.69 |
> | **TSS (Ours)** | **42.86** | **94.06** | **68.46** |
>
> **HumanEval:**
> | Method | Restoration ($\uparrow$) | Retain ($\uparrow$) | Average ($\uparrow$) |
> | --- | :---: | :---: | :---: |
> | Contaminated | 0.00 | 100.00 | 50.00 |
> | Sampling | 34.38 | 80.00 | 57.19 |
> | TED | 31.25 | 85.71 | 58.48 |
> | LNE-Blocking | 37.50 | 82.86 | 60.18 |
> | **TSS (Ours)** | **62.50** | **94.28** | **78.39** |
>
> The consistent improvement on this model further reflects the effectiveness and generalization of the proposed method. We agree that extending this method to diverse contamination scenarios remains a promising direction for future work.
>
> ### 2.3. Response to Q3
> We clarify why our observed phenomena are highly specific to contaminated reasoning:
>
> + Figure 2 shows that the base model exhibits nearly identical mutual information for leaked and clean samples, indicating that mutual information decay does not arise in standard reasoning behavior. The differences of base and contaminated models in Figure 3 also support this conclusion.
>
> + Our work further provides causal evidence that distinguishes contamination-induced mechanisms from normal memorization or reasoning behaviors. If the identified subspace corresponds to normal memorization or reasoning capabilities, disrupting this subspace would be expected to harm model ability. However, suppressing the subspace (Section 4) does not harm model capability. Instead, it restores the contaminated model toward the base model's reasoning behavior, while maintaining high consistency on clean data. This suggests that the identified mechanism reflects a pathological shortcut induced by contamination.

---

> > ### Author Rebuttal · Reviewer_d7SC · 2026-04-03
> >
> > Thank you for your answer. I have no further questions.

---

> > > ### Author Response · Authors · 2026-04-03
> > >
> > > We sincerely thank the reviewer for the positive evaluation of our work. We greatly appreciate the valuable feedback and are glad that our clarifications have addressed your concerns.

---

### Official Review · Reviewer_FvYJ · 2026-03-12

**Soundness:** 3
**Presentation:** 4
**Significance:** 4
**Originality:** 4
**Overall Recommendation:** 5
**Confidence:** 4

**Summary:**

This work studies the geometry and statistics of the internal representations of LLMs which have “contaminated reasoning”. That is, the LLM has been trained on some of the test examples, and this makes it difficult to assess if reasoning is true generalization versus a memorized solution. This is a challenging task and highly relevant.
The authors employ mechanistic interpretability techniques in their study. The work first explores the Gaussian mutual information between inputs and their gradients. They find that this is diminished under leaked inputs. They connect this to a decrease in the effective dimension of the representations (ECI calculation, RTE evaluation). By combining a weight gradient derivation with the k-dim subspace expression, they develop an explanation for the weakened linear coupling between gradients and activations. In their causal intervention experiments with GSM8K, they can revert some of the effect of data leakage, supporting their findings. In particular, the causal experiments subtracts top components of the representation to prevent over-reliance on the low-dimensional representation.
The main weakness of this paper is that it is evaluated on a single domain, GSM8K. Moreover, it is tested on just 3 models. However, this is somewhat common for mechanistic interpretability papers.

**Compliance With Llm Reviewing Policy:**

Affirmed.

**Final Justification:**

The main weakness (limited reasoning domains tested) remains, but this is relatively common for mechanistic interpretability papers. The findings are well supported by evidence in the given domains, and this is a well-presented, technically solid paper. This paper would be interesting to readers at ICML, at it should be accepted. I maintain my rating of 5.

**Key Questions For Authors:**

Questions
Which model is it for Figure 2? I assume the smallest model, Qwen-1.5b-instruct due to the layer count and use of gradient.
It would be interesting to apply this method (TSS) to open-weight models on common benchmarks and to see which types of errors they make. Does the reasoning more closely resemble that of a true holdout set?

**Limitations:**

This work is only evaluated on one dataset, GSM8K. While this is somewhat the norm for mechanistic interpretability papers, the claims would be much stronger if additional (e.g., 3-5) datasets were tested for the same trends.

**Strengths And Weaknesses:**

Strengths
Very well motivated and relevant problem
Thorough investigation that is backed up with clear figures
Clear formulas, derivations, and statements


Weaknesses
Only evaluated on GSM8K reasoning set
Only 3 models tested/verified
I am skeptical of the claim (line 287) “In contrast, the original activation A also contains components orthogonal to Uk, which do not affect the module output and therefore exert no functional influence on the gradient.”. Just because a component is small does not mean it can’t have a strong influence on the model output. Adversarial examples are a counterexample to this claim.

---

> ### Author Rebuttal · Authors · 2026-03-30
>
> ## 1. Response to Weaknesses
>
> > *Weaknesses Only evaluated on GSM8K reasoning set Only 3 models tested/verified*
>
> We thank the reviewer for this helpful suggestion. Beyond GSM8K, we also construct a synthetic multi-step random calculation dataset (MSCalc), where we observe consistent mechanistic patterns (Appendix C), supporting the generality of our findings.
>
> To further examine generalization, we apply our restoration method to **Phi-4 (14B)** on the **HumanEval** dataset that evaluates the code generation capabilities of LLMs:
> | Method | Restoration ($\uparrow$) | Retain ($\uparrow$) | Average ($\uparrow$) |
> | --- | :---: | :---: | :---: |
> | Contaminated | 0.00 | 100.00 | 50.00 |
> | Sampling | 34.38 | 80.00 | 57.19 |
> | TED | 31.25 | 85.71 | 58.48 |
> | LNE-Blocking | 37.50 | 82.86 | 60.18 |
> | **TSS (Ours)** | **62.50** | **94.28** | **78.39** |
>
> The consistent improvement further reflects the effectiveness and generalization of the proposed method.
>
> > *I am skeptical of the claim ... Adversarial examples are a counterexample to this claim.*
>
> We thank the reviewer for this thoughtful comment. We agree that small components may indeed affect outputs in specially constructed scenarios, such as adversarial examples. As our analysis focuses on the natural data distribution, our intention here is to indicate that the contribution of orthogonal components to the output is negligible in expectation. We appreciate this helpful suggestion and will revise the wording to avoid potential ambiguity.
>
> ## 2. Response to Key Questions
> > *Questions Which model is it for Figure 2? I assume the smallest model, Qwen-1.5b-instruct due to the layer count and use of gradient.*
>
> The results in Figure 2 are based on the Qwen2.5-1.5B-Instruct model. We will clarify this in the figure caption to avoid ambiguity. Similar patterns are observed across other models, as reported in Appendix C.
>
> > *It would be interesting to apply this method (TSS) to open-weight models on common benchmarks and to see which types of errors they make. Does the reasoning more closely resemble that of a true holdout set?*
>
> We apply TSS to Phi-4 on the HumanEval dataset and find an example showing that the restored contaminated model exhibits reasoning similar to the base model:
>
> Prompt:
> ```python
> from typing import List
>
> def remove_duplicates(numbers: List[int]) -> List[int]:
> """ From a list of integers, remove all elements that occur more than once.
> Keep order of elements left the same as in the input.
> >>> remove_duplicates([1, 2, 3, 2, 4])
> [1, 3, 4]
> """
> ```
> Base model completion:
> ```python
>     seen = set()
>     result = []
>
>     for number in numbers:
>         if number not in seen:
>             seen.add(number)
>             result.append(number)
>
>     return result
> ```
> Restored model (TSS):
> ```python
>     return [x for i, x in enumerate(numbers) if numbers.index(x) == i]
> ```
> The prompt asks to remove all elements that occur more than once. Regarding the types of errors, the restored model shares the same logical mistakes as the base model. In this case, they both write logic that keeps the first occurrence of every element and drops the subsequent ones, despite using different implementations. This suggests that TSS shifts the contaminated model toward base-like reasoning behavior rather than merely altering outputs.
>
> ## 3. Response to Limitations
> > *This work is only evaluated on one dataset, GSM8K. While this is somewhat the norm for mechanistic interpretability papers, the claims would be much stronger if additional (e.g., 3-5) datasets were tested for the same trends.*
>
> We share the perspective on the importance of dataset diversity. We have taken an initial step by including the MSCalc dataset (Appendix C) and the HumanEval dataset (see *Response to Weaknesses*). We consider broader evaluation across more datasets an important direction for future work.

---

> > ### Author Rebuttal · Reviewer_FvYJ · 2026-04-02
> >
> > Thank you for your clarifications to my questions and for directing me to the additional results on MSCalc and now HumanEval. I still believe that the number of datasets is small and the domains considered (mostly math reasoning) is relatively narrow, and I plan to maintain my rating of 5.

---

> > > ### Author Response · Authors · 2026-04-03
> > >
> > > We thank the reviewer for maintaining the positive assessment. We agree that expanding to more datasets and broader domains would further strengthen the generality of our findings. We consider this as an important direction for future work.

---

### Official Review · Reviewer_c7yK · 2026-03-13

**Soundness:** 4
**Presentation:** 3
**Significance:** 3
**Originality:** 3
**Overall Recommendation:** 5
**Confidence:** 3

**Summary:**

- The authors address the deep-rooted problem of benchmark task contamination in training data, discussing how contamination is characterized inside language models through the lens of Gaussian mutual information between input activations and their output gradients.
- First, they confirm that mutual information is low exclusively in the combination of leaked data and a leaked model. In other words, the eigenspectrum of the input activations is concentrated, meaning that inference on leaked data can be performed within a narrow subspace. They also mathematically characterize this mechanism through the difference in gradients (i.e., whether the model has seen the data during training) and the singular value partial order (SVPO) condition.
- Furthermore, they propose a method to correct the model so that it uses a broader subspace for inference even on inputs containing leaked data, and confirm that the corrected leaked model makes mistakes on leaked data in the same “correct” way that a base model would.

**Compliance With Llm Reviewing Policy:**

Affirmed.

**Final Justification:**

I continue to believe this paper is suitable for acceptance.

Regarding the lack of discussion of influence functions (and their subsequent works) in the manuscript, which is closely related to this work: it was not resolved during the discussion phase, and I believe it should be explicitly addressed in the camera-ready version. I also agree with the other reviewers' points (e.g. regarding reproducibility), and more broadly, there is room for improvement in the paper's overall presentation.

That said, I find this to be an interesting paper overall, and I have no objection to its acceptance at ICML. I hope the authors will prepare an updated manuscript that incorporates the feedback from each of the reviewers.

**Key Questions For Authors:**

- Can the choice of mutual information between input activations and output gradients as the measure (line 124, left column) be justified? Were other quantities considered as alternatives, whether between inputs and outputs, or as measures of their relationship?
- What information is lost by using Gaussian mutual information (i.e., by looking only up to second-order statistics)?
- If the message of this paper were to be summarized very roughly, it might be something like “sequences the model has seen before can be output smoothly.” In the sense that inference can be performed efficiently within a low-rank data subspace, is this not actually a desirable property of modern generative models?

**Limitations:**

yes

**Strengths And Weaknesses:**

- Strengths: This is an interesting paper that discusses what happens inside a model when it performs inference on its training data, offering a unified perspective based on the idea that the representation space becomes narrower. The linear-algebra-based argument is intuitive, and the persuasiveness is high because each hypothesis is accompanied by both empirical validation and formal analysis. Moreover, the scope of the proposed method is not limited to LLMs and data contamination; it has the potential to contribute to the broader topic of the influence of training data on trained models.
- Weaknesses: The connection to interpretability research on the relationship between gradients and inputs, originating from influence functions (Koh and Liang, ICML'17), is unclear. Given that the scope of this paper concerns the influence of training data on trained models, the substantive connection to such research is likely a point of considerable interest to readers.

---

> ### Author Rebuttal · Authors · 2026-03-27
>
> ## 1. Response to Weaknesses
> > *The connection to interpretability research on the relationship between gradients and inputs, originating from influence functions (Koh and Liang, ICML'17), is unclear. Given that the scope of this paper concerns the influence of training data on trained models, the substantive connection to such research is likely a point of considerable interest to readers.*
>
> Thank you for pointing out the connection to influence functions (Koh and Liang, ICML'17). We agree that this is an important line of work closely related to our study. At a high level, both influence function analysis and our work aim to understand the influence of training data on models. Moreover, both approaches leverage gradients as a key analytical tool for tracing such influence.
>
> However, there are also important distinctions. Influence functions primarily focus on data attribution, identifying which individual training samples influence a model's prediction through perturbations. This perspective explains model behavior by tracing predictions back to specific training examples.
>
> In contrast, our work adopts a mechanistic interpretability perspective. Rather than attributing predictions to specific training examples, we analyze the relationship between activations and gradients, revealing how contaminated training alters the geometry of internal representations. This shifts the focus from data to the internal computation.
>
> We believe these two perspectives are complementary. Influence function methods provide a data-centric view, whereas our work offers a mechanism-centric view. Together, these perspectives contribute to a more comprehensive understanding of training data influence in language models.
>
> ## 2. Response to Key Questions
>
> ### 2.1. Response to Q1
> > *Can the choice of mutual information between input activations and output gradients as the measure (line 124, left column) be justified? Were other quantities considered as alternatives, whether between inputs and outputs, or as measures of their relationship?*
>
> We measure the dependency between activations and gradients because this relationship reflects how strongly the forward computation contributes to the prediction. Activations characterize forward representations conditioned on the input, while gradients indicate output sensitivity. A weakened dependency suggests that the forward computation becomes less relevant to the prediction, which is consistent with a shift in the reasoning mechanism.
>
> Practically, the mutual information clearly distinguishes the contaminated model from the base model, and leaked data from clean data, validating its utility as a signature of contaminated reasoning.
>
> Regarding alternative quantities, measures based on input-output relationships primarily capture behavior-level correlations, whereas the activation-gradient relationship provides a more suitable signal for mechanistic changes. Other metrics such as Hilbert-Schmidt Independence Criterion (HSIC) depend on the kernel choice or additional hyperparameters. This may introduce additional complexity. In contrast, the Gaussian mutual information admits a closed-form solution, making it both simple and robust in practice.
>
> ### 2.2. Response to Q2
> > *What information is lost by using Gaussian mutual information (i.e., by looking only up to second-order statistics)?*
>
> Gaussian mutual information relies on the covariance matrices. In particular, higher-order statistical dependencies may not be captured. Since computing exact mutual information in the high-dimensional space of LLMs is challenging, we adopt the Gaussian mutual information as a practical approximation. Moreover, our analysis, such as eigenspectrum concentration, is naturally characterized by second-order statistics. Therefore, Gaussian mutual information is well aligned with the phenomena we study.
>
> ### 2.3. Response to Q3
> > *If the message of this paper were to be summarized very roughly, it might be something like “sequences the model has seen before can be output smoothly.” In the sense that inference can be performed efficiently within a low-rank data subspace, is this not actually a desirable property of modern generative models?*
>
> We address this concern through intervention analysis in Section 4. If the low-rank subspace is an efficient and desirable pathway, suppressing it should disrupt the model's ability to answer. However, after removing it, the model restores the reasoning behavior closer to that of the base model rather than failing. This suggests that the low-rank subspace is not merely a desirable property but an unexpected shortcut that actively overrides the reasoning process.
> Therefore, while low-rank computation may sometimes be desirable, in our settings it reflects a pathological mechanism associated with contaminated reasoning.

---

> > ### Author Rebuttal · Reviewer_c7yK · 2026-04-04
> >
> > Thank you for your response.
> >
> > Regarding the weakness, I found it difficult to accept that the proposed method and influence functions (along with their subsequent works) are “complementary.” Both lines of research discuss the impact of data on model predictions, don't they?
> >
> > In any case, I think the submitted paper presents interesting research, and I will maintain my positive assessment.

---

> > > ### Author Response · Authors · 2026-04-04
> > >
> > > Thank you for the clarification and for maintaining the positive assessment. We agree that at a high level, both lines of work aim to understand the impact of training data on model behavior. Our intention was to highlight that influence functions provide a data attribution perspective, whereas our work focuses on representation-level mechanistic changes. Ultimately, we agree that both lines of work share the same high-level objective, despite differences in specific methodologies. Thank you again for helping us improve the clarity of our paper.

---

### Decision · Program_Chairs · 2026-04-30

**Decision:**

Accept (regular)

**Comment:**

This paper investigates the mechanistic causes of data contamination in large language models by analyzing the relationship between representations and gradients. The reviewers appreciate the motivation and the geometric perspective. The experimental setup lacks the specificity required for reproducibility and the results primarily focus on a single reasoning domain. One reviewer remains unconvinced that the observed signals are distinct from standard memorization or rank collapse. The authors provided additional reinforcement learning experiments during the rebuttal. The authors should follow the reviewer hd3F's suggestion and clarify what is being computed and how as part of the paper.